# Diagenetic evolution of fault zones in Urgonian microporous carbonates, impact on reservoir properties (Provence – SE France).

*Irène Aubert [a], Philippe Léonide [a], Juliette Lamarche [a], Roland Salardon [a]*

*[a] Aix-Marseille Université, CNRS, IRD, Cerege, Um 34, 3 Place Victor Hugo (Case 67), 13331 Marseille Cedex 03, France*

Microporous carbonate rocks form important reservoirs with permeability variability depending on sedimentary, structural and diagenetic factors. Carbonates are very sensitive to fluid-rock interactions that lead to secondary diagenetic processes like cementation and dissolution capable of modifying the reservoir properties. Focusing on fault-related diagenesis, the aim of this study is to identify impact of the fault zone on reservoir quality. This contribution focuses on two fault zones east to La Fare Anticline (SE France) cross-cutting Urgonian microporous carbonates. 122 collected samples along four transects orthogonal to fault strike were analysed. Porosity values have been measured on 92 dry plugs. Diagenetic elements were determined through the observation of 92 thin sections using polarized light microscopy, cathodoluminescence, red alizarin, SEM and stable isotopic measurements ($\delta^{13}$C and $\delta^{18}$O). Eight different calcite cementation stages and two micrite micro-fabrics were identified. As a main result, this study highlights that the two fault zones acted as drains canalizing low-temperature fluids at their onset, and induced calcite cementation which strongly altered and modified the local reservoir properties.

## 1. INTRODUCTION

Microporous carbonates form important reservoirs (Deville de Periere et al., 2017; Lambert et al., 2006; Sallier, 2005; Volery et al., 2009), with porosity values up to 35% (Deville de Periere et al., 2011). Due to their heterogeneous properties which depends on sedimentary, structural and diagenetic factors, microporous carbonates may determine a high variability of reservoir permeability (Bruna et al., 2015; Deville de Periere et al., 2011, 2017; Eltom et al., 2018; Florida et al., 2009; Hollis et al., 2010). Moreover, fault zones in carbonates play an important role on reservoir properties (Agosta et al., 2010, 2012; Caine et al., 1996; Delle Piane et al., 2016; Ferraro et al., 2019; Knipe, 1993; Laubach et al., 2010; Rossetti et al., 2011; Sinisi et al., 2016; Solum et al., 2010; Solum and Huisman, 2016; Tondi, 2007; Wu et al., 2019). Fault zones are complex structures composed of damage zones and the fault core encompassed by the host rock (Caine et al., 1996; Chester and Logan, 1986, 1987; Hammond and Evans, 2003). Faults can act as barriers (Agosta et al., 2010; Tondi, 2007), drains (Agosta et al., 2007, 2008, 2012; Delle Piane et al., 2016; Evans et al., 1997; Molli et al., 2010; Reches and Dewers, 2005; Sinisi et al., 2016; Solum and Huisman, 2016), or mixed hydraulic behaviour zones (Matonti et al., 2012) depending on their architecture and diagenetic evolution. Because of their hydraulic properties, fault zones influence the fluid flows in the upper part of Earth's crust (Bense et al., 2013; Evans et al., 1997; Knipe, 1993; Sibson, 1994; Zhang et al., 2008), and are capable of increasing the fluid-rock interactions. Carbonates are very sensitive to these interactions, which lead to

diagenetic secondary processes like cementation and dissolution (Deville de Periere et al., 2017; Fournier and Borgomano, 2009; Lambert et al., 2006). Fault-related diagenesis locally modifies the initial rock properties (mineralogy and porosity), and therefore the reservoir properties (Hodson et al., 2016; Knipe, 1993; Knipe et al., 1998; Laubach et al., 2010; Woodcock et al., 2007). In case of a polyphasic fault zone, repeating fluid pathways-barriers behaviour in times leads to very complex diagenetic modifications. The initial vertical and lateral compartmentalization of microporous limestones is, therefore, accentuated by fault-related diagenesis. Hence, understanding faulting processes and diagenesis is crucial for a better exploration and production in carbonates. Urgonian microporous carbonates of Provence, are made of facies and reservoir properties analogue to Middle East microporous carbonate reservoirs (Thamama, Kharaib and Shuaiba Formations; Borgomano et al. 2002, 2013; Sallier 2005; Fournier et al. 2011; Leonide et al. 2012; Léonide et al. 2014). Although Urgonian microporous carbonates of Provence are analogue to Middle East reservoirs, the analogy can be extended to other faulted microporous carbonate reservoirs. To have a better comprehension of diagenetic modifications linked to fault zones on these rocks, the aim of this paper is (i) to determine the diagenetic evolution of polyphasic fault zones; (ii) to identify their impact on reservoir properties and (iii) to link the fault evolution with the fluid flow and geodynamic history of the basin.

## 2. GEOLOGICAL CONTEXT

We studied two faults cross-cutting microporous Valanginian-to-Early Aptian Urgonian carbonates of the South-East Basin (Provence-SE France) deposited along the southern margin of the Vocontian Basin (Léonide et al., 2014; Masse and Fenerci Masse, 2011). The "Urgonian" platform carbonates (Masse, 1976) reached their maximum areal extension during the late Hauterivian–Early Aptian (Masse and Fenerci-Masse, 2006). From Albian to Cenomanian, the regional Durancian uplift triggered exhumation of Early Cretaceous carbonates, bauxitic deposition (Guyonnet-Benaize et al., 2010; Lavenu et al., 2013; Léonide et al., 2014; Masse and Philip, 1976; Masse, 1976), and development E-W-trending extensional faults (Guyonnet-Benaize et al., 2010; Masse and Philip, 1976). During the Late-Cretaceous times, platform environment led to a transgressive rudist platform deposition (Philip, 1970). From Late Cretaceous to Eocene, the convergence between Iberia plate and Eurasia plates (e.g. Bestani 2015, and references therein) caused a regional N-S shortening (e.g. Molliex et al. 2011 and references therein). The so-called "Pyrénéo-Provençal" shortening, gave rise to E-W-trending north-verging thrust faults and ramp folds (e.g. Bestani et al. 2016, and references therein). From Oligocene to Miocene, the area underwent extension associated to Liguro-Provençal Basin opening (e.g. Demory et al. 2011). During Mio-Pliocene times, the Alpine shortening dimly impacted the studied area (Besson, 2005; Bestani, 2015), and reactivated the "Pyrénéo-Provençal" structures (Champion et al., 2000; Molliex et al., 2011).

We studied two faults pertaining to a Km-scale fault system on the E-W-trending La Fare anticline near Marseille (Fig. 1A). The southern limb of this anticline dips 25° S, and is constituted by Upper Hauterivian, Lower Barremian and Santonian rocks (Fig. 1B). The Upper Barremian carbonates are composed, from bottom to top, of a 120  m-thick calcarenitic unit swith cross-beddings, a 40 m-thick massive coral-rich calcarenite unit, and an upper 10 m-thick

calcarenite unit (Masse, 1976; Matonti et al., 2012; Roche, 2008). Santonian age coarse rudist limestones uncomfortably overlap the Barremian carbonates (Fig. 1A).

The Castellas fault zone is a 2.14 km-long left-lateral strike-slip fault, N060 to 070-trending and 40° to 80°N-dipping (Fig. 2A, 2B; table 1) composed horse structures, secondary faults and lenses (Fig. 2A, 2C; Aubert et al. (2019b)). The second investigated fault zone "D19" is composed of 5 sub-fault zones (F1 to F5) restricted in a 50m-long extension (Fig. 2E, 2H; Table 1; (Aubert et al., 2019a)). Sub-faults are organised into two sets. The first one comprises F3 and F4, N040 to N055-trending, 60-80°NW-dipping (orange traces on Fig. 2F). Set 2 is N030-trending, dipping 80°E, with left-lateral strike-slip slickensides pitch 20 to 28°SW (F1, F2, F5, red traces on Fig. 2F).

The internal structure of both fault zones results from three distinct tectonic events:
- the Durancian uplift dated as mid-Cretaceous leading to extension and to normal *en echelon* normal faults. The Castellas fault nucleated during this first extensional event and bear early dip-slip normal striations (Matonti et al., 2012),
- the Early Pyrenean compression with N000° to N170°-trending $\sigma_H$ (see cited references in Espurt et al. 2012) whichreactivated the Castellas fault as sinistral (Matonti et al., 2012) and ledto the newly-formed strike-slip faults of the D19 outcrop (Aubert et al., 2019a).
- the Pyrenean to Alpine folding, triggering the 25°S tilting of the strata and fault zones. Faults of the D19 outcrop were reactivated while the Castellas fault tilting led to an apparent present-day reverse throw (Aubert et al., 2019a).

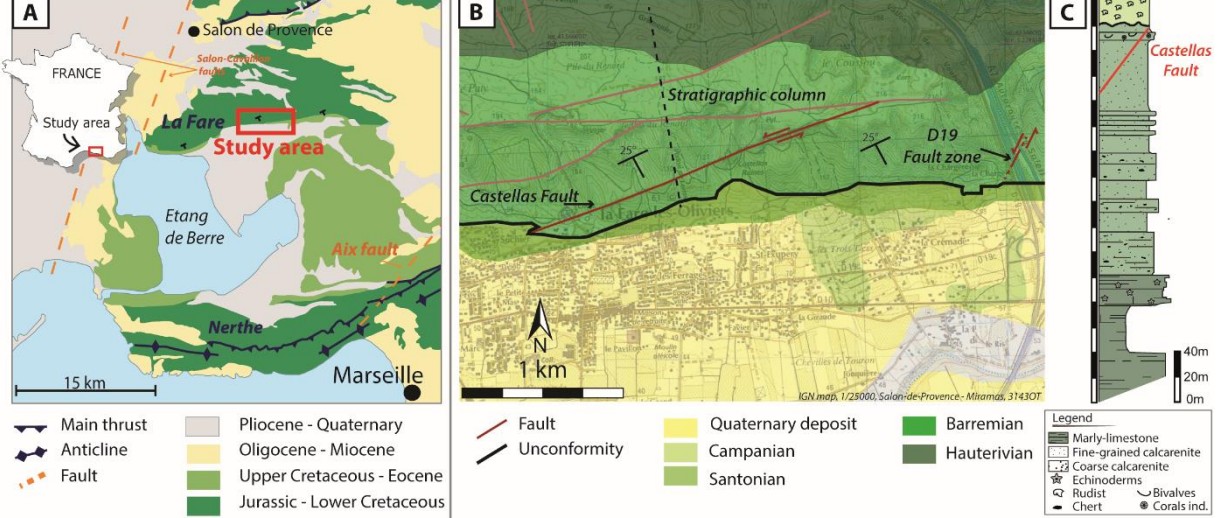

***Figure 1*** *: Geological context of the study area. A: geological map of Provence, B: Simplified structural map with the location of the Castellas fault and the stratigraphic column (black dashed line); C: Stratigraphic column of exposed Cretaceous carbonates (modified from Roche, 2008)*

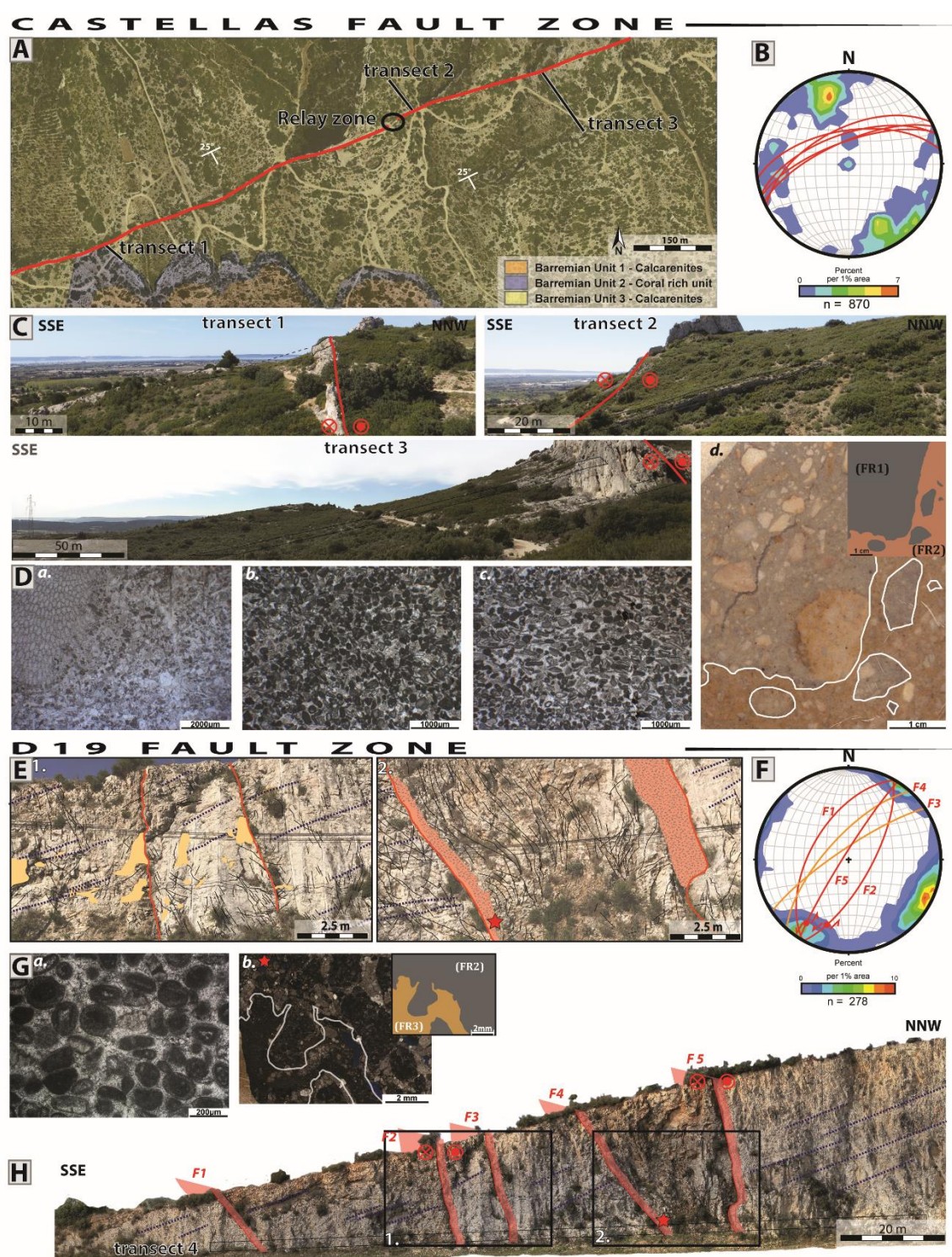

114

*Figure 2* : A: Castellas fault map on aerial photo with position of the studied transects and the relay zone; B: stereographic projections of poles to fractures (density contoured) and faults (red lines) (Allmendinger et al., 2013; Cardozo and Allmendinger, 2013); C: Photos of transects 1 to 3; D: Photomicrographs of carbonate host-rock facies (a) transect 1 coral rich unit, (b) transect 2 calcarenites, (c) transect 3 calcarenites and (d) fault rocks 1 and 2 (FR1 and FR2); E: Pictures of D19 outcrop F: Stereographic projections of poles to fractures (density contoured), set one faults (orange line) and set 2 faults (red line) G: Photomicrograph of host rock facies (a) and of fault rocks (b; red stars on the pictures)); H: D19 outcrop including the five faults F1 to F5.


***Table 1***: *structural properties of the fault zones.*

| Fault zones | Fault | Direction | Dip | Dip direction | Pitch striation | Fault core thickness | Fault Rocks | | |
|---|---|---|---|---|---|---|---|---|---|
| | | | | | | | FR1 | FR2 | FR3 |
| Castellas | Castellas | 060 - 070 | 40 to 80 | N | 14 W - | 0 to 4 m | sparsely present | majoritarely present | / |
| D19 | F1 | 030 | 56 | W | | 20 | / | <10 cm | / |
| | F2 | 029 | 70 | E | 28 S | 10 to 15 | / | ? | variable thickness |
| | F3 | 056 | 80 | N | | 0 to 15 | / | ? | ? |
| | F4 | 042 | 70 | W | | 20 | / | in the clasts of FR3 | variable thickness |
| | F5 | 032 | 85 | N | 20 SW | 50 to100 | / | / | variable thickness |


## 3. DATA BASE

We performed 4 transects across the Castellas Fault and the D19 Fault (Fig. 2). Transect 1 is
located along the coral rich unit 2. This lithostratigraphic unit is essentially composed of
peloidal grains and bioclasts (corals, bivalves and stromatoporidae; Fig. 2Da). Transects 2 and
3 cross-cut in unit 3, made of fine calcarenites with peloidal grains and a rich fauna
(foraminifera, bivalves, ostracods and echinoderms; Fig. 2Db, c). Transect 4 was conducted
along the D19 outcrop (Fig. 3), which exposes Barremian outer platform bioclastic calcarenite
with current ripples. The grains are mainly peloids with minor amounts of bioclasts (solidary
corals, bryozoans, bivalves and some rare miliolids; Fig. 2Ga).

The different tectonic events impacted the fault zone and fault core structure. Three different
fault rock types were identified in the fault core of the two investigated fault zones ( see Aubert
et al. 2019a; Matonti et al., 2012). Fault rock 1 (FR1) results from the extensional activation of
the Castellas fault during Durancian uplift. It is a cohesive breccia composed of sub-rounded to
rounded clasts from the nearby damage zone and <30% of fine-grey matrix (Fig. 2Dd). Fault
rock 2 (FR2), is linked to the strike reactivation of the Castellas fault and to the onset of D19
fault zone during the Pyrenean shortening. FR2 presents two morphologies depending on the
fault zones. Within Castellas fault, FR2 is an un-cohesive breccia with an orange/oxidized
matrix with angular to sub-rounded clasts belonging to the nearby damage zone and from FR1
(Fig. 2Dd). In the D19 fault zone, FR2 is a cohesive breccia with rounded clasts of the damage
zone and a white cemented matrix (Fig. 2Gb). Fault rock 3 (FR3) is formed by the reactivation
of D19 fault zone. The timing of D19 fault reactivation is tricky to determine as it can be related
both to Pyrenean or alpine shortening.  FR3 is composed of angular to sub-angular clasts from
FR2 and from the nearby damage zone dispersed in an orange/oxidized matrix (<20%) (Fig.
2Gb).

## 4. METHODS

The data set comprises 122 samples, 62 from Castellas and 60 from D19 outcrops, collected
along the 4 transects. Porosity values were measured on 92 dry plugs with a Micromeritics
AccuPyc 1330 helium pycnometer. Microfacies were determined on 92 thin sections.
Impregnation with a blue-epoxy resin allowed us to decipher the different pore types. Thin
sections were coloured with a solution of hydrochloric acid, Alizarin red S and potassium
ferricyanide to distinguish carbonate minerals (calcite and dolomite). Thin sections were
analysed using cathodoluminescence to discriminate the different generation of calcite cements.

The paragenetic sequence was defined based on superposition and overlap principles observed on thin sections using a Technosyn Cold Cathode Luminescence Model 8200 Mk II coupled to an Olympus_BH2 microscope and to a Zeiss_MR C5. Micrite micro-fabric and major element composition of two samples from the fault zone, two from the host rock and 1 from the D19 karst infilling were measured using PHILIPS XL30 ESEM with a beam current set at 20 kV on fresh sample surfaces and on thin sections. To determine stable carbon and oxygen isotopes ($\delta^{13}$C and $\delta^{18}$O), 204 microsamples (<5 mg) were drilled, 194 of them were micro-drilled from polished thin sections with an 80 μm diameter micro-sampler (Merkantec Micromill) at the VU University (Amsterdam, The Netherlands). We micro-sampled bulk rocks (57), sparitic cements (101), fault rocks (9) and micrite (27). The Bulk rock values are related to a non-selective sampling giving information on the whole rock isotopic values. These values do not capture the signature of isolated cement (Swart, 2015). Carbon and oxygen isotopic values were acquired with Thermo Finnigan Delta + mass spectrometer equipped with a GASBENCH preparation device at VU University Amsterdam. The internationally used standard IAEA-603, with official values of +2.46‰ for $\delta^{13}$C and -2.37‰ for $\delta^{18}$O, is measured as a control standard. The standard deviation (SD) of the measurements is respectively < 0.1‰ and < 0.2 ‰ for $\delta^{13}$C and $\delta^{18}$O, respectively. Ten whole rock samples were analysed using a Gasbench II connected to a Thermo Fisher Delta V Plus mass spectrometer at the FAU University (Erlangen, Germany). Measurements were calibrated by assigning $\delta^{13}$C values of +1.95‰ to NBS19 and -47.3‰ to IAEA-CO9 and $\delta^{18}$O values of -2.20‰ to NBS19. All values are reported in per mil relative to V-PDB.

## 5. RESULTS

### 1.    MICROPOROSITY AND POROSITY

Porosities measuredon the 92 samples show a strong decrease towards the fault core (Fig. 3): dropping from more than 10% in the host carbonates (mean: 15%, SD: 2.68 for Castellas and mean 12.3%, SD: 2.52 for D19) to less than 5% within fault zones (mean: 4.8%, SD: 2.07 for Castellas and mean: 3.16%, SD: 2.35 for D19).

Along transects, some porosity variations occur as follows:

-    North of the Castellas fault, along the 60 m-long transect 2 the porosity is constantly lower than7% (mean of 4.4%, SD: 1.53; Fig. 3A).
-    South of the Castellas fault, the reduced porosity zone is wider than 40 m in transect 3 and 30 m in transect 1 (Fig. 3A). In a 10 m-thick zone from the fault plane, porosity reduction occurs with lower values in transect 1 (average 4.9%) than in transect 3 (average 5.6%).

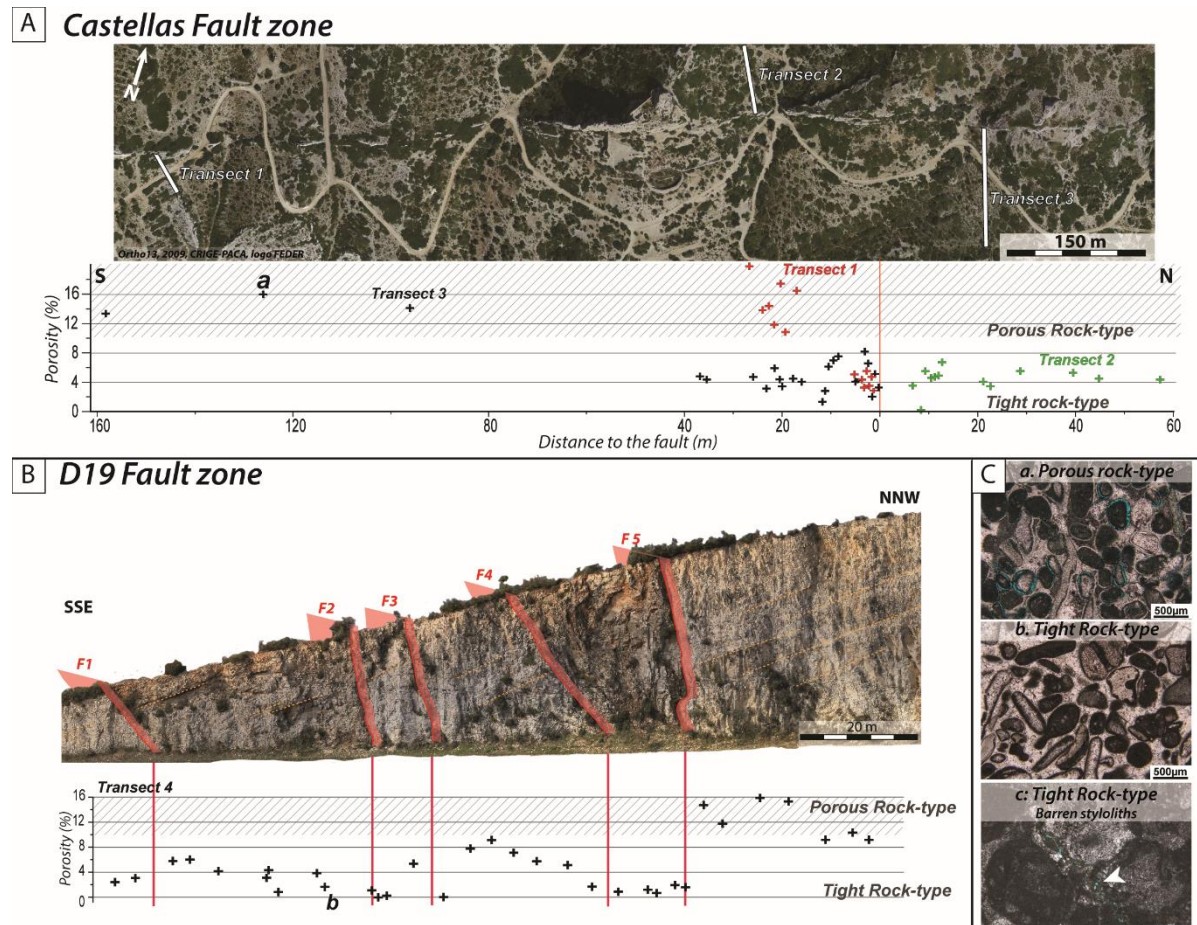

**Figure 3**: *A: Castellas fault zone aerial view (Ortho13, 2009, CRIGE-PACA, logo FEDER) and porosity values measured along transect 1 (Red Cross), transect 2 (green cross) and transect 3 (black cross); B: porosity values measured along D19 fault zone; C: Pore types in the host rock (a) and in the fault zones (b and c).*

In the D19 fault zone, the lowest porosity values are found in narrow zones around the faults (less than 2 m-wide) and in the lens between F4 and F5. Though, this porosity decrease is not homogeneous in fault zone and high values are found north of F1 and F3 (Fig. 3B).

Microscope observation of thin sections impregnated with blue-epoxy resin allowed to identified a porous rock-type with $\phi > 10\%$ mainly in micritized grains as microporosity and moldic porosity (Fig. 3Ca), and a tight rock-type with $\phi < 5\%$, where the porosity is mostly linked to barren stylolites (Fig. 3Cb, c).

## 2. DIAGENETIC PHASES

### a. Micrite micro-fabric

Micritised bioclasts, ooids and peloids were observed after SEM analysed of two fault zones samples and two host rock samples. Two micro-fabrics of micrite occur with specific crystal shape, sorting and contacts according to Fournier et al. (2011). Within both fault zones, the micrite is tight, with compact subhedral mosaic crystals of less than 10 µm-wide (MF1; Fig. 4A, B). In the host rock, the micrite is loosely packed, and partially coalescent with puntic rarely serrate, subhedral to euhedral crystals of less than 5 µm-wide (MF3; Fig. 4C, D, E). MF1 correlates with low porosity values ($< 5\%$), while MF3 with higher porosity ($> 10\%$).

### b. Diagenetic cements

Eight different cement stages were identified (Fig. 5). The red stain links to Alizarin Red S coloration and shows that all visible cements made of calcite, which exhibit variable characteristics (morphology, luminescence, size and location).

The first two cement phases occur in both fault zones. The first cement (C0) is non-luminescent isopachous calcite of constant thickness (~10 µm) around grains (Fig. 5A). The second cement (C1) is divided in two sub-phases: a non-luminescent calcite, C1a, with a crystal size ranging from 50 µm to more than 200 µm, a dog-tooth morphology in intergranular spaces, and a bright luminescence calcite, C1b, covering C1a with a maximum thickness of 100 µm (Fig. 5A, B, D, G). C1b also fills micro-porosity in micritised grains (Fig. 5B). C1b areal occurrence strongly increases in Castellas fault zone.

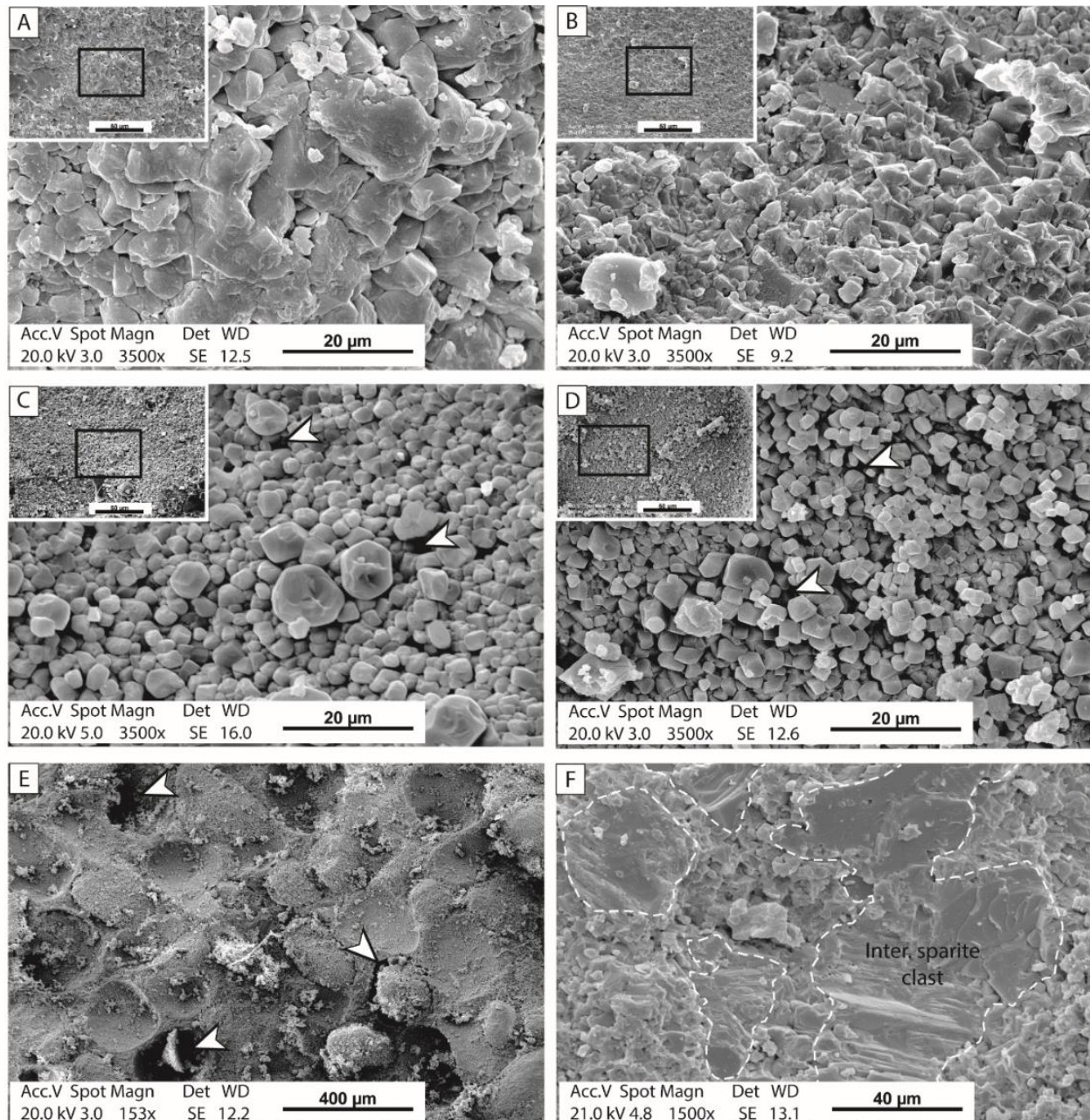

***Figure 4*** *: MEB pictures of micrite micro-fabric and microporosity (white arrows); A. MF1 micrite micro-fabric in Castellas fault zone (2.5 m to fault plane); B: MF1 micrite micro-fabric within D19 fault zones (2 m away from F5 fault plane); C: MF3 micrite micro-fabric within Castellas host rock (188 m away from the fault plane); D: MF3 micrite micro-fabric within D19 host rock (95 m away from F5 fault plane); E: D19 host rock moldic porosity; F: Karst infilling.*

Five cements or replacive phases extensively occur in the Castellas sector and rarely in the D19
outcrop:

- C2 is a sparitic cement, with dull-orange luminescent crystals sized of maximum 100
229          µm only found in veins of the fault core (Fig. 5B). SEM measurements show the Si and
Al in the C2 veins. Most of Si crystals are automorphic and have black luminescence.

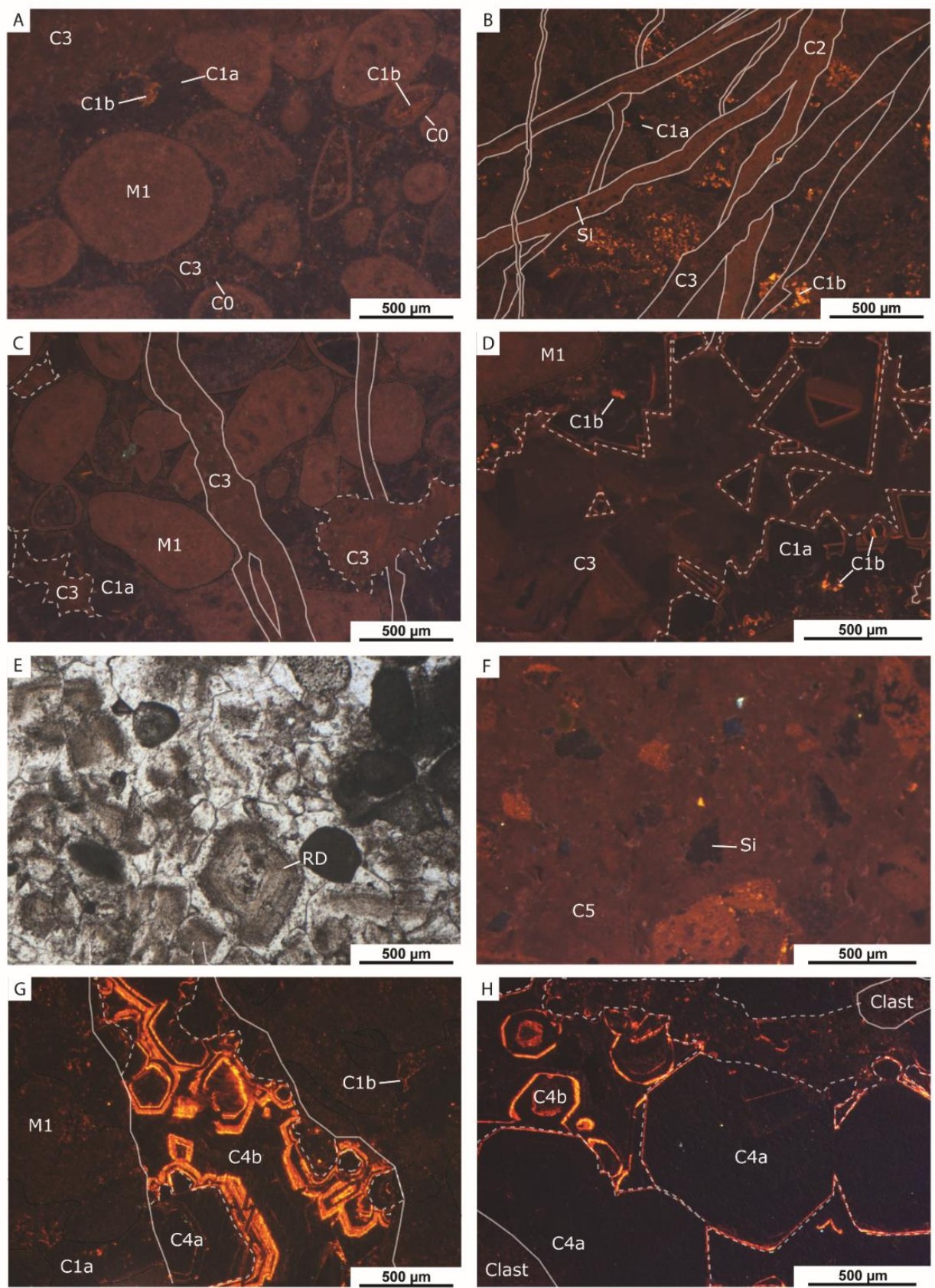

***Figure 5*** *: Thin-sections under cathodoluminescence; A: Calcarenite in transect 3 with micritised grain (M1), and intergranular volume cemented with C1 a and b and C3; B: C2 (with Si) and C3 veins affecting Castellas FR1 clasts with micritszed grains cemented by C1b; C: C3 veins, cements and intergranular volumes in Castellas fault zone; D: C1 (a and b) and C3 cementing moldic porosity of transect 3 calcarenite; E: FR1 matrix with phantom of cloudy appearance replacive dolomite (RD); F: FR1 matrix de-dolomitized by C5 containing quartz grains; G: C4 (a and b) cementing vein of D19 fault zone; H: matrix of D19 FR2 cemented by C4 (a and b).*

-    C3 is a blocky calcite with non to red-dull luminescence in veins, moldic and
intergranular pores (Fig. 5B, C, D). This cement also occurs in few veins of D19 sectors
but is not restricted only to the fault zone.
-    Phantoms of planar-e (euhedral) dolomite crystals (Sibley and Gregg, 1987) with a
maximum size of 500 µm affect the matrix of FR1 (Fig. 5E). They are vestiges of a
previous dolomitization phase. They have a cloudy appearance caused by solid micritic
inclusion inside crystals and can be considered as replacive dolomite (RD; Machel,
2004). Within the FR1 matrix, an important concentration of angular grains of quartz
with a maximum size of 300 µm is noticed (Fig. 5F).
-    A blocky calcite C4 (referred to as S2 in Aubert et al. (2019a)) is mainly present in veins
of the D19 outcrop, in matrix of FRA, and intergranular and moldic pores (Fig. 5G, H).
This cement shows zonation of non-luminescent and bright luminescent bands and can
be divided in two sparitic sub-phases: C4a which is non-luminescent with some highly
luminescent bands and C4b is bright luminescent with some thin non-luminescent
zones. C4a occurs in lesser proportion in some veins along transect 2 and 3 of the
Castellas fault.
-    A sparitic cement C5, with a red-dull luminescence replaces the RD phase (Fig. 5F).
**c.  Additional diagenetic features**
In addition to cementation phases, other diagenetic elements affected both fault zones. Karst
infilling occurs in the F2 fault zone of the D19 outcrop. It is composed of well-sorted grains
deposited in laminated layers. This karst deposit presents a stack of alternating micrite-rich and
grain-rich layersfrom the latter composed of former blocky calcite belonging to dissolved
grainstones. The laminated layers are affected by veins and stylolites; some of these are
deformed due to the grain fall on sediments. Micritic layers have been observed under SEM,
and the micrite appeared tight with compact subhedral mosaic crystals (Fig. 4F). We observed
oxide filling mainly in the Castellas area in dissolution voids affecting C1a, C1b and C3
cementation phases and in D19 in karstic fill. The areal amount of oxides increase close to
stylolites.
**3.    CARBON AND OXYGEN ISOTOPES**
Isotope measurements were realized on samples collected along transects of the fault zones. A
hundred and eighty-nine measurements of C and O isotopes were performed on 16 samples and
32 thin sections (Fig. 6A, table 2).
Sampling was done in bulk rock (66), sparitic cement (101; veins, intergranular volume and
fault rock cements) and in fault rocks (10) in order to determine their isotopic signature. Isotopic
values range from -10.40‰ to -3.65‰ for $\delta^{18}O$ and from -7.20‰ to +1.42‰ for $\delta^{13}C$ (Fig. 6A,
B, table 2). The bulk rock values range from -9.18‰ to -4.34‰ for $\delta^{18}O$ and from -4.80‰ to
+1.19‰ for $\delta^{13}C$ (Fig. 6A, table 2). These values are split in two sets. Set 1 includes transect
transect 1 and 3 of the Castellas Fault. Bulk values range from -6.07‰ to -4.34‰ for $\delta^{18}O$ and
from -1.41‰ to +1.19‰ for $\delta^{13}C$. Set 2 includes transect 2 (Castellas) and transect 4 (D19).
Bulk values range from -9.18‰ to -5.20‰ for $\delta^{18}O$ and from -4.80‰ to -0.60‰ for $\delta^{13}C$ (Fig.
6B, table 2). In transect 3, the isotopic values only slightly vary, ranging from -6.13‰ to -
4.50‰ for $\delta^{18}O$ and from -1.41‰ to +0.47‰ for $\delta^{13}C$ respectively (Fig. 6C, table 2). On the
contrary, values are more variable along the D19 transect; they range from -9.18‰ to -5.20‰

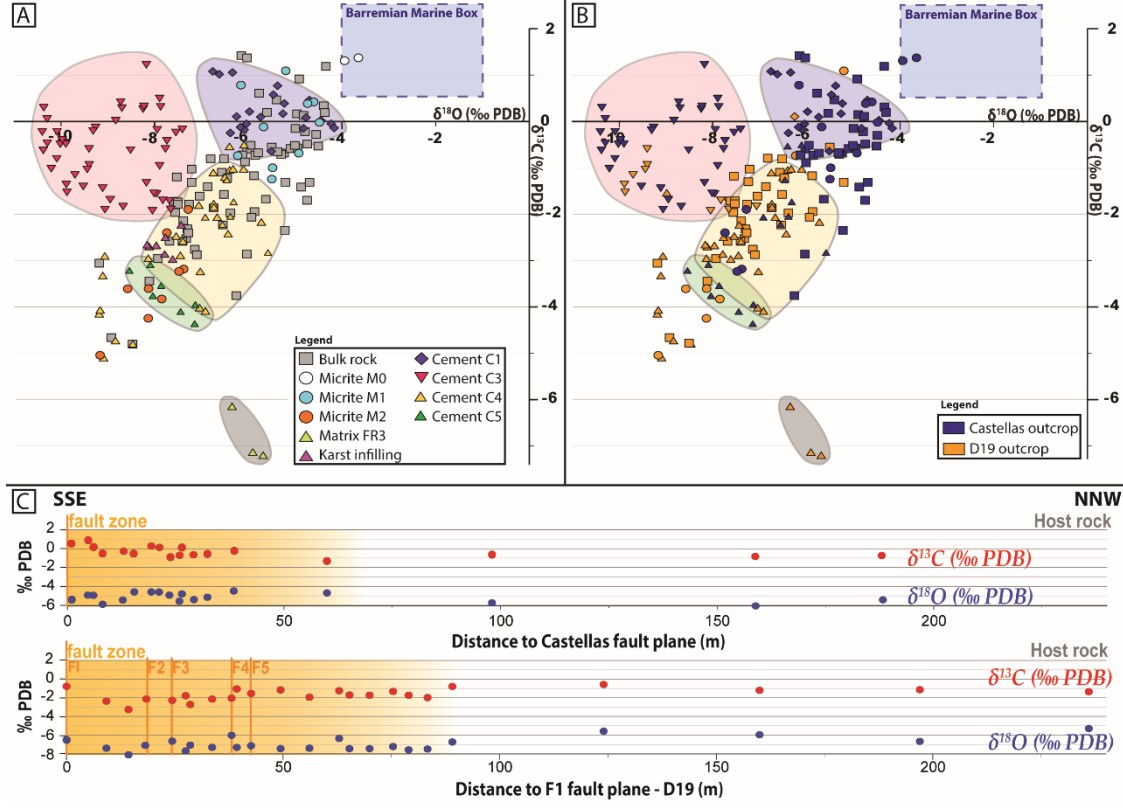

*Figure 6* : *Isotopic values of δ13C and δ18O measured on bulk rock, cement phases, and micrite. Range values of "Urgonian marine box" from Moss & Tucker (1995) and Godet et al. (2006); A: set of values sorted by the nature of diagenetic phases and B: values sorted by the fault zone; C: lateral evolution of δ13C and δ18O bulk isotopic values in Castellas (top) and in D19 (bottom) fault zones.*

for $\delta^{18}O$ and from -4.80‰ to -0.60‰ for $\delta^{13}C$ (Fig. 6C, table 2). The $\delta^{13}C$ values depletes approaching to faults, especially south of F2.

Isotopic values of cements filling veins, intergranular volumes, karst infillings, and fault rocks are divided into 5 groups (Fig. 6A, table 2):

- Isotopic values of C1 cement fluctuates from -6.76‰ to -4.45‰ for $\delta^{18}O$ and from -1.28 to +1.08‰ for $\delta^{13}C$;

- Isotopic values of C3 cement ranges from -10.40‰ to -6.73‰ for $\delta^{18}O$ and from -2.09 to +1.22‰ for $\delta^{13}C$;

- Isotopic values of C4 cement in FR1 and FR2 matrix and in karst infillings ranges from -9.18‰ to -4.60‰ for $\delta^{18}O$ and from -5.10‰ to -0.74‰ for $\delta^{13}C$ with a positive covariance between $\delta^{18}O$ and $\delta^{13}C$. FR2 matrix values (from -6.55 to -7.06‰ for $\delta^{18}O$ and from -1.10 to -2.24‰ for $\delta^{13}C$) present slightly less depleted values than karst infillings with mean values of -7.83‰ and -2.53‰ respectively for $\delta^{18}O$ and $\delta^{13}C$ respectively. (Fig. 6A). In the Castellas fault, 4 isotopic values from two veins are enriched with means of -6.25 and -4.2‰ for $\delta^{18}O$ -0.64 and -0.09‰ for $\delta^{13}C$ having similar positive covariance than the other C4 values;

- Isotopic values of C5 cement, sampled in FR1 matrix display mean of -7.49‰ for $\delta^{18}O$ and -4.01‰ for $\delta^{13}C$ (Fig. 6A);

-    Isotopic values of FR3 matrix with a mean of -5.98‰ for $\delta^{18}O$ and -6.83‰ for $\delta^{13}C$

(Fig. 6A)


***Table 2****: Carbon and oxygen isotope values of bulk carbonates for Castellas fault zone and D19*
*fault zones. B: bulk measurement; M: micrite value; C1, C3, C4, C5: cement isotopic value;*
*FR: fault rock isotopic value.*

| Transect | Sample | $\delta^{13}C$ (‰VPDB) | $\delta^{18}O$ (‰VPDB) | Class | Distance to F. (m) |
|---|---|---|---|---|---|
| Transect 1 (Cast.) | 201 | 1,19 | -4,34 | B | 1,3 |
| Transect 1 (Cast.) | 201 | 1,02 | -6,62 | C1 | 1,3 |
| Transect 1 (Cast.) | 201 | 1,31 | -3,94 | M | 1,3 |
| Transect 1 (Cast.) | 201 | 1,37 | -3,65 | M | 1,3 |
| Transect 1 (Cast.) | 213 | -0,68 | -5,24 | B | 22,7 |
| Transect 1 (Cast.) | 213 | -0,58 | -5,10 | B | 22,7 |
| Transect 1 (Cast.) | 213 | -0,18 | -6,09 | C1 | 22,7 |
| Transect 1 (Cast.) | 213 | 0,03 | -4,45 | C1 | 22,7 |
| Transect 1 (Cast.) | 213 | 0,09 | -4,77 | C1 | 22,7 |
| Transect 1 (Cast.) | 213 | -2,09 | -6,92 | C4 | 22,7 |
| Transect 1 (Cast.) | 213 | -0,68 | -4,92 | M | 22,7 |
| Transect 2 (Cast.) | c3b17 | -0,52 | -5,95 | B | 4,6 |
| Transect 2 (Cast.) | c3b17 | -2,07 | -6,38 | C4 | 4,6 |
| Transect 2 (Cast.) | c3b7 | -0,64 | -5,51 | B | 9,3 |
| Transect 2 (Cast.) | c3b26 | -3,76 | -6,26 | B | 22,6 |
| Transect 2 (Cast.) | c3b26 | -2,85 | -5,58 | C4 | 22,6 |
| Transect 2 (Cast.) | c3b26 | -1,31 | -4,69 | B | 57,3 |
| Transect 2 (Cast.) | c3b7 | -1,76 | -6,31 | C1 | 57,3 |
| Transect 2 (Cast.) | c3b7 | -1,28 | -6,46 | C1 | 57,3 |
| Transect 2 (Cast.) | c3b26 | -2,35 | -5,22 | M | 57,3 |
| Transect 2 (Cast.) | c3b26 | -1,70 | -4,75 | M | 57,3 |
| Transect 3 (Cast.) | 327 | -0,24 | -7,55 | C3 | 0,3 |
| Transect 3 (Cast.) | 325 | -1,90 | -9,06 | C3 | 0,3 |
| Transect 3 (Cast.) | 325 | -1,69 | -8,95 | C3 | 0,3 |
| Transect 3 (Cast.) | 327 | -3,11 | -8,09 | C4 | 0,3 |
| Transect 3 (Cast.) | 327 | 0,47 | -5,40 | B | 1,0 |
| Transect 3 (Cast.) | 327 | -0,18 | -7,95 | C3 | 1,0 |
| Transect 3 (Cast.) | 327 | -0,17 | -7,41 | C3 | 1,0 |
| Transect 3 (Cast.) | 328 | 0,10 | -5,74 | C1 | 1,6 |
| Transect 3 (Cast.) | 328 | -1,32 | -8,18 | C3 | 1,6 |
| Transect 3 (Cast.) | 328 | -0,59 | -7,77 | C3 | 1,6 |
| Transect 3 (Cast.) | 328 | -0,42 | -7,74 | C3 | 1,6 |
| Transect 3 (Cast.) | 328 | -0,13 | -9,26 | C3 | 1,6 |
| Transect 3 (Cast.) | 328 | 0,02 | -8,83 | C3 | 1,6 |

| | | | | | |
|---|---|---|---|---|---|
| Transect 3 (Cast.) | 328 | 0,29 | -8,70 | C3 | 1,6 |
| Transect 3 (Cast.) | 328 | 0,42 | -8,73 | C3 | 1,6 |
| Transect 3 (Cast.) | 328 | 0,50 | -7,89 | C3 | 1,6 |
| Transect 3 (Cast.) | 328 | 1,22 | -8,18 | C3 | 1,6 |
| Transect 3 (Cast.) | 333 | -1,84 | -8,67 | C3 | 1,6 |
| Transect 3 (Cast.) | 333 | -0,96 | -7,89 | C3 | 1,6 |
| Transect 3 (Cast.) | 328 | -0,14 | -4,17 | C4 | 1,6 |
| Transect 3 (Cast.) | 328 | -0,05 | -4,23 | C4 | 1,6 |
| Transect 3 (Cast.) | 329 | 0,16 | -4,95 | B | 2,4 |
| Transect 3 (Cast.) | 333 | -0,25 | -6,38 | C1 | 4,6 |
| Transect 3 (Cast.) | 333 | -0,12 | -6,17 | C1 | 4,6 |
| Transect 3 (Cast.) | 333 | -0,62 | -8,52 | C3 | 4,6 |
| Transect 3 (Cast.) | 333 | -0,12 | -5,67 | M | 4,6 |
| Transect 3 (Cast.) | 333 | -0,02 | -4,48 | M | 4,6 |
| Transect 3 (Cast.) | 333 | 0,42 | -4,60 | M | 4,6 |
| Transect 3 (Cast.) | 337 | 0,19 | -5,59 | B | 9,5 |
| Transect 3 (Cast.) | 302 | -0,53 | -4,50 | B | 11,8 |
| Transect 3 (Cast.) | 302 | -0,49 | -4,74 | B | 11,8 |
| Transect 3 (Cast.) | 302 | -0,62 | -10,38 | C3 | 11,8 |
| Transect 3 (Cast.) | 302 | -0,49 | -10,02 | C3 | 11,8 |
| Transect 3 (Cast.) | 305 | 0,33 | -4,38 | B | 16,0 |
| Transect 3 (Cast.) | 306 | 0,21 | -4,35 | B | 17,8 |
| Transect 3 (Cast.) | 307 | -0,01 | -4,46 | B | 18,2 |
| Transect 3 (Cast.) | 308 | -0,57 | -4,95 | B | 20,0 |
| Transect 3 (Cast.) | 308 | -1,44 | -9,11 | C3 | 20,0 |
| Transect 3 (Cast.) | 308 | -0,23 | -10,40 | C3 | 20,0 |
| Transect 3 (Cast.) | 308 | -0,22 | -10,08 | C3 | 20,0 |
| Transect 3 (Cast.) | 309 | -1,41 | -4,87 | B | 20,5 |
| Transect 3 (Cast.) | 309 | -0,52 | -5,01 | B | 20,5 |
| Transect 3 (Cast.) | 309 | -0,15 | -4,82 | C1 | 20,5 |
| Transect 3 (Cast.) | 309 | -1,56 | -7,96 | C3 | 20,5 |
| Transect 3 (Cast.) | 309 | -1,55 | -8,01 | C3 | 20,5 |
| Transect 3 (Cast.) | 312 | 0,12 | -4,81 | B | 23,2 |
| Transect 3 (Cast.) | 314 | -0,71 | -5,30 | B | 25,9 |
| Transect 3 (Cast.) | 314 | -0,80 | -10,09 | C3 | 25,9 |
| Transect 3 (Cast.) | 314 | -0,49 | -9,90 | C3 | 25,9 |
| Transect 3 (Cast.) | 314 | -0,47 | -10,29 | C3 | 25,9 |
| Transect 3 (Cast.) | 314 | -0,40 | -9,97 | C3 | 25,9 |
| Transect 3 (Cast.) | 314 | 0,06 | -10,30 | C3 | 25,9 |
| Transect 3 (Cast.) | 316 | -1,24 | -5,50 | B | 29,2 |
| Transect 3 (Cast.) | 316 | -1,00 | -5,48 | B | 29,2 |
| Transect 3 (Cast.) | 316 | -0,22 | -4,79 | B | 29,2 |
| Transect 3 (Cast.) | 316 | -1,02 | -10,21 | C3 | 29,2 |
| Transect 3 (Cast.) | 316 | -0,18 | -9,31 | C3 | 29,2 |
| Transect 3 (Cast.) | 316 | 0,30 | -10,37 | C3 | 29,2 |
| Transect 3 (Cast.) | 318 | -0,28 | -4,53 | B | 35,4 |

| Transect | Sample | $\delta^{13}C$ (‰VPDB) | $\delta^{18}O$ (‰VPDB) | Class | Distance to F1 (m) |
|---|---|---|---|---|---|
| Transect 3 (Cast.) | 320 | -0,68 | -5,79 | B | 96,1 |
| Transect 3 (Cast.) | 322 | -0,88 | -6,07 | B | 158,0 |
| Transect 3 (Cast.) | 323 | -0,65 | -5,37 | B | 188,0 |
| Castellas (ZF1) | Z1,1 | 0,17 | -5,26 | C1 | 0,0 |
| Castellas (ZF1) | Z1,1 | 0,39 | -5,23 | C1 | 0,0 |
| Castellas (ZF1) | Z1,1 | 0,46 | -4,70 | C1 | 0,0 |
| Castellas (ZF1) | Z1,2 | 0,21 | -5,98 | C1 | 0,0 |
| Castellas (ZF1) | Z1,1 | -0,55 | -6,40 | C4 | 0,0 |
| Castellas (ZF1) | Z1,1 | -0,52 | -6,10 | C4 | 0,0 |
| Castellas (ZF1) | Z1,2 | -4,12 | -7,45 | C5 | 0,0 |
| Castellas (ZF1) | Z1,2 | -0,15 | -4,99 | FR | 0,0 |
| Castellas (ZF1) | Z1,2 | 0,39 | -4,73 | M | 0,0 |
| Castellas (ZF1) | Z1,2 | 0,61 | -5,77 | M | 0,0 |
| Castellas (ZF1) | Z1,1 | 0,78 | -6,16 | M | 0,0 |
| Castellas (ZF2) | Z2,2 | 0,77 | -5,38 | C1 | 0,0 |
| Castellas (ZF2) | Z2,7 | -1,40 | -9,52 | C3 | 0,0 |
| Castellas (ZF2) | Z2,7 | -4,38 | -7,15 | C5 | 0,0 |
| Castellas (ZF2) | Z2,7 | -3,97 | -7,13 | C5 | 0,0 |
| Castellas (ZF2) | Z2,7 | -3,78 | -8,04 | C5 | 0,0 |
| Castellas (ZF2) | Z2,7 | -3,56 | -7,86 | C5 | 0,0 |
| Castellas (ZF2) | Z2,7 | -3,24 | -7,48 | C5 | 0,0 |
| Castellas (ZF2) | Z2,7 | -3,23 | -8,54 | C5 | 0,0 |
| Castellas (ZF2) | Z2,2 | 0,58 | -5,47 | FR | 0,0 |
| Castellas (ZF2) | Z2,2 | 0,92 | -4,91 | FR | 0,0 |
| Castellas (ZF2) | Z2,7 | -1,68 | -5,63 | FR | 0,0 |
| Castellas (ZF2) | Z2,7 | -2,24 | -6,55 | FR | 0,0 |
| Castellas (ZF2) | Z2,7 | -3,18 | -7,38 | M | 0,0 |
| Castellas (ZF2) | Z2,7 | -2,86 | -6,03 | FR | 1,0 |
| Castellas (ZF5) | Z5,4 | 0,27 | -8,25 | C3 | 0,0 |
| Castellas (ZF5) | Z5,4 | 0,31 | -7,87 | C3 | 0,0 |
| Castellas (ZF5) | Z5,4 | 0,32 | -8,23 | C3 | 0,0 |
| Castellas (ZF5) | Z5,4 | 1,06 | -6,34 | C1 | 0,4 |
| Castellas (ZF5) | Z5,4 | 1,08 | -6,76 | C1 | 0,4 |
| Castellas (ZF5) | Z5,4 | 1,05 | -7,13 | FR | 0,4 |
| Castellas (ZF5) | Z5,4 | 1,37 | -6,03 | FR | 0,4 |
| Castellas (ZF5) | Z5,4 | 1,42 | -6,15 | FR | 0,4 |
| **Transect** | **Sample** | $\delta^{13}C$ (‰VPDB) | $\delta^{18}O$ (‰VPDB) | **Class** | **Distance to F1 (m)** |
| Transect 4 (D19) | 3B | -0,81 | -6,52 | B | 0,0 |
| Transect 4 (D19) | 3B | -1,20 | -6,50 | C1 | 0,0 |
| Transect 4 (D19) | 3B | -1,02 | -6,33 | C1 | 0,0 |
| Transect 4 (D19) | 3B | 0,11 | -6,25 | C1 | 0,0 |
| Transect 4 (D19) | 3B | -0,74 | -6,23 | M | 0,0 |
| Transect 4 (D19) | 9 | -2,32 | -7,30 | B | 9,2 |

| | | | | | |
|---|---|---|---|---|---|
| Transect 4 (D19) | 13a | -3,44 | -8,11 | B | 14,3 |
| Transect 4 (D19) | 13a | -2,96 | -7,93 | B | 14,3 |
| Transect 4 (D19) | 13C | -2,97 | -7,62 | M | 14,3 |
| Transect 4 (D19) | 13C | -2,86 | -7,79 | M | 14,3 |
| Transect 4 (D19) | 13C | -2,70 | -8,12 | M | 14,3 |
| Transect 4 (D19) | 13C | -2,67 | -7,96 | M | 14,3 |
| Transect 4 (D19) | 13C | -2,66 | -8,16 | M | 14,3 |
| Transect 4 (D19) | 13C | -2,50 | -7,77 | M | 14,3 |
| Transect 4 (D19) | 13C | -1,54 | -8,98 | M | 14,3 |
| Transect 4 (D19) | 17 | -2,58 | -7,68 | B | 18,7 |
| Transect 4 (D19) | 14A | -1,97 | -6,38 | B | 18,7 |
| Transect 4 (D19) | 14A | -1,87 | -6,74 | B | 18,7 |
| Transect 4 (D19) | 15B | -2,23 | -7,43 | B | 18,7 |
| Transect 4 (D19) | 17 | -1,05 | -6,40 | C1 | 18,7 |
| Transect 4 (D19) | 14A | -1,77 | -6,74 | C1 | 18,7 |
| Transect 4 (D19) | 14A | -2,42 | -6,43 | C4 | 18,7 |
| Transect 4 (D19) | 14A | -2,06 | -6,67 | C4 | 18,7 |
| Transect 4 (D19) | 21 | -2,23 | -6,54 | B | 24,4 |
| Transect 4 (D19) | RSG | -1,90 | -7,66 | B | 28,4 |
| Transect 4 (D19) | RSG | -1,70 | -7,83 | B | 28,4 |
| Transect 4 (D19) | RSD | -2,87 | -7,10 | B | 29,5 |
| Transect 4 (D19) | RSD | -2,76 | -7,14 | B | 29,5 |
| Transect 4 (D19) | RSD | -0,93 | -9,40 | C3 | 29,5 |
| Transect 4 (D19) | RSF1 | -2,40 | -7,28 | B | 34,7 |
| Transect 4 (D19) | RSF2 | -2,14 | -7,39 | B | 34,7 |
| Transect 4 (D19) | RSF2 | -1,78 | -7,27 | B | 34,7 |
| Transect 4 (D19) | RSF1 | -1,03 | -9,44 | C3 | 34,7 |
| Transect 4 (D19) | RSF2 | -1,93 | -8,05 | C3 | 34,7 |
| Transect 4 (D19) | RSF2 | -0,59 | -9,40 | C3 | 34,7 |
| Transect 4 (D19) | RSF2 | -2,95 | -8,14 | C4 | 34,7 |
| Transect 4 (D19) | RSE 1 | -2,53 | -7,33 | B | 35,0 |
| Transect 4 (D19) | RSE 2 | -2,59 | -7,41 | B | 35,0 |
| Transect 4 (D19) | RSE 1 | -1,71 | -7,68 | C3 | 35,0 |
| Transect 4 (D19) | RSE 2 | -1,84 | -6,73 | C3 | 35,0 |
| Transect 4 (D19) | 57 | -2,07 | -5,93 | B | 38,1 |
| Transect 4 (D19) | 57 | -1,94 | -5,87 | B | 38,1 |
| Transect 4 (D19) | 57 | -1,83 | -7,06 | C3 | 38,1 |
| Transect 4 (D19) | 57 | -1,10 | -6,75 | C3 | 38,1 |
| Transect 4 (D19) | 57 | -4,02 | -7,04 | C4 | 38,1 |
| Transect 4 (D19) | 57 | -2,17 | -5,72 | C4 | 38,1 |
| Transect 4 (D19) | 57 | -1,58 | -6,52 | FR | 38,1 |
| Transect 4 (D19) | 57 | -7,20 | -5,68 | M | 38,1 |
| Transect 4 (D19) | 57 | -7,13 | -5,90 | M | 38,1 |
| Transect 4 (D19) | 28b | -1,03 | -7,21 | B | 39,3 |
| Transect 4 (D19) | 28b | -1,03 | -6,10 | C3 | 39,3 |
| Transect 4 (D19) | 28b | -4,09 | -6,92 | C4 | 39,3 |

| | | | | | |
|---|---|---|---|---|---|
| Transect 4 (D19) | 28b | -2,58 | -7,40 | C4 | 39,3 |
| Transect 4 (D19) | 28b | -2,47 | -7,54 | C4 | 39,3 |
| Transect 4 (D19) | 30a | -1,61 | -7,04 | B | 42,6 |
| Transect 4 (D19) | 30a | -1,41 | -6,87 | B | 42,6 |
| Transect 4 (D19) | 30a | -3,23 | -7,03 | C4 | 42,6 |
| Transect 4 (D19) | 30a | -2,89 | -7,45 | C4 | 42,6 |
| Transect 4 (D19) | 24a | -1,21 | -7,52 | B | 51,1 |
| Transect 4 (D19) | 27b | -1,92 | -7,48 | B | 57,9 |
| Transect 4 (D19) | 31 | -1,24 | -6,44 | B | 65,0 |
| Transect 4 (D19) | 32 | -1,75 | -7,50 | B | 67,4 |
| Transect 4 (D19) | 34 | -1,79 | -7,49 | B | 72,2 |
| Transect 4 (D19) | 36 | -1,32 | -7,21 | B | 77,8 |
| Transect 4 (D19) | 38 | -1,73 | -7,59 | B | 81,5 |
| Transect 4 (D19) | 62 | -1,96 | -7,56 | B | 86,0 |
| Transect 4 (D19) | 42 | -0,81 | -6,80 | B | 91,9 |
| Transect 4 (D19) | 63 | -0,55 | -5,50 | B | 124,0 |
| Transect 4 (D19) | 64 | -1,17 | -5,88 | B | 160,0 |
| Transect 4 (D19) | 65 | -1,10 | -6,57 | B | 197,0 |
| Transect 4 (D19) | 66 | -1,31 | -5,21 | B | 236,0 |
| Transect 4 (D19) | 60a | -3,06 | -9,18 | B | 255,2 |
| Transect 4 (D19) | 60B | -4,80 | -8,47 | B | 255,2 |
| Transect 4 (D19) | 60B | -4,66 | -8,92 | B | 255,2 |
| Transect 4 (D19) | 61 | -1,53 | -9,87 | C3 | 255,2 |
| Transect 4 (D19) | 61 | -1,36 | -9,89 | C3 | 255,2 |
| Transect 4 (D19) | 60a | -1,15 | -9,70 | C3 | 255,2 |
| Transect 4 (D19) | 60a | -3,32 | -9,11 | C4 | 255,2 |
| Transect 4 (D19) | 60B | -5,10 | -9,09 | C4 | 255,2 |
| Transect 4 (D19) | 60B | -4,73 | -8,84 | C4 | 255,2 |
| Transect 4 (D19) | 60B | -4,15 | -9,18 | C4 | 255,2 |
| Transect 4 (D19) | 60B | -4,07 | -9,16 | C4 | 255,2 |
| Transect 4 (D19) | 60B | -2,90 | -9,06 | C4 | 255,2 |
| Transect 4 (D19) | 60a | -3,83 | -7,85 | M | 255,2 |
| Transect 4 (D19) | 60B | -5,04 | -9,17 | M | 255,2 |
| Transect 4 (D19) | 60B | -4,25 | -8,14 | M | 255,2 |
| Transect 4 (D19) | 60B | -3,61 | -8,58 | M | 255,2 |
| Transect 4 (D19) | 60B | -3,61 | -8,13 | M | 255,2 |

## 6. DISCUSSION

### 1. DIAGENETIC EVOLUTION OF THE FAULT ZONES

The chronological relations between cements can be established via to cross-cutting relations and inclusion principles. Indeed, the veins filled with cement C2 cross-cut C1a and C1b cements (Fig. 5B). Thus, C2 cementation post-dated C1 cement. C3 veins cross-cut the C2 veins, but are included within FR1 clasts (Fig. 5B). Hence, C3 cement is prior to FR1 development but after C2 cementation. The fault rock 1 (FR1) is related to the first extensional fault activity, consequently, C1, C2 and C3 cementation phases occurred prior to the proper fault plane and

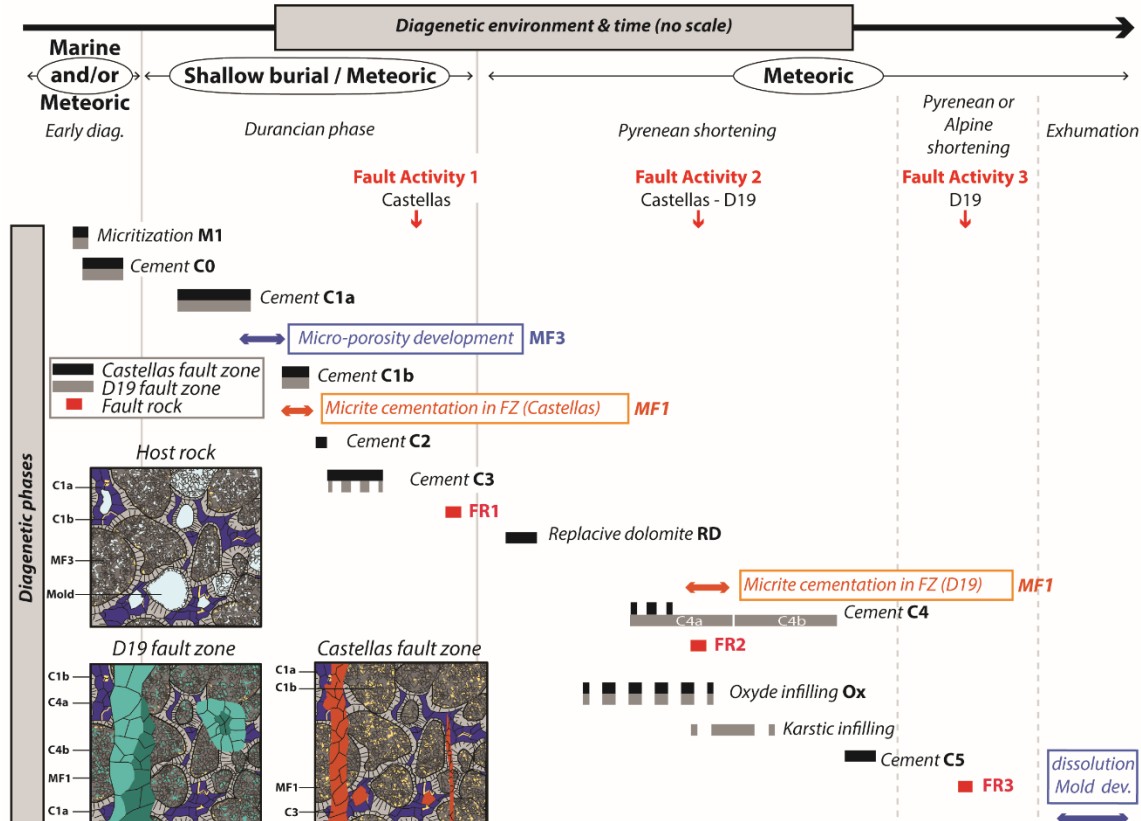

**313**

**314** *Figure 7: Paragenetic sequence of the both fault zones (black: Castellas, grey: D19) with micro-porosity development*
**315** *(blue),cementation (orange) and fault zone activation events (red).*

**316** fault core formation and are related to the fault nucleation. Replacive dolomite is found within
**317** FR1 matrix (Fig. 5E), therefore, it developed after FR1 formation. Finally, the C4 cement can
**318** be noticed within FR2 matrix indicating that C4 cementation event post-dated FR2 formation.
**319** The fault rock 2 (FR2) developed during strike-slip reactivation of the studied faults. The
**320** combined superposition, overlap, cross-cutting principles and isotopic signature of cements
**321** brought out the chronology between phases, and revealed the paragenetic sequence (Fig. 7).

**322** The Urgonian carbonates in La Fare anticline underwent 3 major diagenetic events, which
**323** impacted the host rock and/or the fault zones. We discriminate among diagenetic events that
**324** occurred before and during faulting.

**325**     **a. Pre fault diagenesis – microporosity development**
**326** During Upper Barremian, just after deposition, micro-bores organisms at the sediment-water
**327** interface enhanced the formation of micritic calcitic envelopes on bioclasts, ooids and peloids
**328** (Purser, 1980; Reid and Macintyre, 2000; Samankassou et al., 2005; Vincent et al., 2007). This
**329** micritisation in marine conditions is typical for Urgonian low-energy inner platform
**330** environment (Fournier et al., 2011; Masse, 1976). Subsequently, C0 cement formed around
**331** grains giving rise to a solid envelop inducing the preservation of the original grain shape during
**332** the later burial compaction (Step 0 on Fig. 8). However, the majority of isotopic values do not
**333** fit in the Barremian sea water calcite box which ranges from -1.00‰ to -4.00‰ for $\delta^{18}O$ and
**334** from +1.00‰ to +3.00‰ for $\delta^{13}C$ (Fouke et al., 1996; Godet et al., 2006). Only two data points
**335** pertaining to micritised grains show isotopic values close the Barremian sea water calcite. The
**336** isotopic depletion of other data indicates the slight impact of C0 cementation on isotopic values.

The next sub-phase of cementation C1a partly fills intergranular porosity. This non luminescent cement with isotopic values ranging from -6.8‰ to -3.9‰ for $\delta^{18}O$ and from -1.0‰ to +1.3‰ for $\delta^{13}C$ is characteristic of mixed fluids. Léonide et al. (2014) measured a calcite cement S1, near La Fare anticline with similar luminescence and isotopic range values (mean: $\delta^{18}O$= −5.49‰; $\delta^{13}C$=+2.34‰). These authors linked this cementation phase to a shallow burial meteoric fluid circulation under equatorial climate during Durancian uplift. This diagenetic event led to micrite re-crystallization, and to the development of microporosity (MF3). Since La Fare carbonates were exhumed at that time (Léonide et al., 2014) the meteoric fluids led to similar diagenetic modifications (Step 1 on Fig. 8):

(i)   Micrite re-crystallization and microporosity MF3 setup by Ostwald ripening processes (Fig. 9B1a; Ostwald, 1886; Volery et al., 2010).
(ii)  Cementation of C1a, partly filling intergranular porosity (Fig. 9B1b)

The micrite re-crystallization strongly increased rock porosity due to enhanced microporosity (Fig. 9B1a). Resulting from this event, Urgonian carbonates formed a type III reservoir *sensu* Nelson (2001).

**b.  Fault-related diagenesis – alteration of reservoir properties**

**Normal faulting-related diagenesis**

The Castellas fault first nucleated during Durancian uplift (Aubert et al., 2019b; Matonti et al., 2012) affecting the host Urgonian carbonates.
In porous granular media, fault nucleation mechanisms can lead to dilation processes (Fossen and Bale, 2007; Fossen and Rotevatn, 2016; Main et al., 2000; Wilkins et al., 2007; Zhu and Wong, 1997) under low-confining pressure (<100 KPa; Alikarami and Torabi 2015). Because this process leads to dilatancy, it increases the rock permeability (Alikarami and Torabi, 2015; Bernard et al., 2002) in the first stage of deformation bands (Heiland et al., 2001; Lothe et al., 2002) enhancing fluid flows.
Castellas fault zone nucleated within a partially and dimly cemented host rock under low-confining pressure, in an extensional stress regime, at a depth <1 km (Lamarche et al. 2012). Under these conditions, Barremian host rock were likely characterised by mechanical and petrographical properties close to porous granular media described above. Moreover, Micarelli et al. (2006) showed that, during early stages of deformation, fault zones in carbonates have a hydraulic behaviour comparable to deformation bands in carbonates. Hence, in the Urgonian carbonates of La Fare area, dilatant processes occurred as an incipient fault mechanism and enhanced fluid circulations along the deformation bands. Fluid flows led to the cementation of C1b (Step 2 on Fig. 8). However, dilation bands were likely unstable and grain collapse occured swiftly after the beginning of the deformation due to an increase in the loading stress (Lothe et al., 2002). This could be the explanation why C1b does not fill all intergranular porosity. Consequently, as all micritic grains in fault zone are cemented by C1b, the bulk isotopic measurements are strongly influenced by C1 cement isotopic values. This is the explanation why in transect 3 the bulk isotopic values 30 m apart from the fault (means of -5.26‰ for $\delta^{18}O$ and -0.82‰ for $\delta^{13}C$) are close to bulk isotopic values far from the fault plane (188 m; -5.37‰ for $\delta^{18}O$ and -0.65‰ for $\delta^{13}C$, Fig. 6A). Dilation bands have also been described by *Kaminskaite et al.* (2019) in the San Vito Lo Capo carbonates grainstones (Sicily, Italy). These dilation bands also led to selective cementation of the carbonate rocks and to a microporosity decrease.

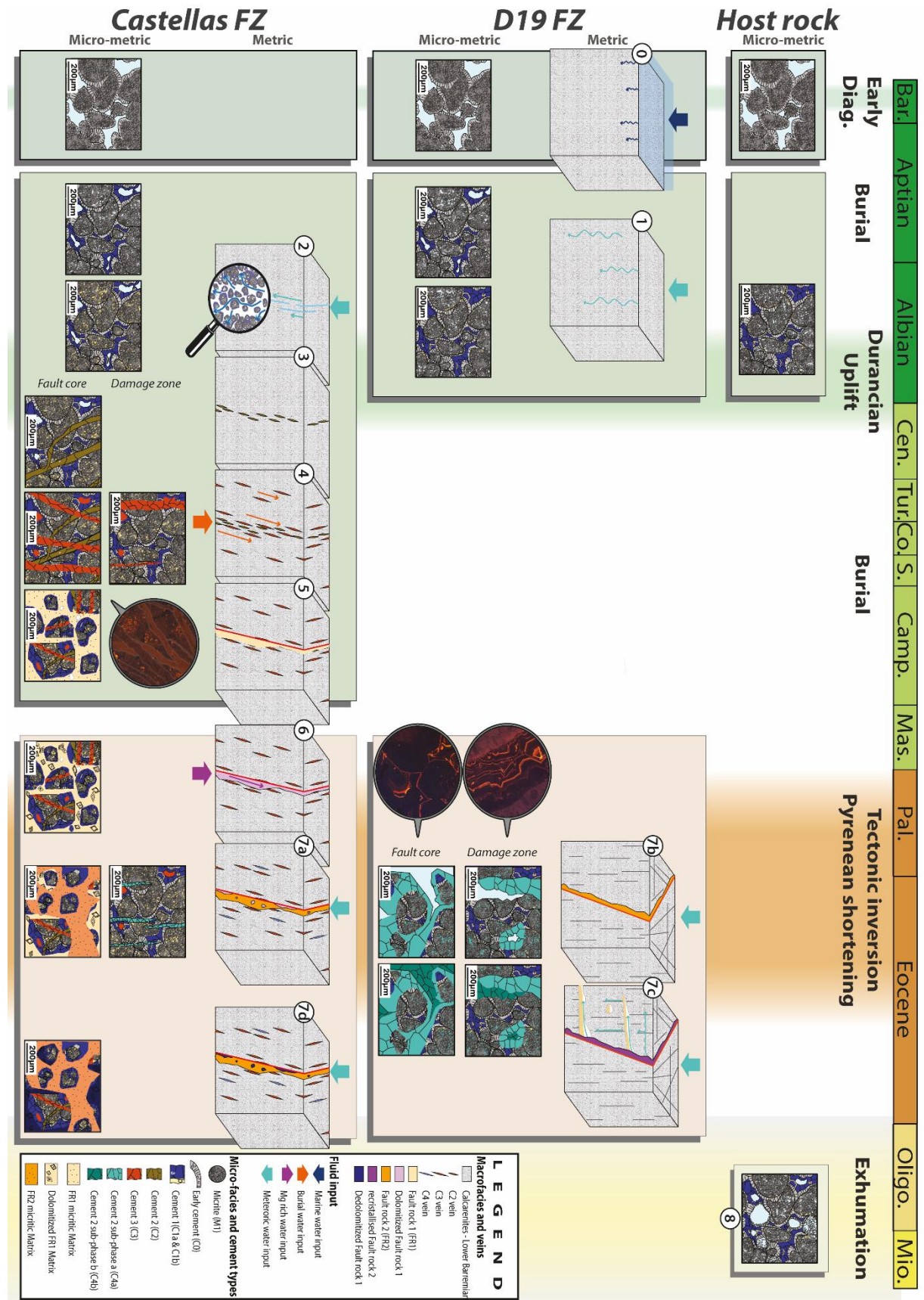

**381**

**Figure 8** : *Diagenetic and geodynamic evolution since the Barremian of both fault zones and host rock at the metric and micro-metric*
**382**
**383** *scale. Numbers 0 to 8 correspond to the steps 0 to 8 (see text for description).*

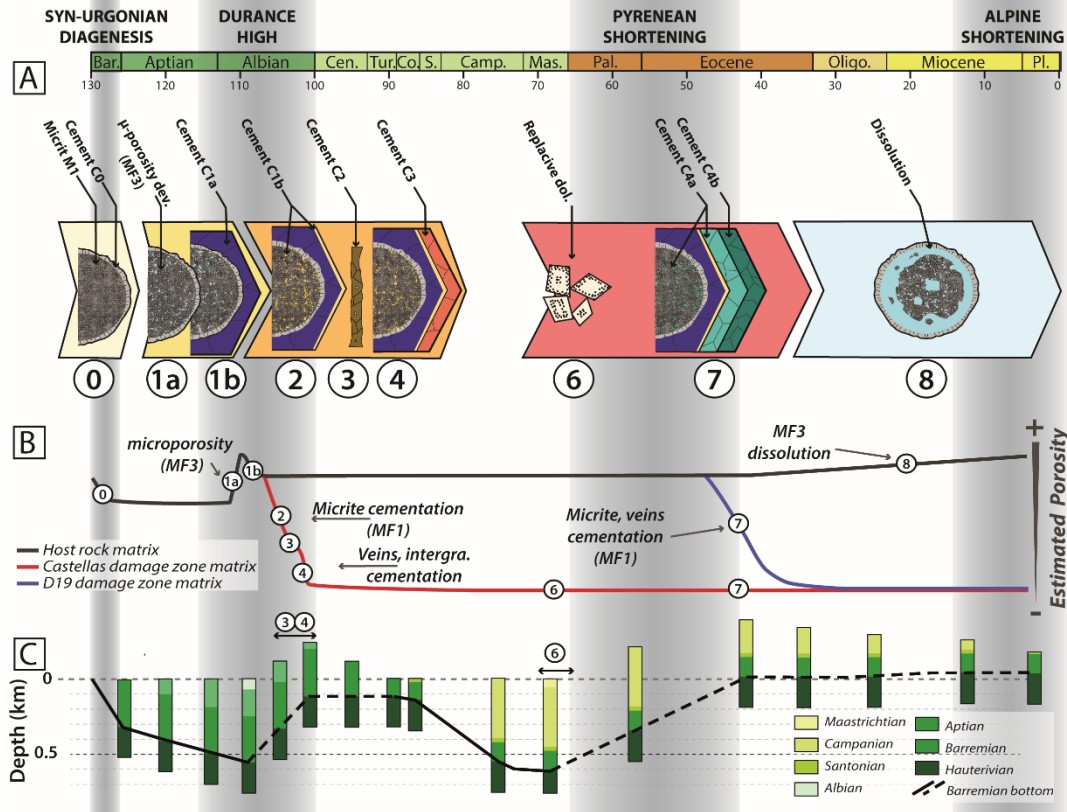

**Figure 9** : *Evolution of reservoir properties. A: different cementation phases;* numbers 0 to 8 correspond to the steps 0 to 8 (see text for description), *B: relative porosity evolution of the host rock and the two fault zones; C: Burial/Uplift curve of Barremian basement (modified from Matonti et al. (2012)).*

Cementation (C1a and C1b) conferred a stiffer response of limestone to deformation, making it prone to deform through brittle structures (joints and veins), rather than via granular particulate flow (deformation bands).. During the first stages of fault evolution in low-porosity limestones, intense fracturing of the fault zone predating fault core formation is known to increase fault permeability (Micarelli et al., 2006). In the studied faults, the first brittle event allowed Al-rich fluids to flow with fine-grained quartz grains in the incipient open fractures leading to precipitation of C2 cement (Step 3 on Fig. 8). The Urgonian facies of the studied area are composed of pure carbonates without siliciclastic input. Quartz grains and Aluminium could have been reworked from surrounding formations. The rocks underlying the studied exposed Urgonian carbonates are limestones and dolostones. Albian and Aptian rocks are marly and sandy limestones, respectively (Anglada et al., 1977). Hence, Aptian layers are very likely to be the source of quartz. The fluids may have carried small grains of quartz from the Aptian sandy limestones via the fracture network. The Al enrichment of C2 could result from the erosion of Albian and Aptian deposits during the Durancian uplift (Guendon and Parron, 1985; Triat, 1982).

As the fault zone grew, new fracture sets formed, leading to new phase of calcite cementation (C3) in veins and intergranular porosity (Step 4 on Fig. 8). The $\delta^{18}O$ isotopic values of C3 range from -10.40‰ to -6.73‰ with $\delta^{13}C$ values between -2.09‰ and +1.22‰. As C3 cementation occurred during the Durancian uplift and denudation, C3 most probably did not cement in deep burial conditions (maximum depth of 500 m; Fig. 9C4).The negative $\delta^{13}C$ values tend corroborate the hypothesis of cementation induced by meteoric fluid rather than marine ones. Hence, C3 would correspond to a shallow burial/meteoric cementation phase. Due to this

cementation, rocks in this zone tightened with porosity down to <5%. The porosity did not change since this event (Fig. 9B5). This porosity reduction due to cementation has also been observed in other cases of brittle-dilatant faults (Agosta et al., 2007; Celico et al., 2006; Gaviglio et al., 2009; Mozley and Goodwin, 1995). Following this, the fault zone was a barrier to fluid flow, leading to a reservoir compartmentalization. Fluids responsible for precipitation of C3 cement also occurred along fracture clusters of the D19 sector and led to vein formation.

In a later stage, the fault core formed and the fault plane *sensu-stricto* developed, leading to FR1 breccia with a permeable matrix with quartz grains >100 µm in size (Step 5 on Fig. 8). These grains either came from silica found inside C2 cement described above or from Aptian overlying rocks. Silica crystals in C2 veins are scarce and smaller than 10 µm. Thus, quartz grains may rather come from Aptian rocks like the ones found in C2 veins. The presence of Aptian quartz in the fault core proves that the Castellas fault affected also Aptian rocks, which were later eroded during the Durancian uplift. According to this, the fault activity occurred before total erosion of Aptian rocks. Un-cemented breccias within the fault core formed good fluid pathways (Billi et al., 2008; Delle Piane et al., 2016). In the studied fault, formation of FR1 breccia allowed the fault core to act as a drain. However, the cemented surrounding host rocks constrained the lateral extent of the drainage area of this high-permeable conduit. Un-cemented breccias acting as good across- and along- fluid pathways were also described on Apennines carbonate formations within fault cores of strike and extensional faults (Billi et al., 2003, 2008; Storti et al., 2003).

**Tectonic Inversion – Castellas fault-related dolomitization**

At the onset of the Pyrenean shortening, compressive stresses led to underground water upwelling through the permeable fault core. This fluid flow triggered the dolomitization of FR1 matrix (Step 6 on Fig. 8). This matrix-selective dolomitization could have been favoured by several factors:

(i)     The matrix has higher permeability than cemented clasts with a smaller grain size, hence a higher grain surface area (Machel, 2004);

(ii)    This type of upwelling fluids, so-called "squeegee-type", are short lived processes (Buschkuehle and Machel, 2002; Deming et al., 1990; Dorobek, 1989; Machel et al., 2000) not favourable for massive dolomitization;

(iii)   Low-temperature fluids, under 50°-80°C, enabled the preservation of FR1 clast initial structure. Contrarily, high-temperature dolomitization tends to be destructive (Machel, 2004);

(iv)    The tight surrounding host rock constrained Mg-rich fluid circulation to the fault core domain.

Gisquet et al. (2013) noticed similar fault related replacive dolomitization phase in the Etoile massif, 23 km South-Est of the studied zones. They linked the dolomitization to contractional stress regime during the early (Late Cretaceous) Pyrenean shortening. According to these authors, the tectonic stress led to low-temperature upwelling fluids likely Mg-enriched by the dissolution of underlying Jurassic dolomites. The Jurassic dolomites also occur in La Fare anticline. Since the fluids leading to dolomitization of fault core were low-temperature and since dolomites occur underground, it is possible that the dolomitization in La Fare and in the Etoile massif was similar and synchronous. Matrix dolomitization can increase inter-crystalline

and/or inter-particle porosity up to 13% but the later dolomite overgrowth reduces the porosity
and permeability (Lucia, 2004; Machel, 2004; Saller and Henderson, 2001). Hence, in the first
stages of dolomitization, the fault core was an important drain. After the growth of dolomite
crystals, the fault core turned into barrier (Fig. 9 B6 and C6)

**458   Sinistral tectonic inversion – meteoric alteration of reservoir properties**

The ongoing tectonic inversion with increasing compressive stresses eventually led to the
Castellas fault sinistral reactivation and to the onset of D19 fault zone (Aubert et al., 2019b).
Aubert et al. (2019a) has shown that this compression reactivated the pre-existing early N030°
background fractures (Step 7 on Fig. 8). This tectonic event formed FR2 in fault cores but with
specific diagenetic consequences. In the D19 fault zone, the fault nucleation and reactivation of
background fractures led to pluri-metric to kilometric fault surfaces with a permeable fault rock
acting as drains and localizing the fluid flow (Aubert et al., 2019a). This fluid flow witnessed
by the cementation of C4a and C4b in veins and micritised grains (MF1, Step 7c on Fig. 8),
leading to a strong porosity decrease in the fault zone (Fig. 9,B7 and C7). However, not all
fractures were cemented by C4, so that fracture porosity/permeability was still partially
preserved. Therefore, the D19 fault zone became a type I reservoir *sensu* Nelson (2001) with a
very low matrix porosity/permeability and high fracture-related secondary permeability (Aubert
et al., 2019a).
Along F2, successive fluids gave rise to karsts, karstic infilling and dissolution/cementation
processes of FR2 matrix (Step 7c on Fig. 8). Then, FR2 was sealed by C4 cementation. Isotopic
values of C4 cement (from -9.2 to -6.1‰ for $\delta^{18}O$ and from -5.01‰ to -1.0‰ for $\delta^{13}C$) highlight
the strong influence of meteoric fluids. This is coherent with the occurrence of karstic infilling
due to fluid circulations in vadose zone, with alternating dissolution and cementation (Swart,
2015). However, the positive covariance between $\delta^{18}O$ and $\delta^{13}C$ of C4 suggests mixed fluids
(Allan and Matthews, 1982) of meteoric water and burial or marine water.
In the Castellas fault zone, the host rocks are slightly impacted by these meteoric fluid
circulations. Yet, some veins filled with C4a cement occur along transect 2 and transect 3 (Step
7a on Fig. 8). Two samples have enriched $\delta^{18}O$ and $\delta^{13}C$ isotopic values (respective means of -
6.25‰ and -4.20‰ for $\delta^{18}O$; -0.64 and -0.09‰ for $\delta^{13}C$) similar to C1 cement (Fig. 6A). This
indicates that C4 cement in the Castellas fault zone was precocious in comparison to the D19.
C4 cement in Castellas area is restricted to transect 2. Transect 2 cross-cuts the Castellas fault
along a relay zone (Fig. 2A). Relay or linkage zones occur where two fault segments overlap
each other during fault grow (Kim et al., 2004; Long and Imber, 2011; Walsh et al., 1999, 2003).
Consequently, the fault complexity, the fracture intensity and the fracture-strike range are
increased (Kim et al., 2004; Sibson, 1996). This process in the studied area resulted in a well-
connected fracture network that increased the permeability and favoured local fluid circulations.
In transect 2, the increase of the local permeability in the relay zone enhanced fluid flow related
to C4 cement. The relay zones along the Castellas fault and their consequences on the fracture
permeability are, therefore, responsible for this local cementation event. On the contrary,
cementation in D19 fault zone is linked to the highly permeable fault surfaces which acted as
drains (Aubert et al., 2019a). This implies that the cementation occurred only after the
development of the fault surface. In the case of Castellas, the relay zone was already present,
inherited from the former extensional activity, allowing early C4 fluid to flow through the fault
zone. This, in addition, explains why the early C4 cementation has not been recorded in D19
fault zone. The C4 cementation in transect 2 reduced the porosity to less than 8% on a larger
zone (>60 m) than in both others transects (transect 1 ≈30m, transect 3>40m).
The reactivation of the Castellas fault formed a new fracture network that locally triggerred the
fracture connectivity and permeability. The Castellas fault zone formed a type I reservoir
(Nelson, 2001), but lateral variation of the fracture network implies lateral variations of the
hydraulic properties. Thus, the fault zone was both a drain and a barrier (Matonti et al., 2012).
In this case, the most appropriate concept would be a sieve, because in this analogy, it is
synchronously closed in places and open in other places.
After these events, the matrix of the Castellas fault core was de-dolomitized (FR1) in relation
to cementation C5 (Step 7d on Fig. 8). The C5 cement isotope values (mean of -7.49‰ for $\delta^{18}$O
and -4.01‰ for $\delta^{13}$C) are comprised within C4 positive covariance between $\delta^{18}$O and $\delta^{13}$C. This
indicates a continuity between C4 and C5 fluid flows. The measurements with the SEM
revealed a lack of Mg in the matrix indicating that C5 totally recrystallised the replacive
dolomite. Following this de-dolomitization phase, no additional diagenetic event is recorded in
Castellas fault zone.
A late Pyrenean to Alpine compression reactivated the D19 fault zone what formed the new
fault rock FR3. The matrix of this fault rock has very low $\delta^{13}$C isotopic values (mean of -6.83‰)
indicating an organic matter input (Swart, 2015). This implies fluids percolating soils, as results
from a near surface fluid circulation. We deduce that the D19 faults was lately reactivated after
the folding of the La Fare anticline. There is no such cementation with similar isotope values
in the fault zone, meaning that fluids and cements did not alter the fault zone diagenetic
properties.
Eventually, the late exhumation of the Urgonian carbonate host rocks led to flows inducing
dissolution of MF3 grains in the host rock. This phase produced the moldic porosity and
increased the porosity/permeability (Step 8 on Fig. 9B and C). These fluids, however, did not
affect fault zones.
**2. EVOLUTION OF FAULT ZONES RESERVOIR PROPERTIES**
The host rock presents a monophasic evolution and switch from a type IV reservoir where
matrix provided storage and flow, to a type III reservoir where fractures behave as pathways
towards fluid flow but the production comes mainly from the matrix (*Nelson* 2001, Fig. 10A).
The fault zones present a more complex polyphasic evolution than the host rock. Indeed, their
reservoir properties evolved from a type IV reservoir corresponding to the host rock to a type I
reservoir where fractures provide both storage and flow pathways (*Nelson* 2001, Fig. 10A).
Both fault zones present slight differences. The Castellas fault zone was completely tight soon
after C3 cementation. Consequently, it did not fit to the Nelson reservoir type classification.
However, after fault core formation, the fault zone presents a high fault core permeability. In
this study we propose a new approach with a triangle diagram taking into account fault core
permeability to remove the flaws of this method (Fig. 10B). The percentage assigned to the
fault core or to the matrix are qualitatively estimated. Further quantification could be evaluated,
for instance, with the width of the fault core and damage zone domains, or by estimating the
fracture network volume. However, no recent study have provided such quantification. Thus,
for Castellas fault zone, permeability evolves from a stage with exclusive contribution from the
host rock permeability (100% matrix; step 0 on Fig. 10B) to a permeability due to 50% to the
matrix and 50% to the fault core during dilation band development (step 2 on Fig. 10B).
Thereafter, during the two

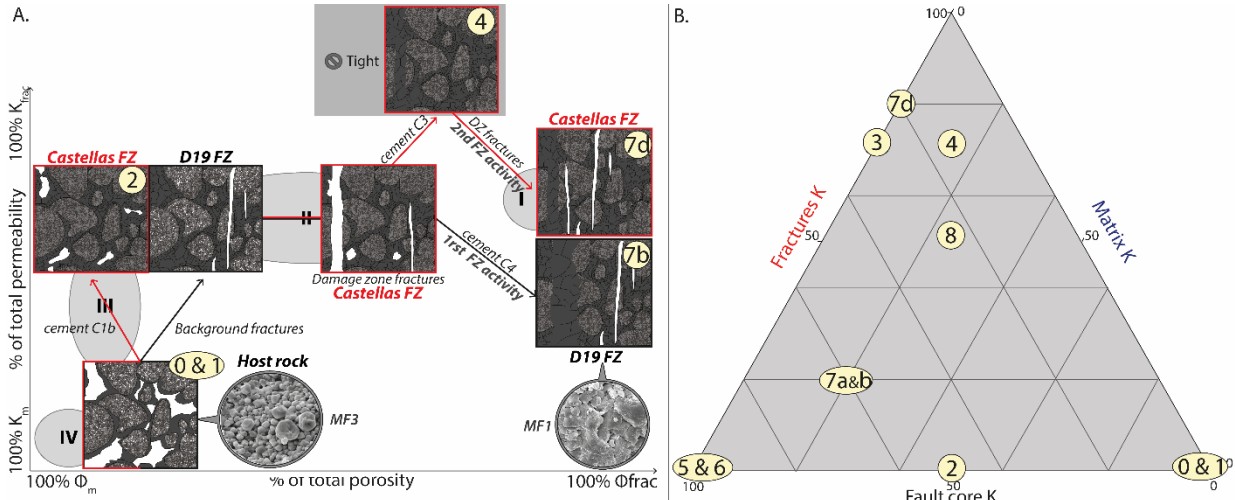


***Figure 10*** *: Castellas and D19 fault zone reservoir properties evolution. A: evolution of permeability and porosity taking into*
*account fault zone fractures and matrix after Nelson (2001) and B: Triangle diagram of permeability evolution with 3*
*components: matrix, fractures and fault core. Numbers 1 to 8 correspond to the steps 1 to 8(see text for description). K:*
*Permeability, Φ: porosity, FZ: Fault Zone, DZ: Damage zone, MF1 and MF3: Micrite micro-fabric.*
fracture events permeability is mainly linked to fracturing (C2: 30% fault core, 70% fractures;
C3: 15% fault core, 15% matrix, 70% fractures; step 3, 4 on Fig. 10B). Then, after fault core
formation and during dolomitization event, permeability is solely provided by the fault core
(step 6, 7 on Fig. 10B). Lastly, after fault zone reactivation, the permeability is due to 20% to
the fault core and 80% to fractures (step 7c on Fig. 10B). The D19 fault zone permeability
during its development was related for 20% to the matrix, 20% to the fractures and 60% to the
fault core (step 7a and 7b on Fig. 10B).

## 555 8. CONCLUSION

This study deciphered the diagenetic evolution of two fault zones and the impact on reservoir
properties of both faults and host rock in the frame of the overall geodynamic context of the SE
Basin. The main outcomes are:
• Fault zones may have a complex diagenetic history, but most diagenetic phases occur
during the nucleation of the fault. In the case of Castellas fault zone, the diagenetic
imprint is mainly influenced by early diagenesis occurring along fractures and diffuse
dilation zones prior to the proper fault plane nucleation. Regarding D19 fault zone, most
of diagenetic alterations occurred just after fault onset in the first stage of its activity. In
both cases, the cementation altered initial reservoir properties in the fault zone vicinity,
switching from type III to type I during the first stages of fault development. Later fault
reactivation slightly impacts matrix porosity/permeability.
• Fault zones act as drains canalizing fluid flows in the beginning of their development.
This induces fault zone cementation but preservation of host rock microporosity. This
important fluid drainage is visible on D19 outcrop where the flowing fluids led to
dissolution/cementation of fault rock matrix and formed karsts.

- All diagenetic stages, including cementation and dolomitization, result from low-temperature fluids with important meteoric water input. These low-temperature fluid flows associated with the deformation and cementation types and, the lack of mineralisation specific to high-temperature fluids disprove any significant hydrothermal influence.

This regional study allows to draw broader rules for complex faults with polyphasic activity affecting granular carbonates at shallow burial conditions (Fig. 9).

- Under extensional context, fault nucleation can lead to the development of dilation bands acting as conduits for fluid flow. Carbonates are very sensitive to rock-fluids interactions. Thus, the onset of dilation bands triggers important diagenetic reactions that strongly alter local reservoir properties. During later fault zone development, the diagenesis depends on faults zones internal architecture.
- Fracture networks related to fault nucleation in granular carbonates form good fluid pathways before proper fault plane formation. However, in the case of pre-fractured carbonates, like D19 fault zone, fault rocks early appear in fault cores. In these cases, fluids flowed preferentially within the permeable breccia rather than the damage zone.

**Acknowledgement**

We would like to thank Suzan Verdegaal, Lionel Marié and Alain Tonetto for support they provide during this study. We grateful to Editor Kei Ogata, and Fabrizio Agosta, Mattia Pizzati and Eric Salomon who made critical suggestions to improve this paper.

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
