# Peer review of "Diagenetic evolution of fault zones in Urgonian microporous carbonates, impact on reservoir properties (Provence – SE France)."

_Solid Earth, 2019_

## Referee Comment (RC1) · Anonymous Referee #1 · 13 Nov 2019

The present study presents original structural, mineralogical, geochemical, and petro-physical data on the control exerted by faults on reservoir properties in microporous carbonates. It is a research article dealing with the structural diagenesis of two high-angle fault zones exposed in a surface analogue of SE France. The studied faults are characterized by a heterogeneous architecture, which includes fractured and frag-mented host rocks , cataclastic fault rocks, and main slip surfaces, and underwent to three distinct deformation stages. The authors were able to assess the main dia-genetic processes associated to each of the aforementioned stages by performing a multi-disciplinary analysis. They tackled a very difficult problem: the unravel of the role played by single deformation and diagenetic processes on the fluid flow properties of

fractured carbonate reservoirs. The authors were able to document how the studied faults first behave as localized fluid conduits for low temperature fluids, and then as localized fluid barriers after two separate stages of calcite precipitation The manuscript is well written (although some modifications can be made throughout the text, see the Specific Comments below), both aims and results of the work are clearly reported, the methods robust and convincing, and data interpretation very convincing. A slight improvement of the manuscript can be made by considering the following three points: 1. Authors should better illustrate the crosscutting relations among the different structural elements (dilational bands, open fractures, shear fractures, etc.) measured both within and outside the study fault zones. 2. Rename section IV as "Discussion", and shorten the single chapters it includes (mainly, the chapter on the fault-related diagenesis). 3. Expand data discussion on the impact of fault zones on reservoir properties by adding references to other surface analogues worldwide. In conclusion, the work done by the authors is intriguing, the topic interesting, the employed methods appropriated, and data interpretation convincing. For this reason, based upon the aforementioned comments, and taking into account the overall quality of the paper, I recommend to accept with minor revisions the submitted manuscript. Specific Comments are reported below.

Specific Comments Abstract: please check for wrong punctuation marks, grammar, and syntax. Introduction: please remove the final sentence. Geological context: check for grammar; remove lines 110-124 (out of place). Methods: please double check the standards used for stable isotope analyses. Results: please check both grammar and syntax. Diagenetic Evolution: please shorten the whole section, and rename it (cf. comment above); re-consider dilation as an incipient faulting mechanisms; please separate data interpretation for granular media from that inferred for a cohesive rock; please state all assumptions required for temperature calculation based upon oxygen isotope data and well-known fractionation curves.

[Figure]

---

## Referee Comment (RC2) · Eric Salomon (Referee) · 29 Nov 2019

This manuscript discusses the evolution of two fault zones displacing Cretaceous carbonate deposits. One of the fault cores is already impressively visible on aerial view, emphasizing this area as a promising site for studying fault-fluid relationships. Based on field and microstructural observations, the authors describe a series of fracture and cement generations and aim at establishing a relation between these stages.

Unfortunately, as I will outline below, the manuscript contains a number of analytical and technical issues making it currently not suitable for publication. Reluctantly, I must therefore recommend rejection with the option for resubmission for full review when the

manuscript is in an improved shape.

——

Maybe, at first, some general remarks:

- The English language and grammar needs to be improved. The use of the article "the" as well as singular and plural is often misplaced. I understand, such things may be difficult and maddening for non-native speakers (I am non-native myself), but at least consistency can be paid attention to. As an example: "fluids-rock interactions" (line 8) vs. "fluid-rock interactions" (line 11) vs. "fluids-rock interaction" (line 37). Also the use of lower- and upper-case needs to be consistent ("early Aptian", "Early Aptian", "early cretaceous", "Late-Cretaceous", "Late cretaceous", "La Fare", "La fare"...). Such things are very annoying and only distract the reader from the scientific content.

- Also typos are frequent, e.g. "height cement stages" (lines 17 and 206) instead of "eight cement stages", or "d18C" (header of table 2) instead of "d18O".

- Throughout the manuscript, the structure needs to be improved. Repetitions of content frequently occur. One example in lines 113-124 "The structure of both polyphase fault zones results from three tectonic events: [list of events]. These tectonic events impacted the fault zone and fault core structure."

- Consistency is needed in decimal places. Occasionally two or one decimal places are given for the same type of data.

- Consistency is needed on writing out numbers (e.g., "2" vs "two", "62" vs "A hundred and eighty-nine" (line 250)).

——

Introduction:

- The introduction is quite chaotic and difficult to follow. It repeatedly jumps from describing lithology/stratigraphy to describing fault zones. Moreover, though the sentences are not identical, their content is often repetitive.

- The authors should reconsider if they want their study area only to be understood as an outcrop analogue to Middle East carbonate reservoirs.

——

Geological context / Data Base

- Same as in the introduction, there is a mixing and jumping of the description of stratigraphy and structure. This needs to be clearly separated.

- Figure 1 caption starts with "Geological context of the study area", but only a geographical outline of France is shown. Also coordinates are missing, and it is essentially impossible for someone not familiar with the study area to readily locate it. Marking the study area in the map inset with a rectangle much larger than the area of interest and only providing "near Marseille (Fig. 1A)" (line 76) is not sufficient.

- In the chapter "Data Base" it is not clear if its content is derived from the authors analysis or from existing literature. If it is the former, it should be moved to the results chapter.

- The data base deals with faults, sub-faults, sets, transects, units, etc.. This easily gets very complicated and therefore it is paramount that the paragraphs are well structured to guide the reader through this complexity. Unfortunately, at the moment, this is not the case. As an example: Line 106 "The set one, constituted of F3 and F4, is…". The problem here is that F3 and F4 have not been defined before. Hence, the reader does not understand this abbreviation and is left in confusion.

——

Results:

- In Figure 3c, pore types are shown. First, the resolution of the photos must be improved. And second, and this is now maybe more a matter of perspective, but it seems

that the two blue pores are just the result of grains falling out of the sample during thin section preparation. Either way, from the picture shown, it is very difficult to reconcile that the host rock has a porosity larger 10 %.

- Subchapter “Carbonate and Oxygene Isotopes”: This is again about consistency, and please excuse for being picky: 189 measurements have been made on 16 samples and 32 thin sections (Line 250; set aside the confusion of what the difference between thin section and sample is), and these distribute on 49 bulk rock, 48 vein, 40 fault rock, and 26 intergranular space measurements. The latter list however adds up to “only” 163 measurements. And table 2 shows even 204 measurements. How do these numbers fit?

——

The discussion and conclusion chapters are at present difficult to evaluate, which partly roots in the circumstance that the previous data presentation is difficult to follow. Therefore, below, I will only exemplarily address to three sections of these chapters. Meanwhile, a major concern is about the interpretation of the d18O data: Large parts of the discussion deal about the origin of fluids, i.e. marine or meteoric, and it appears that the measured d18O carbonate data is directly used as a d18O fluid signature. Without an estimation of the carbonate formation temperature, this is not a correct approach.

—

Chapter IV, 2: Fault related diagenesis

- The start of this chapter deals about potential dilation and I am afraid this discussion is on a weak basis. The authors themselves mention that “[dilation processes under low confining pressure] is only possible in highly porous granular media.” (line 344). It is my impression that this attribute does not apply to the here analyzed rock, which is described as cemented and only comprising “>10 % porosity but located in the grains” (line 336), i.e. secondary porosity due to partially dissolved grains. As a comparison,

Alikarami & Torabi (2015), to which the authors often refer in this regard, deal with quartz sand with porosities of 33-45 % of primary origin. This is a significant difference.

- In line 358 it is claimed: “In the Urgonian carbonates of La Fare sector, dilatant processes enhanced fluid circulation in the rock along the deformation bands and led to the cementation of C1b”. Unfortunately, deformation bands in the study area have neither been mentioned nor described before and thereafter in the manuscript.

—

Line 381-400 deal with formation temperatures of cement generation C3:

- For the calculation, “the formula of Ali (1995)” is used.

1. Ali (1995) presents more than one formula.

2. Ali (1995) is not the original reference! It is Epstein et al. (1953) and Craig (1965) that needs to be cited.

3. As far as I know (I might be mistaken), the equation of Epstein et al. (1953) is based on biogenic calcite. The authors might want to check the equation of Kim & O’Neil (1997, Geochimica et Cosmochimica Acta) for inorganic calcite. Though in the end it might as well not make a significant difference for the authors calculation.

- “temperature of initial fluids: 33°C to 34°C (Littler et al., 2011)”

1. What is meant with “initial fluids”?

2. Littler et al. (2011) present own data on paleo-sea-surface temperatures, which they set in comparison with existing data. The temperature range extends beyond 33-34°C.

3. If original data of Littler et al. (2011) is used, this means that data of the Hauterivian (133-129 Ma) is used. The authors however give an age estimate for C3 as ∼Cenomanian (101-94 Ma). This may be a significant age difference and suitability of the Hauterivian temperatures for the calculation needs at least be discussed.

4. If the data quoted by Littler et al (2011) is used, then the original work needs to be cited.

- “meteoric water: -4.0 d18O (Robinson et al., 2002)”

1. This is the same as with Littler et al. (2011). Robinson et al. (2002) present data from the Barremian, whose suitable application needs to be discussed.

2. The -4.0 value is an average of Robinson et al. (2002) data.

3. Most important: the -4.0 d18O of Robinson et al. (2002) refers to d18O of carbonate and not to d18O of the fluid from which the carbonate precipitated.

- Line 393: “We calculated a C3 fluid temperature 40°C and 60°C.“

1. As the authors do not guide through the calculation and how the parameters have been applied, it is impossible to follow how these values have been determined.

- For the calculation of the formation depth, a geothermal gradient of 26.4°C/km (Ali, 1995) is used. Such precision is quite ambitious.

- Line 395: “The negative d13C values tend to indicate that it would rather be a meteoric fluid than a marine fluid.”

1. The presented d13C data range from -2.09 to +1.22. Did the authors rather mean d18O?

2. Why does it indicate rather a meteoric fluid? This needs to be discussed. The d18O (VPDB) carbonate value does not per se indicate the type of fluid from which the carbonate precipitated. This depends on the formation temperature.

- Line 399: “As C3 cementation occurred during the Durancian uplift and denudation, C3 most probably did not cemented at high depth (Fig. 9C4). More probably, C3 fluids were meteoric burial fluid which were upwelled under tectonic stresses”.

1. What are the arguments for the formation age of C3? This has not been discussed

before. It is simply claimed here that it formed during the Durancian uplift. Why can it not be related to e.g. the Pyrenean shortening?

2. Whether true or not, the authors need to better explain in more detail why fluid upwelling is the likely process. From the current information given, it is difficult to follow the line of reasoning.

—

In the conclusions, line 547-550: "All diagenetic stages [. . .] result from low temperature flows with important meteoric water input. This low temperature disproves any hydrothermal influence. Therefore, both fault zones were not linked to high depth basement faults."

- Set aside the uncertainty in determining formation temperature and source of fluid, absence of hydrothermal fluids does not permit conclusions on the deep structure of a fault. The fault may very well be connected to a basement fault, but the fracture connectivity may just be poor.

——

I apologize for being so fussy here, but these are just too many flaws that they cannot be neglected. When the authors improve the manuscript, I urge them to not only restrict the revision to the passages I pointed out, but to work through the other parts of the manuscript as well, which contain similar issues. Please also excuse my direct language, and I hope not to discourage the authors to work on this study. The study site remains a very interesting outcrop analogue from which unique information can be drawn.

–

Eric Salomon

---

## Author Comment (AC1) · 9 Jan 2020

Dear referee, I am pleased to send you the revised version of our paper on "Diagenetic evolution of fault zones in Urgonian microporous carbonates, impact on reservoir properties (Provence – SE France). You will find enclosed in the supplement, the comments and corrections to your remarks. They are listed together with the actions made: - Comments are in italics - Corrections validated are emphasized in green - Corrections with a red bold part are considered un-useful or inappropriate You will see that most of corrections have been respected as you requested. Best regards, Irène Aubert

I. General remarks 1. Authors should better illustrate the crosscutting relations among

the different structural elements (dilational bands, open fractures, shear fractures, etc.) measured both within and outside the study fault zones. - Chronological relation between cementation events has been added. Lines 266 to 275

"2. Rename section IV as "Discussion", and shorten the single chapters it includes (mainly, the chapter on the fault-related diagenesis)". - Done

3. Expand data discussion on the impact of fault zones on reservoir properties by adding references to other surface analogues worldwide. - Done. we added references lines 338-340, 362-364 and 378-380

II. Specific Comments Abstract: "please check for wrong punctuation marks, grammar, and syntax." - done

Introduction: "please remove the final sentence." - done

Geological context: "check for grammar; remove lines 110-124 (out of place)." - done

Methods: "please double check the standards used for stable isotope analyses." - done

Results: "please check both grammar and syntax." - done

Diagenetic Evolution: "please shorten the whole section, and rename it (cf. comment above);" - done – section was shorten by removing the part on C3 temperature calculation

"re-consider dilation as an incipient faulting mechanisms;" - done – we added this sentence : "Hence, in the Urgonian carbonates of La Fare sector, dilatant processes occurred as an incipient fault mechanism and enhanced fluid circulations along the deformation bands." (line 327-328)

"please separate data interpretation for granular media from that inferred for a cohesive rock;" - Modifications effected. We have made a more marked separation between description of dilation bands within highly porous rock and Castellas host rock

"please state all assumptions required for temperature calculation based upon oxygen isotope data and well-known fractionation curves." - We removed this part. We decided to remove this part from the manuscript for 2 reasons: (1) to shorten the fault related diagenetic part and, (2) $\delta18O$ during Durancian uplift (Aptian/Albian) was difficult to estimate. Moreover, the association of burial/uplift curve and $\delta13C$ values allow an interpretation of the fluid origin. "As C3 cementation occurred during the Durancian uplift and denudation, C3 most probably did not cemented at high depth (depth of maximum 500m; Fig. III. 9C4).The negative $\delta13C$ values tend corroborate that it would rather be a meteoric fluid than a marine fluid." (lines 353-356)

Please also note the supplement to this comment:
https://www.solid-earth-discuss.net/se-2019-153/se-2019-153-AC1-supplement.pdf

[Figure]

[Figure]

**Fig. 1.**

[Figure]

CASTELLAS FAULT ZONE

**Fig. 2.**

[Figure]

**A** *Castellas Fault zone*

*Transect 2*

*Transect 1*

*Transect 3*

150 m

S     **a**     N

*Transect 1*

*Transect 3*

*Porous Rock-type*

*Transect 2*

*Tight rock-type*

Porosity (%): 16, 12, 8, 4, 0

Distance to the fault (m): 160, 120, 80, 40, 0, 20, 40, 60

*Distance to the fault (m)*

**B** *D19 Fault zone*

NNW

F 5

F4

F3

SSE

F2

F1

20 m

**C**

*a. Porous facies*

500μm

*b. Tight facies*

200μm

*c: Tight Rock-type*
*Barren stylolites*

200μm

*Transect 4*

*Porous Rock-type*

*Tight Rock-type*

Porosity (%): 16, 12, 8, 4, 0

**Fig. 3.**

A
Acc.V Spot Magn    Det  WD
20.0 kV 3.0   3500x    SE  12.5
20 µm

B
Acc.V Spot Magn    Det  WD
20.0 kV 3.0   3500x    SE  9.2
20 µm

C
Acc.V Spot Magn    Det  WD
20.0 kV 5.0   3500x    SE  16.0
20 µm

D
Acc.V Spot Magn    Det  WD
20.0 kV 3.0   3500x    SE  12.6
20 µm

E
Acc.V Spot Magn    Det  WD
20.0 kV 3.0   153x    SE  12.2
400 µm

F
Inter. sparite
clast
Acc.V Spot Magn    Det  WD
21.0 kV 4.8   1500x    SE  13.1
40 µm

**Fig. 4.**

[Figure]

**Fig. 5.**

Fig. 6.

[Figure]

Fig. 7.

[Figure]

**Fig. 8.**

[Figure]

[Figure]

**Fig. 9.**

[Figure]

**Fig. 10.**

---

## Author Comment (AC2) · 9 Jan 2020

Dear referee, I am pleased to send you the revised version of our paper on "Diagenetic evolution of fault zones in Urgonian microporous carbonates, impact on reservoir properties (Provence – SE France). You will find enclosed in the supplement, the comments and corrections to your remarks. They are listed together with the actions made: - Comments are in italics - Corrections validated are emphasized in green - Corrections with a red bold part are considered un-useful or inappropriate You will see that most of corrections have been respected as you requested. Best regards, Irène Aubert

I. General remarks by the referee #2 - The English language and grammar needs

to be improved. The use of the article "the" as well as singular and plural is often misplaced. I understand, such things may be difficult and maddening for non-native speakers (I am non-native myself), but at least consistency can be paid attention to. As an example: "fluids-rock interactions" (line 8) vs. "fluid-rock interactions" (line 11) vs. "fluids-rock interaction" (line 37). Also the use of lower- and upper-case needs to be consistent ("early Aptian", "Early Aptian", "early cretaceous", "Late-Cretaceous", "Late cretaceous", "La Fare", "La fare": : :). Such things are very annoying and only distract the reader from the scientific content. - Done

- Also typos are frequent, e.g. "height cement stages" (lines 17 and 206) instead of "eight cement stages", or "d18C" (header of table 2) instead of "d18O". - Done

- Throughout the manuscript, the structure needs to be improved. Repetitions of content frequently occur. One example in lines 113-124 "The structure of both polyphaser fault zones results from three tectonic events: [list of events]. These tectonic events impacted the fault zone and fault core structure." - Done. The structure of this part has been modified

- Consistency is needed in decimal places. Occasionally two or one decimal places are given for the same type of data. - Done

- Consistency is needed on writing out numbers (e.g., "2" vs "two", "62" vs "A hundred and eighty-nine" (line 250)). - Done We decided to write the numbers at the beginning of sentences in letter and in the rest of the sentence in digit. Introduction: - The introduction is quite chaotic and difficult to follow. It repeatedly jumps from describing lithology/stratigraphy to describing fault zones. Moreover, though the sentences are not identical, their content is often repetitive. Corrections have been realized after referee 1 comments

- The authors should reconsider if they want their study area only to be understood as an outcrop analogue to Middle East carbonate reservoirs. - the precision has been added. "Although Urgonian microporous carbonates of Provence are analogue to Mid-

dle East reservoirs, the analogy can be extended to other faulted microporous carbonate reservoirs." (lines 52 to 54)

Geological context / Data Base - Same as in the introduction, there is a mixing and jumping of the description of stratigraphy and structure. This needs to be clearly separated. - Here the basin stratigraphic evolution is related to the structural evolution. As they are closely related, it is difficult to separate them.

- Figure 1 caption starts with "Geological context of the study area", but only a geographical outline of France is shown. Also coordinates are missing, and it is essentially impossible for someone not familiar with the study area to readily locate it. Marking the study area in the map inset with a rectangle much larger than the area of interest and only providing "near Marseille (Fig. 1A)" (line 76) is not sufficient. We added a regional map and coordinates within the figure 1 to precise the location of the study area.

- In the chapter "Data Base" it is not clear if its content is derived from the authors analysis or from existing literature. If it is the former, it should be moved to the results chapter. Done. The structure of this part has been modified

- The data base deals with faults, sub-faults, sets, transects, units, etc.. This easily gets very complicated and therefore it is paramount that the paragraphs are well structured to guide the reader through this complexity. Unfortunately, at the moment, this is not the case. As an example: Line 106 "The set one, constituted of F3 and F4, is: : :". The problem here is that F3 and F4 have not been defined before. Hence, the reader does not understand this abbreviation and is left in confusion. Done. The terms have been defined

Results: - In Figure 3c, pore types are shown. First, the resolution of the photos must be improved. And second, and this is now maybe more a matter of perspective, but it seems that the two blue pores are just the result of grains falling out of the sample during thin section preparation. Either way, from the picture shown, it is very difficult to reconcile that the host rock has a porosity larger 10 %. - The quality of the photos

correspond to resolution of the Olympus_ BH2 microscope and to a Zeiss_ MR C5 camera. The porosity of the sample is mainly due to micro-porosity (the precision has been added) "From thin sections impregnated with blue-epoxy resin, a porous rock-type with $\varphi$>10% mainly in micritized grains as microporosity and moldic porosity" (lines 70-71). Moldic porosity only represent ∼1% of the porosity in such microporous carbonates (see Fournier et al 2011; Léonide et al., 2014, Fournier et al ; 2014). The majority of porosity is within the micro-pore of the micrite (non-visible on the photo; size < 10$\mu$m; (Deville de Periere et al. 2011)). In any case the porosity related to grain falling is <1% of the total porosity.

- Subchapter "Carbonate and Oxygene Isotopes": This is again about consistency, and please excuse for being picky: 189 measurements have been made on 16 samples and 32 thin sections (Line 250; set aside the confusion of what the difference between thin section and sample is), and these distribute on 49 bulk rock, 48 vein, 40 fault rock, and 26 intergranular space measurements. The latter list however adds up to "only" 163 measurements. And table 2 shows even 204 measurements. How do these numbers fit? The correction have been done. We homogenized the number of measurements

Chapter IV, 2: Fault related diagenesis - The start of this chapter deals about potential dilation and I am afraid this discussion is on a weak basis. The authors themselves mention that "[dilation processes under low confining pressure] is only possible in highly porous granular media." (line 344). It is my impression that this attribute does not apply to the here analyzed rock, which is described as cemented and only comprising ">10 % porosity but located in the grains" (line 336), i.e. secondary porosity due to partially dissolved grains. As a comparison, Alikarami & Torabi (2015), to which the authors often refer in this regard, deal with quartz sand with porosities of 33-45 % of primary origin. This is a significant difference. - When the dilation band nucleate, the host rock was not totally cemented and presented porosity higher than the current one. Though, we made a more marked separation between description of dilation bands within highly porous rock and Castellas host rock.

- In line 358 it is claimed: "In the Urgonian carbonates of La Fare sector, dilatant processes enhanced fluid circulation in the rock along the deformation bands and led to the cementation of C1b". Unfortunately, deformation bands in the study area have neither been mentioned nor described before and thereafter in the manuscript. - The instability of dilation bands would have led to a collapse of these structures. Moreover, as the dilation bands occurred as an incipient faulting mechanisms, the later fault development and reactivation would have altered the dilation band morphology.

Line 381-400 deal with formation temperatures of cement generation C3: "- For the calculation, "the formula of Ali (1995)" is used. 1. Ali (1995) presents more than one formula. 2. Ali (1995) is not the original reference! It is Epstein et al. (1953) and Craig (1965) that needs to be cited. 3. As far as I know (I might be mistaken), the equation of Epstein et al. (1953) is based on biogenic calcite. The authors might want to check the equation of Kim & O'Neil (1997, Geochimica et Cosmochimica Acta) for inorganic calcite. Though in the end it might as well not make a significant difference for the authors calculation. - "temperature of initial fluids: 33_C to 34_C (Littler et al., 2011)" 1. What is meant with "initial fluids"? 2. Littler et al. (2011) present own data on paleo-sea-surface temperatures, which they set in comparison with existing data. The temperature range extends beyond 33-34_C. 3. If original data of Littler et al. (2011) is used, this means that data of the Hauterivian (133-129 Ma) is used. The authors however give an age estimate for C3 as _Cenomanian (101-94 Ma). This may be a significant age difference and suitability of the Hauterivian temperatures for the calculation needs at least be discussed. 4. If the data quoted by Littler et al (2011) is used, then the original work needs to be cited. - "meteoric water: -4.0 d18O (Robinson et al., 2002)" 1. This is the same as with Littler et al. (2011). Robinson et al. (2002) present data from the Barremian, whose suitable application needs to be discussed. 2. The -4.0 value is an average of Robinson et al. (2002) data. 3. Most important: the -4.0 d18O of Robinson et al. (2002) refers to d18O of carbonate and not to d18O of the fluid from which the carbonate precipitated. - Line 393: "We calculated a C3 fluid temperature 40_C and 60_C." 1. As the authors do not guide through the calculation

and how the parameters have been applied, it is impossible to follow how these values have been determined. - For the calculation of the formation depth, a geothermal gradient of 26.4_C/km (Ali, 1995) is used. Such precision is quite ambitious." - We removed this part. We decided to remove this part from the manuscript for 2 reasons: (1) to shorten the "fault related diagenetic" part and, (2) $\delta$18O during Durancian uplift (Aptian/Albian) was difficult to estimate. Moreover, the association of burial/uplift curve and $\delta$13C values allow an interpretation of the fluid origin. "As C3 cementation occurred during the Durancian uplift and denudation, C3 most probably did not cemented at high depth (depth of maximum 500m; Fig. III. 9C4).The negative $\delta$13C values tend corroborate that it would rather be a meteoric fluid than a marine fluid." (lines 353-356)

- Line 395: "The negative d13C values tend to indicate that it would rather be a meteoric fluid than a marine fluid." 1. The presented d13C data range from -2.09 to +1.22. Did the authors rather mean d18O? 2. Why does it indicate rather a meteoric fluid? This needs to be discussed. The d18O (VPDB) carbonate value does not per se indicate the type of fluid from which the carbonate precipitated. This depends on the formation temperature. - $\delta$13C is the good isotope here. Depletion of $\delta$13C values tend to indicate a meteoric influence.

- Line 399: "As C3 cementation occurred during the Durancian uplift and denudation, C3 most probably did not cemented at high depth (Fig. 9C4). More probably, C3 fluids were meteoric burial fluid which were upwelled under tectonic stresses". 1. What are the arguments for the formation age of C3? This has not been discussed before. It is simply claimed here that it formed during the Durancian uplift. Why can it not be related to e.g. the Pyrenean shortening? - Cross-cutting relationships have been added within part "diagenetic evolution of the fault zones – discussion" lines 266 to 275

2. Whether true or not, the authors need to better explain in more detail why fluid upwelling is the likely process. From the current information given, it is difficult to follow the line of reasoning. - Modifications effected.

In the conclusions, line 547-550: "All diagenetic stages [: : :] result from low temperature flows with important meteoric water input. This low temperature disproves any hydrothermal influence. Therefore, both fault zones were not linked to high depth basement faults." - Set aside the uncertainty in determining formation temperature and source of fluid, absence of hydrothermal fluids does not permit conclusions on the deep structure of a fault. The fault may very well be connected to a basement fault, but the fracture connectivity may just be poor. - Modifications effected the paragraph has been modified. "All diagenetic stages, including cementation and dolomitization, result from low temperature flows with important meteoric water input. This low temperature flows associated with the deformation and cementation types and, the lack of mineralisation specific to high temperature flows disprove any hydrothermal influence." (lines 502-505)

Please also note the supplement to this comment:
https://www.solid-earth-discuss.net/se-2019-153/se-2019-153-AC2-supplement.pdf

———————————————————

[Figure]

**Fig. 1.**

[Figure]

**Fig. 2.**

A | *Castellas Fault zone*

B | *D19 Fault zone*

C

**Fig. 3.**

[Figure]

Fig. 4.

[Figure]

**Fig. 5.**

**Fig. 6.**

[Figure]

**Fig. 7.**

**Fig. 8.**

[Figure]

Fig. 9.

[Figure]

**Fig. 10.**

---

## Author Comment (AC4) · 15 Jan 2020

Dear referee, I am pleased to send you the revised version of our paper on "Diagenetic evolution of fault zones in Urgonian microporous carbonates, impact on reservoir properties (Provence – SE France). You will find enclosed in the supplement, the comments and to your remarks. They are listed together with the actions made: - Comments are in italics - Corrections validated are emphasized in green - Corrections with a red bold part are considered un-useful or inappropriate You will see that most of corrections have been respected as you requested. Best regards, Irène Aubert I. General remarks 1. Authors should better illustrate the crosscutting relations among the different struc-

tural elements (dilational bands, open fractures, shear fractures, etc.) measured both within and outside the study fault zones. ïČŸ Chronological relation between cementation events has been added. Lines 266 to 275

"2. Rename section IV as "Discussion", and shorten the single chapters it includes (mainly, the chapter on the fault-related diagenesis)". ïČŸ Done

3. Expand data discussion on the impact of fault zones on reservoir properties by adding references to other surface analogues worldwide. ïČŸ Done. we added references lines 338-340, 362-364 and 378-380

II. Specific Comments Abstract: "please check for wrong punctuation marks, grammar, and syntax." ïČŸ done

Introduction: "please remove the final sentence." ïČŸ done

Geological context: "check for grammar; remove lines 110-124 (out of place)." ïČŸ done

Methods: "please double check the standards used for stable isotope analyses." ïČŸ done

Results: "please check both grammar and syntax." ïČŸ done

Diagenetic Evolution: "please shorten the whole section, and rename it (cf. comment above);" ïČŸ done – section was shorten by removing the part on C3 temperature calculation

"re-consider dilation as an incipient faulting mechanisms;" ïČŸ done – we added this sentence : "Hence, in the Urgonian carbonates of La Fare sector, dilatant processes occurred as an incipient fault mechanism and enhanced fluid circulations along the deformation bands." (line 327-328)

"please separate data interpretation for granular media from that inferred for a cohesive rock;" ïČŸ Modifications achieved. We have made a more marked separation between

description of dilation bands within highly porous rock and Castellas host rock

"please state all assumptions required for temperature calculation based upon oxygen isotope data and well-known fractionation curves." ïČŸ We removed this part. We decided to remove this part from the manuscript for 2 reasons: (1) to shorten the fault related diagenetic part and, (2) $\delta$18O during Durancian uplift (Aptian/Albian) was difficult to estimate. Moreover, the association of burial/uplift curve and $\delta$13C values allow an interpretation of the fluid origin. "As C3 cementation occurred during the Durancian uplift and denudation, C3 most probably did not cemented at high depth (depth of maximum 500m; Fig. III. 9C4).The negative $\delta$13C values tend corroborate that it would rather be a meteoric fluid than a marine fluid." (lines 353-356)

Please also note the supplement to this comment:
https://www.solid-earth-discuss.net/se-2019-153/se-2019-153-AC4-supplement.pdf

---

## Referee Report (RR1)

Review of manuscript **se-2019-153** submitted to Solid Earth, Aubert et al. "***Diagenetic evolution of fault zones in Urgonian microporous carbonates, impact on reservoir properties (Provence - SE France)***" by Mattia Pizzati.

**General remarks**

Dear Authors and Editor, below you can find the review of the submitted manuscript. Revisions are made by describing the issues found line by line and also on the text file of the manuscript, in which critical points were highlighted in green. Comments on figures and figure captions are presented at the end of this file as well.

The manuscript deals with the diagenetic and petrophysical evolution of two fault zones affecting Cretaceous carbonate rocks. The adopted methodology combines field (geological survey) and laboratory analyses (petrographic observation, cathodoluminescence, stable isotopes).

Although the scientific part of the manuscript is of great interest, the quality of the written English is often limiting the full comprehension of the geological topics the authors are discussing. I fully understand how difficult can it be to translate concepts and descriptions in a fluent manner, since I'm not a native English speaker as well, still there is a lot of work to be done before this manuscript can be considered suitable for final publishing. I'm sorry to be so blunt on this, but it was also a bit disappointing noting such a huge number of errors as this version should be the fourth one and it underwent several review processes. I recommend all the authors to carefully review word by word the entire text and read it several times. As you will notice, I tried to highlight every issue found in the pages below, so you will find a lot of corrections to be made. Do not be scared though, the vast majority of them are related to typo fixing, grammar corrections and sentence reworking.

Figures are overall of good quality, and I insert a few suggestions to improve their visibility and minor corrections to be done on legends. You will find the exact position of the suggested changes also in the pdf file as sticky notes next to the figures.

Relative to the scientific part of the manuscript, my main concern is related to the final model in which the permeability contribution was calculated. To me it feels like a good-faith effort, but it would need to be grounded on your field data. To be more explicit, the percentage you calculate for permeability from fault core, fracture network and matrix, should be calibrated according to the width of the fault structure, fracture connectivity and so on. I firmly believe that you could address this inside the discussion section, to provide much stronger constraints to the model you are presenting.

I think this paper is worth to be accepted on the Solid Earth special issue after the completion of moderate to major revisions depending on how much text has to be revised or partially rewritten. I really hope this review will be helpful to the improvement of the final version of your manuscript. Do not hesitate to contact me in case any questions or doubts arise from these comments: mattia.pizzati@studenti.unipr.it

**Detailed comments line by line**

Line 10: Please make plural the term "*reservoir*" changing it to "*reservoirs*"; moreover, I would erase "*high a*" before "*permeability variability*".

Lines 12-13: Please add "*diagenetic*" before "*processes*"; modify "*that modify*" with "*capable of modifying*".

Line 13: Please change "*Focussing*" with "*Focusing*".

Line 14: Please move the word "*impact*" before "*the fault zone*". Rather than using "*reservoir properties*" maybe it is better to adopt a more general term such as "*reservoir quality*".

Line 14: I would change "*It*" with "*This contribution focuses*...". Throughout all the text try to use the same writing style. If you are describing a number of analyses or samples than use the form "*69 samples*", while if you are referring to the number of fault zones, fracture sets and so on, write as "*two fracture sets*" .The number "*2*" should have been written in words "*on two fault zones*".

Line 15: Please correct "*La Fare Anticlinal*" with "*La Fare Anticline*". In the same sentence change from "*which cross-cut*" to "*cross-cutting*".

Line 16: Please correct as follows from the form "*orthogonal to the fault zones*" to "*orthogonal to fault strike*". It this way the reader is aware that you worked on exposures that cross fault zones parallel to the direction of tectonic transport.

Line 17: Please change from "*Diagenetic elements were determined on 92 thin section...*" to "*Diagenetic history was determined through the observation of 92 thin sections...*".

Line 18: Why are these words "*Polarized Light Microscopy*" in capitals? Is that necessary?

Line 18: Maybe here it is better to state more precisely "*stable isotope measurements*" rather than "*isotopic measurements*".

Line 19: Here, I would modify "*2*" with "*two*".

Line 20: Please correct "*highlight*" with the third person form "*highlights*".

Line 20: Here, I would modify "*2*" with "*two*".

Line 20: Please modify and make "*drain*" plural "*drains*".

Line 21: Please add a line separator between "*low temperature*" as follows "*low-temperature*".

Line 21: Here I would erase "*fault zone*" before "*calcite cementation*".

Line 22: Here, I would modify "*2*" with "*two*".

Line 22: You are mentioning here "*two subsequent phases*". Do you refer to tectonic activity or to diagenetic ones? Please be more specific.

Line 22: Please add "*petrophysical*" before "*properties*".

Line 25: Please change "*porosities*" with "*porosity values*".

Line 26: Please change "*heterogeneous properties*" with "*heterogeneity*".

Line 26: Please correct "*depend*" with the third person form "*depends*".

Line 27: Please erase "*they*" and add "*carbonates may*" before "*determine*".

Line 29. Please correct the beginning of the sentence as indicated "*moreover, fault zones...*".

Lines 32-33: Here, I would modify the sentence as follows: "*Fault zones in cohesive rocks are complex structures, typically composed of two separate domains, namely the fault core and the damage zone. The vast majority of deformation is usually accommodated inside the fault core, while the damage zone is subjected to delocalization and partitioning of strain and is encompassed by the undeformed host rock*". You can keep the reference list and distribute the references according to the topic they deal with (damage zone or fault core).

Line 37: Instead of "*mixed zones*", maybe is better "*structures with mixed hydraulic behaviour*".

Line 37: Please correct "*depending of*" with "*depending on*".

Line 38: Please make singular "*fluid flows*" to "*fluid flow*".

Line 39: Please correct "*Earth crust*" with "*Earth's crust*".

Line 40: Here, maybe you can change the structure of the sentence as indicated: "*, and are capable of increasing the...*".

Line 40: Please correct " *fluids-rock interaction* " with "*fluid-rock interaction*".

Line 41: You can add "*diagenetic*" before "*secondary processes*".

Line 42: Please add a line separator between "*Fault related*" as follows "*Fault-related*".

Line 45: Here maybe I would change "*duplication of fluid pathways/barriers*" with "*repeating fluid pathway-barrier behaviour in time leads to...*".

Line 48: Please erase "*of the*" before "*faulting*" and restructure the sentence as indicated: "*Hence, understanding faulting and diagenetic processes is crucial...*".

Line 51: Maybe here "*formations*" should be capitalized and also please erase the space after the word "*formations*".

Line 54: To what other carbonate reservoirs are you referring here, please be more specific and if necessary list them.

Line 56: Please add a hyphen between "*poly-phasic*", as you did above.

Line 60: Here, I would modify "*2*" with "*two*".

Line 60: Please correct "*crosscutting*" with "*cross-cutting*".

Line 60: Please erase "*facies*" before "*carbonates*".

Line 61: Please capitalized "*South-East Basin*".

Line 63: With "*larger extension*" are you referring to the areal extension of the carbonate platform? If so, you should correct as follows "*The Urgonian platform carbonates reached their maximum areal extension...*".

Line 64: Is necessary the hyphen between "*Fenerci-Masse*"?

Line 65: Please correct "*bauxite deposits*" with "*bauxite deposition*".

Line 67: Please modify these words as follows: "*, and development of E-W-trending extensional faults*".

Line 67: Is necessary the hyphen between "*Guyonnet-Benaize*"?

Lines 68-69: Here I would restructure the sentence as indicated: "*During Late Cretaceous times, in a shallow-water platform environment, a transgression led to widespread deposition of rudists (Philip, 1970)*".

Line 70: Please change the structure of this part of the sentence: "...*between Iberian and Eurasian plates caused a regional...*".

Line 70: Please modify from "*cited references*" to "*references therein*".

Line 71: Please modify from "*cited references*" to "*references therein*".

Line 72: Please erase "*which*" before "*gave rise*".

Line 72: Here you should add "*E-W-trending north-verging thrust faults*" otherwise the reader may not fully understand how faults are oriented.

Line 72: Please change "*ramp folds*" with "*thrust-related folds*".

Line 73: Please modify from "*cited references*" to "*references therein*".

Line 75: Please change "*dimly*" with "*weakly*".

Line 76: To what "*structures*" are you referring? I guess they are the contractional ones.

Line 79: Here, I would modify "*2*" with "*two*".

Line 79: Maybe it is better to write "*a Km-scale*" instead of "*kilometric-scale*".

Line 79: Please change from "*...fault system on the E-W-trending...*" to "*...fault system related to the E-W-trending...*".

Line 80: Can we state that the southern limb is actually the backlimb of the La Fare Anticline?

Line 80: Please correct "*anticlinal*" with "*anticline*".

Line 82: Please add a space between "*120m-thick*" to separate the length from the measurement unit "*120 m-thick*".

Line 82: Please change "*calcarenite unit*" with "*calcarenitic unit*".

Line 83: Please add a space between "*40m-thick*" to separate the length from the measurement unit "*40 m-thick*". Try to do the same and keep this style throughout all the text.

Line 83: Please change the structure as indicated: "*...coral-rich calcarenite unit, and an upper 10 m-thick....*". Separate the length from the measurement unit.

Lines 84-85: Here, I would modify as follows: "*Santonian age coarse grained rudist limestones unconformably overlay the Barremian carbonates (Fig. 1A)*".

Line 86: Here you state that the Castellas Fault has a length of one Km, but in Fig.1A the trace of the fault is at least three Km. Please correct accordingly with the real length.

Line 86: You should also define the kinematic of the fault: I would recommend to write as follows: "*left-lateral strike-slip fault*", before the info concerning the trend and dip.

Line 87: Here change the structure of the sentence as suggested: "*(Fig. 2A, B; table 1). Its structure comprises horses, secondary faults and tectonic lenses (Fig. 2A, C; Aubert et al. (2019b))*". Check if in Solid Earth is necessary to repeat the figure number when you are citing 2 sub-images from the same figure (es. Fig. 2A, 2C or Fig. 2A, C): try to do the same in the entire manuscript. There is a parenthesis missing at the end of the sentence where you cited Aubert et al. (2019b).

Line 88: You can modify from "*The second fault zone*" to "*The second investigated fault zone*".

Line 88: Here, I would modify "5" with "*five*".

Line 89: Maybe here change "*50m-long interval*" with "*50 m-wide outcrop*". Pay attention to separate the length from the measurement unit. "*Interval*" is a too generic term, while here I guess you are describing an outcrop with 50 m of lateral extension.

Line 90: Maybe "*Sub-fault are organised in two sets*" sounds better than "*Sub-faults are made of 2 sets*".

Line 90: Instead of "*Set one*" use "*The first one*".

Line 91: Please change "*orange on Fig. 2F*" to "*orange traces in Fig. 2F*".

Line 91: Add the kinematics of the fault set "*with left-lateral strike slip...*".

Line 92: Please change "*orange on Fig. 2F*" to "*red traces in Fig. 2F*".

Line 92: Here, I would modify "5" with "*five*".

Line 93: What is this asymmetry about? Is related to a different structure or width of the hanging wall and footwall damage zones? If so it think it should be mentioned here briefly.

Line 95: Please add "*distinct*" before "*tectonic events*".

Line 96-98: I tried to modify the two sentences as indicated: "...*the Middle-Cretaceous Durancian uplift leading to extensional en echelon faults. The Castellas fault nucleated during this first extensional event and bears early dip-slip normal striations (Matonti et al., 2012).*".

Line 100: I merged the first two sentences: "...(*see references cited in Espurt et al. 2012), which reactivated the...*".

Line 101: Please change "*leads*" to "*led*".

Line 101: Please modify "*neo-formed*" with "*newly-formed*".

Line 105: Please add "*present-day*" before "*reverse throw*".

Line 108: Here, I would modify "*4*" with "*four*".

Line 108: Throughout all the text I saw that sometimes you have used numbers to identify transects (1, 2, 3, 4), but I also found abbreviations such as T1. I suggest you to use only one style and to keep it for the entire text. In the following corrections I considered the transects to be identified by their number, so maybe I would erase "*(T1 to T4)*" from this line.

Line 109: Please change "*transect T1*" with "*transect 1*".

Line 109: Instead of "*bed*", which can be misunderstood maybe here you can use "*lithostratigraphic unit*".

Line 109: Please correct "*pelloidal*" with "*peloidal*", I think it should be the correct form.

Line 110: Please erase the space in "*Fig. 2D a*" to "*Fig. 2Da*".

Lines 110-111: You can restructure the sentence as suggested: "*Transects 2 and 3 cross-cut unit 3, made of fine-grained calcarenites with peloidal grains...*". Be careful to correct "*pelloidal*" to "*peloidal*".

Line 112: Make the plural form of "*echinoderm*" to "*echinoderms*".

Line 114: Make the plural form of "*amount*" to "*amounts*".

Line 114: Make the plural form of "*bryozoan*" to "*bryozoans*".

Line 115: Eliminate the comma in the reference to the figure: "*Fig. 2G, a*" change to "*Fig. 2Ga*".

Lines 117-119: I tried to restructure the sentence as follows: "*Three different fault rock types were identified in the fault core of the two investigated fault zones (see Aubert et al. 2019b; Matonti et al. 2012)*".

Line 119: Please change "*normal*" with "*extensional*".

Line 121: Add this detail at the end of the sentence: "*...<30% of fine-grained grey matrix*".

Line 122: Add "*strike-slip*" before "*reactivation*".

Line 122: Please add "*to*" before "*the onset*".

Line 123: Please correct the third person form "*present*" with "*presents*" and also change "*2*" with "*two*".

Line 125: Here maybe modify the sentence in this way: "*...sub-rounded clasts belonging to the nearby damage zone...*".

Line 126: What is the nature of the cemented matrix? Are you able to distinguish the composition and the type of cement? If so add these details at the end of this sentence.

Line 127: Here you are mentioning the reactivation of the D19 fault zone: be more specific and define the kinematics and the age in which it occurred.

Line 128: Make the plural form of "*clast*" to "*clasts*".

Line 128: Add this word at the end of the sentence: "*from the nearby damage zone dispersed in an...*".

Line 133: Here, I would modify "*4*" with "*four*".

Line 134: Please correct the sentence as indicated: "*Microfacies were determined...*". Petrography is just too generic.

Line 136: Please, modify this detail: "*with a solution of hydrochloric acid, Alizarin Red S and potassium ferricyanide...*".

Line 137: Please erase "*The*" at the beginning of the sentence and capitalize "*Thin...*".

Line 137: Please modify "*analyzed*" with "*analysed*". Pay attention to the use of UK or USA English. Solid Earth is an European journal so it think you should always refer to the UK English.

Line 138: Please add these words: "*...the different generation of calcite cements*".

Line 140: Is this underscore necessary to identify the instrument?

Line 141: Is this underscore necessary to identify the instrument?

Line 141: Here, I would modify "*2*" with "*two*".

Line 142: Here, I would modify "*2*" with "*two*".

Line 143: Please add "*beam*" before "*current*".

Line 143: Keep always a space between the value and the measurement unit; do this in all the text. "*20 KV*" and not "*20KV*".

Line 143: Please make the plural form of "*surface*" "*surfaces*".

Line 146: Keep always a space between the value and the measurement unit: "*80 µm,*" and not "*80µm* ".

Line 147: What do you mean with the words "*bulk rock*"? Are you referring to the undeformed host rock outside the damage zone, if so I think you should correct this and be more precise.

Line 148: Please add "*isotopic*" before "*values*".

Line 149: Is correct the name of the spectrometer with the symbol "+" rather than the word "*plus*"? Please check again.

Line 151: The symbol you used for the delta is not the same adopted previously in Line 144. Choose only one and keep it in the entire text. I would keep this form "*$\delta^{13}$C, $\delta^{18}$O"*.

Lines 151-152: I modified the sentence as follows, check if suits you: "*The standard deviation (SD) of the measurements is <0.1‰ and <0.2‰ for $\delta^{13}$C and $\delta^{18}$O, respectively*".

Lines 159-160: I tried to fix in this way: "*Porosity measured on 92 plug samples show a strong decrease towards the fault core (Fig. 3): it drops from more than 10% in the undeformed host carbonates...*".

Line 161: Please change "*in*" with "*within*".

Line 163: Please add these words at the beginning of the sentence: "*Along transects, some porosity variations...*".

Line 164: Insert a space between the value and the measurement unit: "*60 m*" and not "*60m*".

Line 164: Please change "*transect T2*" to "*transect 2*".

Line 165: Please change "*low < 7%*" to "*lower than 7%*".

Line 165: Delete the space after 1.53.

Line 165: Substitute the comma at the end with a full stop.

Line 166: Here I would write "*is wider than 40 m*" rather than "*is >40m*".

Line 166: Insert a space between the value and the measurement unit: "*30 m*" and not "*30m*".

Line 167: Here please change "*In a 10m-thick*" with "*In a 10 m-wide*".

Line 168: If you want to keep the nomenclature used above than change "*T1 and T3*" with the form "*transect 1 and 3*".

Line 169: Please add "*found*" between "*are in narrow*".

Lines 169-170: I tried to write as follows this part of the sentence: "...*in narrow zones (less than 2 m-wide) around the faults...*".

Line 170: With the term "*lens*" are you describing the rock volume comprised between F4 and F5?

Lines 172-173: Please modify the beginning of the sentence as indicated: "*Microscope observations of thin sections impregnated with blue-epoxy resin allowed to identify a porous rock-type with ϕ>10%, mainly composed of micritised grains...*".

Line 173: Please remove the space in "*Fig. 3C a*" to "*Fig. 3Ca*".

Line 174: Add a comma after "*ϕ<5%*".

Line 174: Please remove the space in "*Fig. 3C b, c*" to "*Fig. 3Cb, c*".

Line 174: It is not clear to me what do you mean with "*barren stylolites*". Is this word related to the incipient nature of stylolites? Please correct also the spelling of the word "*styloliths*" with "*stylolites*".

Line 174: Erase "*are distinguished*" at the end of the sentence.

Line 177: Please correct "*micritized*" with "*micritised*", if you want to keep the UK version of the word.

Line 177: Here, I would modify "*2*" with "*two*".

Line 178: Here, I would modify "*2*" with "*two*".

Line 179: Why is this citation reported in italics?

Line 180: The reference to "*Fig. 4A, 4B*" should be "*Fig.4A, B*".

Line 181: Here I'm really struggling with the terminology you used "*puntic and serrate*" are terms not widely used in literature.

Line 182: Please put the porosity value between parenthesis and erase the space before the comma.

Line 183: Please insert the porosity percentage in parenthesis.

Lines 177-183: Inside this paragraph you should also give some info concerning the crystal size of the micrite.

Line 185: Please add "*different*" between "*Eight cement*".

Line 185: "*Alizarin Red S*" should be with all first letters in capitals.

Line 185: Add "*and*" after "*coloration*".

Line 186: Please correct "*made up of*" with "*made of*".

Line 186: Please correct from the third person form to plural "*exhibits*" to "*exhibit*".

Line 188: Here, I would modify "*2*" with "*two*".

Line 189: With "*thickness*" here are you describing the maximum thickness of the isopachous calcite fringe? Is that possible to assimilate this data to the crystal size?

Line 189: Here, I think it would be more correct to use this symbol "~" to indicate the word "*about*"; also insert a space between "*10*" and "*µm*".

Line 190: Here, I would modify "*2*" with "*two*".

Line 190: Add an hyphen between "*dog-tooth*".

Line 192: Again, here is that possible to describe the crystal size rather than the "*thickness*" of the cement?

Line 192: Here, I think it would be more correct to use this symbol "~" to indicate the word "*about*"; also insert a space between "*100*" and "*µm*".

Line 192: Be more specific concerning the reference to the figure: I believe here you are referring to Fig. 5A, is that right?

Line 193: Better than "*C1b values*" maybe you should use "*C1b areal occurrence*".

Line 193: Please correct the third person form "*increases*" instead of "*increase*".

Line 201: Please change the sequence of words from "*replacive phases occur largely...*" to "*replacive phases extensively occur...*".

Line 203: Use the hyphen between "*dull orange*" "*dull-orange*".

Line 203: Please change the structure as indicated from "*...only found in fault core veins*" to "*...only found in veins of the fault core*".

Line 204: Please erase "*elements*" after "*Si and Al*".

Line 205: Please erase "*an*" and add at the end of the sentence "*and have black luminescence.*"

Lines 203-205: Please add also some info relative to the size of the cement crystals you are describing in this section.

Line 206: Use the hyphen between "*red dull*" "*red-dull*".

Line 208: Add "*only*" before "*to the fault zone*".

Lines 206-208: Please add also some info relative to the size of the cement crystals you are describing in this section.

Line 211: Please insert a space between "*500*" and "*µm*".

Line 211: Please insert "*previous*" before "*dolomitization phase*".

Lines 212-213: Please change here from "*micritic inclusion in the crystal and...*" to "*micritic inclusions inside crystals and...*".

Line 215: Please insert a space between "*3*00" and "*µm*".

Line 217: Please correct the reference to "*Fig. 5G, 5H*" with "*Fig. 5G, H*".

Line 219: Here, I would modify "*2*" with "*two*".

Line 219: Please erase "*which*" after "*C4a*".

Line 220: Please correct the plural form of "*band*" with "*bands*".

Line 220: Please add "*thin*" before "*non-luminescent zones*". Change also "*bands*" with "*zones*" to avoid any repetition.

Line 221: Correct the nomenclature of transects, here maybe use "*transects 1 and 3*" as you did in the text above.

Line 223: Use the hyphen between "*red dull*" "*red-dull*".

Line 227: Please change "*formation*" with "*karst deposit*".

Lines 228-229: See if this correction suits you: "*This karst deposit present a stack of alternating micrite-rich and grain-rich layers, the latter composed of former blocky calcite cement belonging to dissolved grainstones*".

Line 230: Please correct "*clasts fall*" with "*grain fall*". Keep in mind that the term clast is used for fault rock material, while grain is more generic and does not mean that any deformation occurred. Following this, use clasts to describe a fault breccia and grains if you are dealing with undeformed sediments or rocks.

Line 230: Please correct the singular "*has*" to plural form "*have*".

Lines 230-231: I tried to improve the clarity of the sentence: "*Micritic layers have been observed under SEM, and the micrite composing them appears tight with subhedral mosaic crystals (Fig. 4F)*".

Line 233: Please change "*proportion*" with "*areal amount*" and also modify the third person form of "*increase*" to "*increases*".

Lines 236-238: I fell this is a repetition of what was previously presented inside the method section, maybe it is better to do not duplicate this inside the results.

Line 239: Again, what is the meaning of "*bulk rock*"? If you are referring to the undeformed host carbonates than you should be more precise and state clearly. Do this in all the manuscript. I highlighted this issue every time I noticed it.

Line 239: "*Intergranular volume*" is better than "*intergranular space*".

Lines 241-242: If it is more correct change the reference to the figure from "*Fig. 6A, 6B*" to "*Fig. 6A, B*".

Line 242: Again, "*bulk rock*" isn't this the "*host rock*"?

Line 243: Here, I would modify "*2*" with "*two*".

Line 243: For "*Set one*" you can use the number "*Set 1*". I believe to identify the type of fracture sets, isotopic datasets and so on you can adopt the number instead of the spelled word.

Line 244: I don't think that using the symbol "*&*" is suitable here, just write "*and*".

Line 245: As above, "*Bulk values*" is too generic. If you are describing the isotopic dataset of the undeformed carbonates adopt "*Host rock isotopic values*".

Line 247: Please erase "*the*" before "*transect 3*".

Line 247: Please erase "*along transect*" after "*slightly vary*".

Line 248: At the end of this sentence after "*$\delta^{13}C$* " add "*, respectively*".

Line 248: Change "*Contrarily*" with "*On the contrary, ...*".

Lines 248-251: I tried to fix these sentences as follows: "*On the contrary, values are more variable along the D19 transect; they range from -9.18‰ to -5.20‰ for $\delta^{18}O$ and from -4.80‰ to -0.60‰ for $\delta^{13}C$ (Fig. 6C, table 2). The $\delta^{13}C$ values decrease approaching to faults, especially south of F2.*". Please check carefully this statement, because in Fig. 6C the isotopic data for D19 do not show values of $\delta^{18}O$ lower than -8.00‰ and $\delta^{13}C$ values lower than -3.50‰. So check back the values reported in the graph and correct the text accordingly.

Line 252: Please change "*spaces*" with "*volume*".

Line 252: Maybe "*infillings*" is better than "*fills*".

Line 252: Please make the plural form "*fault rocks*".

Line 253: Here, I would modify "*5*" with "*five*".

Line 254: Personally I modified the sentence as follows and fell like it is clearer: "*isotopic values of C1 cement ...*".

Line 256: Similar to the comment above: "*isotopic values of C3 cement...*".

Line 258: Again: "*isotopic values of C4 cement...*".

Line 258: Maybe "*infillings*" is better than "*fill*".

Line 259: Please add a space between "*from-5.10‰*" to have "*from -5.10‰*".

Line 260: Please erase the space between "*FR 2*" to "*FR2*".

Line 260: Please add a space between "*from-6.55‰*" to have "*from -6.55‰*".

Line 261: Maybe "*infillings*" is better than "*fill*".

Line 262: Please restructure the last part of the sentence as follows: *"...for $\delta^{18}O$ and $\delta^{13}C$, respectively*".

Line 263: Here, I would modify "*4*" with "*four*".

Line 263: Here, I would modify "*2*" with "*two*".

Line 264: Please add a semicolon "*;*" at the end of the sentence.

Lines 265-266: Please start the sentence as follows: "*isotopic values of C5 cement, sampled in FR1 matrix display mean...*". Add also a semicolon "*;*" at the end of the sentence.

Lines 267-268: Please start the sentence as follows: "*isotopic values of FR3 matrix have a mean...*". Add also a semicolon "*;*" at the end of the sentence.

Line 275: Please change "*thanks to*" with "*via*".

Line 275: Please pluralise "*cross-cutting relation*" to "*cross-cutting relations*".

Line 276: I modified the order and position of some words here: "*Indeed, the veins filled with C2 cement cross-cut C1a and C1b cements (Fig. 5B)*".

Line 277: I would change this sentence as follows: "*Thus, C2 cementation postdated C1 cement*".

Line 277: Please erase "*The*" at the beginning of the sentence.

Line 278: Here there is a missing reference to the correct figure. I guess it should be "*Fig. 5B*".

Line 278: Please change "*is ante-FR1...*" with "*formed prior to FR1...*".

Line 279: Please change "*post-C2*" with "*after C2*".

Line 279: I would use "*extensional*" rather than "*normal*".

Line 281: Please remove the comma after "*formation and, are related...*".

Lines 281-282: Please modify the sentence as indicated: "*Replacive dolomite is found within FR1...*".

Line 282: Here I guess that this is the wrong reference to the figure. I believe it should be "*Fig. 5E*" and not "*Fig. 3E*".

Line 282: Please make the past simple version of "*develop*" "*developed*".

Line 282: Please, move "*C4*" before "*cement*".

Line 283: Please change "*postponed*" with "*postdated*".

Line 284: Please modify the order of the words: "*...developed during the strike-slip reactivation of the studied faults*".

Line 287: Please correct "*La Fare anticlinal*" with "*La Fare anticline*".

Line 287: Here, I would modify "*3*" with "*three*". Also substitute "*important*" with "*major*".

Line 291: I don't know if this is the right term, but perhaps "*micro-boring organisms*" sounds better than "*micro-bores*".

Line 294: Please add a hyphen between "*low and energy*" "*low-energy*".

Line 294: Please add "*environment*" at the end of the sentence after "*inner platform*".

Line 295: Move "*C0*" before "*cement*".

Lines 295-296: I tried to fix this sentence: "*...formed around grains giving rise to a solid envelop, inducing the preservation of the original grain shape during the late-stage burial compaction (Step 0 in Fig. 8)*".

Line 299: Here, I would modify "*2*" with "*two*".

Line 299: Please substitute "*points*" with "*pairs*", and "*sampled of*" with "*pertaining to*".

Line 300: Please add "*isotopic*" before "*depletion*".

Line 304: Please modify "*characteristic for*" with "*characteristic of*".

Line 307: Please change "*meteoric flow*" with "*meteoric fluid circulation*".

Line 308: Please add "*to the*" before "*development*".

Line 315: Please add a hyphens as indicated: "*low-to-moderate matrix....*".

Lines 316-318: I did some changes to this sentence: "*Even if Barremian limestones of La Fare anticline show porosity > 10%, it is mainly intra-granular, thus limiting flow pathways*".

Line 318: Please change "*Resulting from this event,...*" with "*Due to this characteristic,...*".

Line 320: Please add a hyphen between "*Fault and related*" "*Fault-related*".

Line 324: Better that "*impacting*" maybe you should try with "*affecting*".

Line 327: Please erase "*, and*" just after the references in parenthesis and also add a hyphen between "*low confining pressure*" "*low-confining pressure*".

Line 327: Add a space between "*<100KPa*" to have "*<100 KPa*".

Line 327: Modify the reference to "*Alikarami & Torabi 2015*" as "*Alikarami and Torabi, 2015*".

Lines 329-330: Here I fixed in this way: "*...of deformation band development  (Heiland et al., 2001; Lothe et al., 2002), enhancing fluid flow.*".

Line 331: "poorly" sounds better than "*dimly*".

Lines 331-332: Add a hyphen between "*low confining pressure*" "*low-confining pressure*".

Line 332: Please change the term "*pattern*" with "*regime*".

Line 332: Add a space between "*<1Km*" to have "*<1 Km*".

Lines 333-334: I reworked this sentence as indicated: "*Under these conditions, Barremian host rocks were likely characterised by mechanical and petrophysical properties close to porous granular media described above.*".

Line 334: Please correct "*showned*" with "*showed*".

Line 335: Please add "*of deformation*" after "*early stages*".

Line 336: Please add "*in carbonates*" after "*deformation bands*".

Line 336: Please change "*sector*" with "*area*".

Line 337: Please correct to the singular form "*circulation*" instead of "*circulations*".

Lines 337-338: Please erase "*These*" at the beginning of the sentence and make the singular of "*fluid flows*" "*fluid flow*".

Lines 338-339: I would change the sentence as follows: "*however, dilation bands were likely unstable and grain collapse occurred...*".

Line 340: Please modify to the singular form "*loading stress*" instead of "*loading stresses*".

Line 342: Here maybe start the sentence as indicated. "*This could be the explanation...*".

Line 343: Add a space between "*<30m*" to have "*<30 m*".

Line 345: Add a space between "*<188m*" to have "*<188 m*".

Line 345: Here adjust the reference to "*Fig. III 6A*" with "*Fig. 6A*".

Line 345: Modify the beginning of the sentence as suggested: "*Dilation bands have also been...*".

Line 346: Please correct "*Sicilly*" with "*Sicily*".

Line 347: Please add "*selective*" before "*cementation*". Also pluralise "*rock*" to "*rocks*".

Lines 348-349: I tried to change the structure of this sentence: "*Cementation (C1a and C1b) conferred a stiffer response of limestone to deformation, making it prone to deform through brittle structures (joints and veins), rather than via granular particulate flow (deformation bands)*".

Line 349: Add a hyphen between "*low porosity*" "*low-porosity*".

Line 350: Please change the last part of the sentence as suggested: "*...is known to increase fault permeability*".

Line 351: Please erase "*an*" before "*Al-rich*".

Line 352: Please pluralise "*fluid*" to "*fluids*".

Line 352: You may explain this with "*fine-grained*" instead of "*micro-metric*".

Line 352: Again the term "*barren*" is very unfamiliar to me. Are you referring to an incipient open fracture? If so you can state "*early-stage fractures*" or "*incipient open fracture*".

Line 352: If you feel this is an option try to put this last part of the sentence as indicated: "*...fractures, leading to precipitation of C2 cement*".

Line 358: Use "*may*" and not "*must*" you are not 100% sure that this happened, you are doing discussions constrained with the petrographic observations.

Line 361: A few adjustments to this sentence: "*As the fault grew, new fracture sets formed, leading to a new phase of calcite...*".

Lines 364-365: Please correct "*at high depth*" with "*in deep burial conditions*". in parenthesis correct from "*depth of maximum 500m; Fig. 9C4*" to "*maximum depth of 500 m; Fig. 9C4*". Be careful to the space between "*500 and m*".

Lines 365-366: A bit of reworking on this sentence: "*...corroborate the hypothesis of cementation acted by meteoric fluids rather than marine ones*".

Line 367: Please change the beginning of the sentence from "*Resulting from*" to "*Due to*".

Line 368: Before "*down to <5%*" insert "*with porosity*".

Line 368: Cancel the space in "*Fig. 9 B5*" to "*Fig. 9B5*". Also in Fig. 9 what is stage 5 since it not reported anywhere? Are you referring to the time interval existing between stages 4 and 6?

Line 371: Please change "*Implicitly*", with "*Following this*".

Line 371: Please add "*in this stage*" before "*was a barrier*".

Line 372: I would change the beginning of the sentence: "*Fluids responsible for precipitation of C3 cement...*".

Line 375: Please insert a space between "*100*" and "*µm*".

Line 376: I modified the sentence as indicated: *"...came from silica found inside C2 cement described above..."*.

Line 377: Please modify the beginning of the sentence: "*Silica crystals in C2 veins...*".

Line 377: Please insert a space between "*100*" and "*µm*".

Line 378: Please add "*grains*" before "*quartz*".

Line 379: Please add "*also*" before "*Aptian rocks*".

Line 380: Please substitute "*Implicitly*" with "*According to this,*".

Line 381: Add a hyphen between "*Uncemented*" "*Un-cemented*".

Line 381: Please change to the pas simple form "*formed*" instead of "*form*".

Line 383: Please add "*lateral extent of the...*" before "*drainage area*".

Line 384: Add a hyphen between "*high permeable*" "*high-permeable*".

Lines 385-386: I would change as follows: "*...formations within fault core of strike-slip and extensional faults (Billi et al., 2003, 2008; Storti et al., 2003)*".

Line 390: Please make the past simple form of "*lead*" to "*led*".

Line 392: Please substitute "*can be*" with "*could have been*".

Line 395: Please add a reference at the end of this sentence, relative to the difference in grain surface area from the clasts to matrix.

Line 398: I would change a bit the end of the sentence as indicated: "*...not favourable for extensive dolomitization outside fault zones*".

Line 399: Add a hyphen between "*Low temperature*" "*Low-temperature*".

Line 400: Add a hyphen between "*high temperature*" "*high-temperature*".

Line 402: Here you can change from "*high Mg fluid circulation*" to "*Mg-rich fluid circulation*".

Line 403: Please add "*domain*" after "*fault core*" at the end of the sentence.

Line 405: Please insert a space between "*23*" and "*Km*".

Lines 405-406: Please change from "*compressive conditions*" to "*contractional stress regime*".

Line 406: At the beginning of the sentence please change from "*From these authors...*" to "*according to these authors...*".

Line 407: Add a hyphen between "*low temperature*" "*low-temperature*".

Line 407: After "*upwelling fluids*" add "*, likely Mg-enriched*".

Line 409: Add a hyphen between "*low temperature*" "*low-temperature*".

Line 410: Please correct "*were*" with "*was*".

Line 412: Please correct "*reduce*" with "*reduces*".

Line 415: Please change "*to*" with "*into*". Also check the reference to "*Fig. 9 B6 and C6*" there is a space you should erase "*Fig. 9B6 and C6*".

Line 417: Please substitute "*finally*" with "*eventually*".

Line 419: Please erase the hyphen between "*back-ground*" to "*background*".

Line 420: Please change from "*lead to FR2...*" to "*formed FR2...*".

Line 422: Please erase the hyphen between "*back-ground*" to "*background*".

Lines 423-424: I would modify the structure of the sentence as indicated: "*This fluid flow is witnessed by the cementation...*".

Line 424: Please correct "*micritized*" with "*micritised*".

Line 425: Please change "*what led*" with "*leading to*".

Line 425: Erase the comma and space in the reference to figure "*Fig. 9, B7 and C7*" to "*Fig. 9B7 and C7*".

Line 426: Please change "*the*" before "*fracture porosity*" with "*that.*"

Line 426: At the end of the sentence modify as follows: "*...permeability was still partially preserved*".

Line 427: Use italics for the term "*sensu*".

Line 428: I would modify this as indicated: "*...and high fracture-related secondary permeability*".

Line 429: Maybe here is better to use "*infillings*" rather than "*fill*".

Line 429: After "*dissolution/cementation*" add the term "*processes*".

Line 431: Please add "*cement*" after "*C4*".

Line 432: Maybe here is better to use "i*nfillings*" rather than "*fill*".

Line 433: I did a few changes to this part of the sentence: "*...fluid circulation in the vadose zone, with alternating*".

Line 435: Maybe the last part of this sentence would sound better as: "*... of meteoric and marine origin with different burial depth*". At the end add also a reference to Fig. 6 where you reported the isotopic data.

Lines 436-437: Please change "*these*" with "*this*" and also make the singular form of "*circulations*" to "*circulation*".

Line 437: Please add "*cement*" after "*C4*".

Line 437: Please change "*on*" with "*in*" in reference to Fig. 8.

Line 438: "*Higher*" is not the most precise term to describe isotopic data. if you are talking about values more negative you should use the term "*depleted*", while less negative or positive values are usually addressed with the word "*enriched*". Try to do the same in all the text.

Line 439: Please add "*cement*" after "*C1*".

Line 440: Please add "*cement*" after "*C4*".

Line 440: Move "*C4*" before "*cement*", and change from the plural to the singular form "*cements*" to "*cement*".

Line 441: Please change "*are*" with "*is*".

Lines 441-442: I tried to fix this sentence in this way: "*Transect 2 cross-cuts the Castellas fault along a relay zone (Fig. 2A)*".

Line 442: Here, I would modify "*2*" with "*two*".

Line 446: Please erase the term "*local*" before "*permeability*" and also change "*allowed*" to "*favoured*".

Line 447: Please make the singular of "*circulations*" to "*circulation*".

Line 448: Move "*C4*" before "*cement.*

Line 449: Please change "*Contrarily*" with "*On the contrary,*".

Line 450: Please erase the word "*a*" before "*drains*".

Line 451: Please change "*That*" with "*This*".

Line 452: Please correct "*formation*" with "*development*".

Line 453: Please change "*normal*" with "*extensional*".

Line 453: I modified the last part of the sentence as follows: "*...C4 fluids to flow through the fault zone*".

Line 455: Please change "*T2*" with "*transect 2*".

Line 455: Please insert a space between "*60*" and "*m*".

Line 456: Use "*transect 1 and transect 3*" instead of "*T1 and T3*", add a space between "*30 and 40*" and "*m*", change also the symbol for "about" using this one "*~*".

Line 461: I don't think that the word "*sieve*" is appropriate to describe the evolution of the hydraulic properties of a fault zone. Maybe "*valve*" is more suitable, since you are describing media behaving as a drain and then as a barrier.

Line 462: Please correct "*de-dolomitization*" with "*de-dolomitized*".

Line 466: Please correct "*recrystallized*" with "*recrystallised*".

Line 469: I would put also "*alpine*" in capitals as you did for "*Pyrenean*".

Line 471: I tried to fix this part as indicated: "*This implies fluids percolating soils, as results from...*".

Line 475: Please correct "*Finally*" with "*Eventually*".

Line 475: Please change from "*incurring*" to "*inducing*".

Line 476: Please change from "*triggered*" to "*produced*".

Line 477: Please change from "*flows*" to "*fluids*".

Lines 481-482: I tried to adjust this part as: "*...reservoir where fractures behave as pathways towards fluid flow, but the production comes mainly from the matrix...*".

Line 483: Please change "*polyphase*" with "*polyphasic*".

Lines 490-491: See if these changes suit you: "*...Castellas fault zone, permeability evolves from a stage with exclusive contribution from the host rock...*".

Line 493: Here, I would modify "*2*" with "*two*".

Line 493: Please change "*fracture*" with "*fracturing*".

Line 493: Please change "*link*" with "*linked*".

Line 494: If you feel it could be an option you can use the extended form "*fault core*" instead of "*FC*".

Lines 495-496: I adjusted the second part of the sentence: "*permeability contribution is solely provided by the fault core...*".

Line 497: Again here if you can use the extended version of the name "*fault core*".

Line 498: Please correct "*at 20%*" with "*for 20%*".

Lines 488-499: The calculation of the permeability contribution is nice and to me it provides useful info relative to the hydraulic evolution of the fault zone in time. I'm sorry for being so blunt here, but maybe you should ground you statement and discussion on the field data. What I mean is try to explain why you assigned such percentage contribution to the fault core or to the matrix and so on... Maybe you can do this by evaluating the width of the fault core and damage zone domains, or by estimating the fracture network volume.

Line 501: Here, I would modify "*2*" with "*two*".

Line 501: Erase "*the*" before "*reservoir*".

Line 502: Please pluralise "*fault*" to "*faults*".

Line 503: I think you should capitalise "*SE basin"* as "*SE Basin*".

Line 508: Please correct "*their*" with "*its*".

Line 511: Please change "*thinly*" with "*slightly*".

Line 512: Please substitute "*formation*" with "*development*".

Line 514: Please change "*the flows*" with "*flowing fluids*".

Lines 516-517: Add a hyphen between "*low temperature*" "*low-temperature*".

Line 517: Please correct "*flows*" with "*fluids*".

Line 517: Please correct "*This*" with "*These*".

Line 517: Add a hyphen between "*low temperature*" "*low-temperature*".

Line 517: Please add "*fluid*" before "*flows*".

Line 519: Add a hyphen between "*high temperature*" "*high-temperature*".

Line 519: Please change "*flows*" with "*fluids*".

Line 519: Please add "*significant*" before "*hydrothermal influence*".

Line 520: I tried to improve the last part as: "*...broader rules for complex faults with polyphasic activity affecting granular carbonates at shallow burial conditions (Fig. 9)*".

Line 522: Please correct "*extensive*" with "*extensional*".

Line 522: See if this is better: "*...can lead to the development of dilation bands acting...*".

Line 523: I tried to improve the clarity: "*Carbonates are very sensitive to rock-fluid interactions*".

Lines 529-531: Again a bit of reworking: "*Late-stage fluids flowed preferentially within the permeable breccia rather than in the highly fractured damage zone*".

Line 566: Check and erase the highlighted space.

Line 568: There are too much spaces that must be corrected.

Line 569: Check and erase the highlighted space.

Line 570: Check and erase the highlighted space.

Line 571: Erase the highlighted full stops.

Line 573: Erase the highlighted full stop.

Line 580: Erase the highlighted full stop.

Line 586: Check and erase the highlighted space.

Line 590: Check and erase the highlighted space.

Line 607: Please erase the comma.

Line 610: Check and erase the highlighted space.

Line 611: The name of the institution is not complete.

Line 621: Check and erase the highlighted space.

Line 621: Please erase the comma.

Line 639: Please erase the comma.

Line 641: Erase the highlighted part since it is a repetition and should not be in front of the reference title.

Line 656: Please capitalise "*jurassic*" to "*Jurassic*".

Line 664: Erase the highlighted full stop.

Lines 669-670: Check and erase the highlighted spaces.

Line 679: Check and erase the highlighted space.

Line 705: Check and erase the highlighted spaces.

Line 714: Check and erase the highlighted spaces and comma.

Line 721: Check the spelling of the journal title.

Line 728: Check and erase the highlighted full stops.

Lines 741-742: Please eliminate the duplicated title.

Lines 754-755: Check and erase the highlighted full stops.

Lines 772-773: Check and erase the highlighted spaces.

Lines 774-775: Check and erase the highlighted space and full stop.

Line 791: Check and erase the highlighted space.

Line 803: Check and erase the highlighted space.

Line 809: Check and erase the highlighted space.

Line 814: Check and erase the highlighted space and comma.

Lines 816-817: Check and erase the highlighted space and comma.

Line 819: Check and erase the highlighted comma.

Line 821: Check and erase the highlighted space.

**Comments on figures and figure captions**

Comments concerning figures have been reported also in the pdf file with the exact position as sticky notes.

**Fig. 1**

You should insert the symbol of the La Fare anticline in Fig. 1A.

The kinematic indicators alongside faults are missing in Fig. 1B.

These names in the legend of Fig. 1A should be all in capitals "Upper Cretaceous, Lower Cretaceous."

Maybe better than "thin calcarenite" you can use "fine-grained", if you are referring to the grain size.

**Fig. 1 caption**

You should erase the space between "*Figure 1 :*" to "*Figure 1:*".

Please add "*trace of*" before "*stratigraphic column*".

Please correct the reference to "C" part of the figure rather than "*B*".

**Fig. 2**

The kinematic indicators in both stereo-nets are indistinguishable from the fault traces.

I would mirror the transect 3 to have SSE on the left and NNW on the right side, just like the other images.

What are the red stars? Are they the positions of samples? If so you should mention them in the caption.

**Fig. 2 caption**

You should erase the space between "*Figure 2 :*" to "*Figure 2:*".

Please change the term "*localization*" with "*position*".

What are the "*red points*" you are referring in the stereo-nets? I can't seen anything but red fault traces there, where are the points you are describing?

In the third line please add "C: *Photos of transect 1 and 2.*"

In the third line please add "*D: Photomicrographs of carbonate host-rock facies...*".

In the fourth line add "*FR1 and FR2*" after "*fault rock 1 and 2*".

In the fifth line please add "G*: Photomicrographs of host-rock facies...*".

**Fig. 3**

Try to improve the visibility of the three petrographic images in Fig. 3C, change the brightness-contrast.

**Fig. 3 caption**

In the first line correct "*&*" with "*and*".

In the third line correct "*b&c*" with "*b and c*".

**Fig. 4 caption**

You should erase the space between "*Figure 4 :*" to "*Figure 4:*".

In the first line please pluralise "*white arrow*" to "*white arrows*".

In the first line add a space between "*MF1micrite*" to "*MF1 micrite*".

In the second line add a space between "*2.5m*" to "*2.5 m*".

In the second line add a space between "*MF1micrite*" to "*MF1 micrite*".

In the second line add a space between "*2m*" to "*2 m*".

In the second line please erase the "*C*" which is duplicated.

In the third line add a space between "188*m*" to "*188 m*".

In the third line add a space between "*95m*" to "*95 m*".

In the fourth line please change "*F.*" to "*F:*".

**Fig. 5 caption**

You should erase the space between "*Figure 5 :*" to "*Figure 5:*".

In the first line please change "*micritized*" with "*micritised*".

In the second line please change "*space*" with "*volume*".

In the second line please substitute "*a&b*" with "*a and b*".

In the second line please pluralize "*clast*" to "*clasts*".

In the second line please change "*micritized*" with "*micritised*".

In the third line I modified as follows: "*C: C3 veins, cements and intergranular volume in...*".

In the third line please substitute "*a&b*" with "*a and b*".

in the fourth line after "*replacive dolomite*" add "*(RD)*".

In the fifth line please correct "*quart*" with "*quartz*".

In the fifth line please substitute "*a&b*" with "*a and b*"; do it twice.

**Fig. 6**

To me it would be more logical to invert Fig. 6A and 6B, to show the reader first all bulk data from fault zones and then the details of each cement phase, veins...

In both graphs insert the X axis labels for every increment of 2 per mil (2, 4, 6...).

In the legend of Fig. 6A it is written "*Bulk rock*", I wonder if this is actually the undeformed host rock, if so please correct accordingly.

In Fig. 6C the title of the graph states "*Distance to Castellas Fault plane*", maybe "*Fault plane*" should be in lower case.

**Fig. 6 caption**

You should erase the space between "*Figure 6 :*" to "*Figure 6:*".

In the first line please correct the symbols you used for the delta notation, it should be "$\delta^{13}C, \delta^{18}O$".

In the first line you state again "*bulk rock*" why not "*host rock*"?

In the third line please correct the symbols you used for the delta notation, it should be "$\delta^{13}C, \delta^{18}O$".

**Fig. 7**

The three photomicrographs are too small to appreciate the details. You have plenty of space, just make them bigger.

**Fig. 7 caption**

In the second line I slightly modified as follows: "*... development (blue), cementation (orange) and fault zone activation events (red)*".

**Fig. 8**

Again, photomicrographs are quite small, but still the reader should be able to see everything. This is a very nicely done image!

In the legend please correct "*Micro-facies & cement types*" with "*Micro-facies and cement types*".

**Fig. 8 caption**

You should erase the space between "*Figure 8 :*" to "*Figure 8:*".

**Fig. 9**

It would be nice to have bigger sketches in Fig. 9A.
Also why stage 5 is not reported? In the text it is mentioned. You should consider to implement this in the figure as well.

**Fig. 9 caption**

You should erase the space between "*Figure 9 :*" to "*Figure 9:*".

In the second line I would modify "*2*" with "*two*".

In the second line please correct "*curved*" with "*curve*".

**Fig. 10**

Also here size matters! Please make these sketches bigger otherwise you will lose al lot of details.

**Fig. 10 caption**

You should erase the space between "*Figure 10 :*" to "*Figure 10:*".

In the third line please add spaces between "*1to 8correspond*" to "*1 to 8 correspond*".

At the end of the caption you should add also explanations of the symbols used: FZ, DZ, MF1, MF2, MF3, K...

**Table 1**

In the caption add a full stop at the end as highlighted.

In the table header increase the width to include entirely the words "*Fault zones*", check also the spelling because "*Fault*" is misspelled as "*Faut*".

Check also the French name "*Faille*" and correct it accordingly.

Capitalize "*pitch striation*" to "*Pitch striation*".

Add a space between the cardinal point and angular value every time has been highlighted. Do the same with length and measurement unit.

"*Non constant*" is not precise, I would use "*variable*".

**Table 2**

Please eliminate "*vs*" from the table header.

Check also the nomenclature of the transects to be the same to the symbols adopted in the text.

In the caption it is not clear what do you mean for "*bulk carbonates*", "*bulk measurements*".

Pay attention also to put the reference always to the singular form (es. micrite value, isotopic value and so on). I highlighted every time where changes are needed.

[revised manuscript text omitted]

We studied 2 faults pertaining to kilometric-scale fault system on the E-W-trending La Fare anticline near Marseille (Fig. 1A). The southern limb of this anticlinal dips 25° S, and is constituted by Upper Hauterivian, Lower Barremian and Santonian rocks (Fig. 1B). The Upper Barremian carbonates are composed, from bottom to top, of a 120m-thick calcarenite unit with cross-beddings, a 40m-thick massive coral-rich calcarenite unit, and a 10m-thick calcarenite unit (Masse, 1976; Matonti et al., 2012; Roche, 2008). Unconformable Santonian rocks are made of coarse rudist limestones (Fig. 1A).

The Castellas fault zone is a kilometre-long strike-slip fault, N060 to 070-trending and 40° to 80°N-dipping (Fig. 2A, 2B; table 1) composed horse structures, secondary faults and lenses (Fig. 2A, 2C; Aubert et al. (2019b)). The second fault zone "D19" is composed of 5 sub-fault zones (F1 to F5) restricted in a 50m-long interval (Fig. 2E, H; Table 1; (Aubert et al., 2019a)). Sub-faults are made of 2 sets. Set one comprises F3 and F4, N040 to N055-trending, 60-80°NW-dipping (orange on Fig. 2F). Set 2 is N030-trending, dipping 80°E, with strike-slip slickensides pitch 20 to 28°SW (F1, F2, F5, red on Fig. 2F). The 5 sub-fault zones show an asymmetric architecture (Aubert et al., 2019a).

The internal structure of both fault zones results from three tectonic events:
- the Durancian uplift dated as mid-Cretoceous leading to extension and to normal *en echelon* normal faults. The Castellas fault is one of them and bear early dip-slip normal striations (Matonti et al., 2012),
- the Early Pyrenean compression with N000° to N170°-trending $\sigma_H$ (see cited references in Espurt et al. 2012). This event reactivates the Castellas fault as sinistral (Matonti et al., 2012) and leads to the neo-formed strike-slip faults of the D19 outcrop (Aubert et al., 2019a).
- the Pyrenean to Alpine folding, triggering the 25°S tilting of the strata and fault zones. Faults of the D19 outcrop were reactivated while the Castellas fault tilting led to an apparent reverse throw (Aubert et al., 2019a).

[Figure]

*Figure 1 : Geological context of the study area. A: geological map of Provence, B: Simplified structural map with the location of the Castellas fault and the stratigraphic column (black dashed line); B: Stratigraphic column of exposed Cretaceous carbonates (modified from Roche, 2008)*

[Figure]

**Figure 2 :** *A: Castellas fault map on aerial photo with localization of the studied transects and the relay zone; B: stereographic projections of poles to fractures (density contoured) and faults (red points)* (Allmendinger et al., 2013; Cardozo and Allmendinger, 2013)*; C: Photos of transects; D: Carbonate host-rock facies (a) transect 1 coral rich unit, (b) transect 2 calcarenites, (c) transect 3 calcarenites and (d) fault rocks 1 and 2; E: Pictures of D19 outcrop F: Stereographic projections of poles to fractures (density contoured), set one faults (orange) and set 2 faults (red) G: Host rock facies (a) and of fault rocks (b); H: D19 outcrop including the five faults F1 to F5.*

*Table 1*: structural properties of the fault zones

| Fault zones | Faille | Direction | Dip | Dip direction | pitch striation | Fault core thickness | Fault Rocks | | |
|---|---|---|---|---|---|---|---|---|---|
| | | | | | | | FR1 | FR2 | FR3 |
| Castellas | Castellas | 060 - 070 | 40 to 80 | N | 14W- | 0 to 4m | sparsely present | majoritarely present | / |
| D19 | F1 | 030 | 56 | W | | 20 | / | <10 cm | / |
| | F2 | 029 | 70 | E | 28 S | 10 to 15 | / | ? | non constant thickness |
| | F3 | 056 | 80 | N | | 0 to 15 | / | ? | ? |
| | F4 | 042 | 70 | W | | 20 | / | in the clasts of FR3 | non constant thickness |
| | F5 | 032 | 85 | N | 20 SW | 50 to 100 | / | / | non constant thickness |

**3. DATA BASE**

We performed 4 transects (T1 to T4) across the Castellas Fault and the D19 Fault (Fig. 2). Transect T1 is located along the coral rich unit 2. This bed is essentially composed of pelloidal grains and bioclasts (corals, bivalves and stromatoporidae; Fig. 2D a). Transects T2 and T3 are in unit 3, made of fine calcarenites with pelloidal grains and a rich fauna (foraminifera, bivalves, ostracods and echinoderm; Fig. 2Db, c). Transect 4 was conducted along the D19 outcrop (Fig. 3), which exposes Barremian outer platform bioclastic calcarenite with current ripples. The grains are mainly peloids with minor amount of bioclasts (solidary corals, bryozoan, bivalves and some rare miliolids; Fig. 2G, a).

The different tectonic events impacted the fault zone and fault core structure. Both faults have different fault cores (Table 1) made of 3 fault rock types in Castellas (Matonti et al., 2012) and D19 fault zones (see Aubert et al. 2019a). Fault rock 1 (FR1) results from the normal activation of the Castellas fault during Durancian uplift. It is a cohesive breccia composed of sub-rounded to rounded clasts from the nearby damage zone and <30% of grey matrix (Fig. 2Dd). Fault rock 2 (FR2), is linked to the sinistral reactivation of the Castellas fault and the onset of D19 fault zone during the Pyrenean shortening. FR2 present 2 morphologies depending on the fault zones. Within Castellas fault, FR2 is an un-cohesive breccia with an orange/oxidized matrix with angular to sub-rounded clasts from the damage zone and from FR1 (Fig. 2Dd). In the D19 fault zone, FR2 is a cohesive breccia with rounded clasts of the damage zone and a white cemented matrix (Fig. 2Gb). Fault rock 3 (FR3) is formed by the reactivation of D19 fault zone. It is composed of angular to sub-angular clast from FR2 and from the nearby damage zone in an orange/oxidized matrix (<20%) (Fig. 2Gb).

**4. METHODS**

The data set comprises 122 samples, 62 from Castellas and 60 from D19 outcrops, collected along the 4 transects. Porosity values were measured on 92 dry plugs with a Micromeritics AccuPyc 1330 helium pycnometer. Microfacies and petrography were determined on 92 thin sections. Impregnation with a blue-epoxy resin allowed us to decipher the different pore types. Thin sections were coloured with Alizarin red S and potassium ferricyanide to distinguish carbonate minerals (calcite and dolomite). The thin sections were analyzed using cathodoluminescence to discriminate the different calcite cements. The paragenetic sequence was defined based on superposition and overlap principles observed on thin sections using a Technosyn Cold Cathode Luminescence Model 8200 Mk II coupled to an Olympus BH2 microscope and to a Zeiss MR C5. Micrite micro-fabric and major element composition of 2

142 samples from the fault zone, 2 from the host rock and 1 from the D19 karst infilling were
143 measured using PHILIPS XL30 ESEM with a current set at 20kV on fresh sample surface and
144 on thin sections. To determine stable carbon and oxygen isotopes ($\delta^{13}$C and $\delta^{18}$O), 204
145 microsamples (<5 mg) were drilled, 194 of them were micro-drilled from polished thin sections
146 with an 80µm diameter micro-sampler (Merkantec Micromill) at the VU University
147 (Amsterdam, The Netherlands). We micro-sampled bulk rocks (57), sparitic cements (101),
148 fault rocks (9) and micrite (27). Carbon and oxygen values were acquired with Thermo Finnigan
149 Delta + mass spectrometer equipped with a GASBENCH preparation device at VU University
150 Amsterdam. The internationally used standard IAEA-603, with official values of +2.46‰ for
151 $\delta^{13}$C and -2.37‰ for $\delta^{18}$O, is measured as a control standard. The SD of the measurements is
152 respectively < 0.1‰ and < 0.2 ‰ for $\delta^{13}$C and $\delta^{18}$O. Ten whole rock samples were analysed
153 using a Gasbench II connected to a Thermo Fisher Delta V Plus mass spectrometer at the FAU
154 University (Erlangen, Germany). Measurements were calibrated by assigning $\delta^{13}$C values of
155 +1.95‰ to NBS19 and -47.3‰ to IAEA-CO9 and $\delta^{18}$O values of -2.20‰ to NBS19. All values
156 are reported in per mil relative to V-PDB.

157 **5. RESULTS**

158   1.   **MICROPOROSITY AND POROSITY**
159 Porosity measurements performed on the 92 samples show that in average, porosity strongly
160 decreases towards the fault core (Fig. 3): from more than 10% (mean: 15%, SD: 2.68 for

[Figure]

**Figure 3**: A: Castellas fault zone aerial view (Ortho13, 2009, CRIGE-PACA, logo FEDER) & porosity values measured along transect 1 (Red Cross), transect 2 (green cross) and transect 3 (black cross); B: porosity values measured along D19 fault zone; C: Pore types in the host rock (a) and in the fault zones (b&c).

Castellas and mean 12.3%, SD: 2.52 for D19) to less than 5% in fault zones (mean: 4.8%, SD: 2.07 for Castellas and mean: 3.16%, SD: 2.35 for D19).

Some variations occur as follows:

- North of the Castellas fault, along the 60m-long transect T2 the porosity is constantly low < 7% (mean of 4.4%, SD:1.53 ; Fig. 3A),
- South of the Castellas fault, the reduced porosity zone is >40m in transect 3 and 30m in transect 1 (Fig. 3A). In a 10m-thick zone from the fault plane, porosity reduction occurs with lower values in T1 (average 4.9%) than in T3 (average 5.6%).
- In the D19 fault zone, the lowest porosity values are in narrow zones around the faults (less than 2m) and in the lens between F4 and F5. Though, this porosity decrease is not homogeneous in fault zone and high values are found north of F1 and F3 (Fig. 3B).

From thin sections impregnated with blue-epoxy resin, a porous rock-type with $\phi$>10% mainly in micritized grains as microporosity and moldic porosity (Fig. 3C a), and a tight rock-type with $\phi$< 5% where the porosity is mostly linked to barren styloliths (Fig. 3C b, c) are distinguished.

**2. DIAGENETIC PHASES**

**a. Micrite micro-fabric**

Micritized bioclasts, ooids and peloids were observed after SEM analysed of 2 fault zones samples and 2 host rock samples. Two micro-fabrics of micrite occur with specific crystal shape, sorting and contacts according to *Fournier et al. (2011).* Within both fault zones, the micrite is tight, with compact subhedral mosaic crystals (MF1; Fig. 4A, 4B). In the host rock, the micrite is loosely packed, and partially coalescent with puntic rarely serrate, subhedral to euhedral crystals (MF3; Fig. 4C, D, E). MF1 correlates with low porosity values < 5% , while MF3 with higher porosity > 10%.

**b. Diagenetic cements**

Eight cement stages were identified (Fig. 5). The red stain links to Alizarin red S coloration shows that all visible cements made up of calcite, which exhibits variable characteristics (morphology, luminescence, size and location).

The first 2 cement phases occur in both fault zones. The first cement (C0) is non-luminescent isopachous calcite of constant thickness ($\approx$10µm) around grains (Fig. 5A). The second cement (C1) is divided in 2 sub-phases: a non-luminescent calcite, C1a, with a dog tooth morphology in intergranular spaces, and a bright luminescence calcite, C1b, covering C1a with a maximum thickness of $\approx$100µm (Fig. 5). C1b also fills micro-porosity in micritised grains (Fig. 5B). C1b values strongly increase in Castellas fault zone.

[Figure]

***Figure 4 :*** *MEB pictures of micrite micro-fabric and microporosity (white arrow); A. MF1micrite micro-fabric in Castellas fault zone (2.5m to fault plane); B: MF1micrite micro-fabric within D19 fault zones (2m away from F5 fault plane); C C: MF3 micrite micro-fabric within Castellas host rock (188m away from the fault plane); D: MF3 micrite micro-fabric within D19 host rock (95m away from F5 fault plane); E: D19 host rock moldic porosity; F. Karst infilling.*

201 Five cements or replacive phases occur largely in the Castellas sector and rarely in the D19
202 outcrop:

203  - C2 is a sparitic cement with dull orange luminescence only found in fault core veins
204   (Fig. 5B). SEM measurements show the Si and Al elements in the C2 veins. Most of Si
205   crystals are an automorphic.
206  - C3 is a blocky calcite with non to red dull luminescence in veins, moldic and
207   intergranular pores (Fig. 5B, C, D). This cement also occurs in few veins of D19 sectors
208   but is not restricted to the fault zone.

[Figure]

***Figure 5***: *Thin-sections under cathodoluminescence; A: Calcarenite in transect 3 with micritized grain (M1), and intergranular space cemented with C1 a&b and C3; B: C2 (with Si) and C3 veins affecting Castellas FR1 clast with micritized grains cemented by C1b; C: C3 vein cement and intergranular space in Castellas fault zone; D: C1 (a & b) and C3 cementing moldic porosity of transect 3 calcarenite; E: FR1 matrix with phantom of cloudy appearance replacive dolomite; F: FR1 matrix de-dolomitized by C5 containing quart grains; G: C4 (a & b) cementing vein of D19 fault zone; H: matrix of D19 FR2 cemented by C4 (a&b).*

210  - Phantoms of planar-e (euhedral) dolomite crystals (Sibley and Gregg, 1987) with a
211    maximum size of 500μm affect the matrix of FR1 (Fig. 5E). They are vestiges of a
212    dolomitization phase. They have a cloudy appearance caused by solid micritic inclusion
213    in the crystal and can be considered as replacive dolomite (RD; Machel, 2004). Within
214    the FR1 matrix, an important concentration of angular grains of quartz with a maximum
215    size of 300μm is noticed (Fig. 5F).
216  - A blocky calcite C4 (referred to as S2 in Aubert et al. (2019a)) is mainly present in veins
217    of the D19 outcrop, in matrix of FRA, and intergranular and moldic pores (Fig. 5G, 5H).
218    This cement shows zonation of non-luminescent and bright luminescent bands and can
219    be divided in 2 sparitic sub-phases: C4a which is non-luminescent with some highly
220    luminescent band and C4b which is bright luminescent with some non-luminescent
221    bands. C4a occurs in lesser proportion in some veins along transect T2 and T3 of the
222    Castellas fault.
223  - A sparitic cement C5, with a red dull luminescence replaces the RD phase (Fig. 5F).

**c.  Additional diagenetic features**

In addition to cementation phases, other diagenetic elements affected both fault zones. Karst infilling occurs in the F2 fault zone of the D19 outcrop. It is composed of well-sorted grains deposited in laminated layers. This formation presents a stack of micrite-rich layers and grain-rich layers. In which grains are intergranular sparitic clasts, remaining from blocky calcite of dissolved grainstones and oxydes. The laminated layers are affected by veins and stylolites; some of these are deformed due to the clasts fall on sediments. Micritic layers has been observed under SEM, the micrite appeared tight with compact subhedral mosaic crystals (Fig. 4F). We observed oxide filling mainly in the Castellas area in dissolution voids affecting C1a, C1b and C3 cementation phases and in D19 in karstic fill. The proportion of oxides increase close to stylolites.

**3.  CARBON AND OXYGEN ISOTOPES**

Isotope measurements were realized on samples collected along transects of the fault zones. A hundred and eighty-nine measurements of C and O isotopes were performed on 16 samples and 32 thin sections (Fig. 6A, table 2).

Sampling was done in bulk rock (66), sparitic cement (101; veins, intergranular spaces and fault rock cements) and in fault rocks (10) in order to determine their isotopic signature. Isotopic values range from -10.40‰ to -3.65‰ for $\delta^{18}O$ and from -7.20‰ to +1.42‰ for $\delta^{13}C$ (Fig. 6A, 6B, table 2). The bulk rock values range from -9.18‰ to -4.34‰ for $\delta^{18}O$ and from -4.80‰ to +1.19‰ for $\delta^{13}C$ (Fig. 6A, table 2). These values are split in 2 sets. Set one includes transect 1 & 3 of the Castellas Fault. Bulk values range from -6.07‰ to -4.34‰ for $\delta^{18}O$ and from -1.41‰ to +1.19‰ for $\delta^{13}C$. Set 2 includes transect 2 (Castellas) and transect 4 (D19). Bulk values range from -9.18‰ to -5.20‰ for $\delta^{18}O$ and from -4.80‰ to -0.60‰ for $\delta^{13}C$ (Fig. 6B, table 2). In the transect 3, the isotopic values only slightly vary along transect, ranging from -6.13‰ to -4.50‰ for $\delta^{18}O$ and from -1.41‰ to +0.47‰ for $\delta^{13}C$ (Fig. 6C, table 2). Contrarily, values vary more along the D19 transect. They range from -9.18‰ to -5.20‰ for $\delta^{18}O$ and from -4.80‰ to -0.60‰ for $\delta^{13}C$ (Fig. 6C, table 2). Indeed, the $\delta^{13}C$ values obviously decrease in the fault vicinity, especially south of F2.

252 Isotopic values of cements filling veins, intergranular spaces, karst fills, and fault rock are
253 divided into 5 groups (Fig. 6A, table 2):

- the group of values from C1 fluctuates from -6.76‰ to -4.45‰ for $\delta^{18}O$ and from -1.28
  to +1.08‰ for $\delta^{13}C$;
- the group of values from C3 ranges from -10.40‰ to -6.73‰ for $\delta^{18}O$ and from -2.09
  to +1.22‰ for $\delta^{13}C$;
- the group of values of C4 in FR1 and FR2 matrix and in karst fill ranges from -9.18‰
  to -4.60‰ for $\delta^{18}O$ and from-5.10‰ to -0.74‰ for $\delta^{13}C$ with a positive covariance
  between $\delta^{18}O$ and $\delta^{13}C$. FR 2 matrix values (from-6.55 to -7.06‰ for $\delta^{18}O$ and from -
  1.10 to -2.24‰ for $\delta^{13}C$) present slightly less negative values than karst fill with mean
  values of -7.83‰ and -2.53‰ respectively for $\delta^{18}O$ and $\delta^{13}C$. (Fig. 6A). In the Castellas
  fault, 4 isotopic values from 2 veins are high with means of -6.25 and -4.2‰ for $\delta^{18}O$ -
  0.64 and -0.09‰ for $\delta^{13}C$ having similar positive covariance than the other C4 values.
- the group of values from C5, sampled in FR1 matrix with a mean of -7.49‰ for $\delta^{18}O$
  and -4.01‰ for $\delta^{13}C$ (Fig. 6A).
- the group of values from FR3 matrix with a mean of -5.98‰ for $\delta^{18}O$ and -6.83‰ for
  $\delta^{13}C$ (Fig. 6A)

[Figure]

***Figure 6 :*** *Isotopic values of δ13C and δ18O measured on bulk rock, cement phases, and micrite. Range values of
"Urgonian marine box" from Moss & Tucker (1995) and Godet et al. (2006); A: set of values sorted by the nature of
diagenetic phases and B: values sorted by the fault zone; C: lateral evolution of δ13C and δ18O isotopic values in
Castellas (top) and in D19 (bottom) fault zones.*

*Table 2: Carbon and oxygen isotope values of bulk carbonates for Castellas fault zone and D19*

270 *fault zones. B: bulk measurements; M: micrite values; C1, C3, C4, C5: isotopic values of*

271

272 *cement C1, C3, C4 and C5; FR: fault rock isotopic values.*

| Transect | Sample | δ¹³C (‰ vs VPDB) | δ¹⁸O (‰ vs VPDB) | Class | Distance to F. (m) |
|---|---|---|---|---|---|
| Castellas (T 1) | 201 | 1,19 | -4,34 | B | 1,3 |
| Castellas (T 1) | 201 | 1,02 | -6,62 | C1 | 1,3 |
| Castellas (T 1) | 201 | 1,31 | -3,94 | M | 1,3 |
| Castellas (T 1) | 201 | 1,37 | -3,65 | M | 1,3 |
| Castellas (T 1) | 213 | -0,68 | -5,24 | B | 22,7 |
| Castellas (T 1) | 213 | -0,58 | -5,10 | B | 22,7 |
| Castellas (T 1) | 213 | -0,18 | -6,09 | C1 | 22,7 |
| Castellas (T 1) | 213 | 0,03 | -4,45 | C1 | 22,7 |
| Castellas (T 1) | 213 | 0,09 | -4,77 | C1 | 22,7 |
| Castellas (T 1) | 213 | -2,09 | -6,92 | C4 | 22,7 |
| Castellas (T 1) | 213 | -0,68 | -4,92 | M | 22,7 |
| Castellas (T 2) | c3b17 | -0,52 | -5,95 | B | 4,6 |
| Castellas (T 2) | c3b17 | -2,07 | -6,38 | C4 | 4,6 |
| Castellas (T 2) | c3b7 | -0,64 | -5,51 | B | 9,3 |
| Castellas (T 2) | c3b26 | -3,76 | -6,26 | B | 22,6 |
| Castellas (T 2) | c3b26 | -2,85 | -5,58 | C4 | 22,6 |
| Castellas (T 2) | c3b26 | -1,31 | -4,69 | B | 57,3 |
| Castellas (T 2) | c3b7 | -1,76 | -6,31 | C1 | 57,3 |
| Castellas (T 2) | c3b7 | -1,28 | -6,46 | C1 | 57,3 |
| Castellas (T 2) | c3b26 | -2,35 | -5,22 | M | 57,3 |
| Castellas (T 2) | c3b26 | -1,70 | -4,75 | M | 57,3 |
| Castellas (T 3) | 327 | -0,24 | -7,55 | C3 | 0,3 |
| Castellas (T 3) | 325 | -1,90 | -9,06 | C3 | 0,3 |
| Castellas (T 3) | 325 | -1,69 | -8,95 | C3 | 0,3 |
| Castellas (T 3) | 327 | -3,11 | -8,09 | C4 | 0,3 |
| Castellas (T 3) | 327 | 0,47 | -5,40 | B | 1,0 |
| Castellas (T 3) | 327 | -0,18 | -7,95 | C3 | 1,0 |
| Castellas (T 3) | 327 | -0,17 | -7,41 | C3 | 1,0 |
| Castellas (T 3) | 328 | 0,10 | -5,74 | C1 | 1,6 |
| Castellas (T 3) | 328 | -1,32 | -8,18 | C3 | 1,6 |
| Castellas (T 3) | 328 | -0,59 | -7,77 | C3 | 1,6 |
| Castellas (T 3) | 328 | -0,42 | -7,74 | C3 | 1,6 |
| Castellas (T 3) | 328 | -0,13 | -9,26 | C3 | 1,6 |
| Castellas (T 3) | 328 | 0,02 | -8,83 | C3 | 1,6 |
| Castellas (T 3) | 328 | 0,29 | -8,70 | C3 | 1,6 |
| Castellas (T 3) | 328 | 0,42 | -8,73 | C3 | 1,6 |
| Castellas (T 3) | 328 | 0,50 | -7,89 | C3 | 1,6 |
| Castellas (T 3) | 328 | 1,22 | -8,18 | C3 | 1,6 |
| Castellas (T 3) | 333 | -1,84 | -8,67 | C3 | 1,6 |

| | | | | | |
|---|---|---|---|---|---|
| Castellas (T 3) | 333 | -0,96 | -7,89 | C3 | 1,6 |
| Castellas (T 3) | 328 | -0,14 | -4,17 | C4 | 1,6 |
| Castellas (T 3) | 328 | -0,05 | -4,23 | C4 | 1,6 |
| Castellas (T 3) | 329 | 0,16 | -4,95 | B | 2,4 |
| Castellas (T 3) | 333 | -0,25 | -6,38 | C1 | 4,6 |
| Castellas (T 3) | 333 | -0,12 | -6,17 | C1 | 4,6 |
| Castellas (T 3) | 333 | -0,62 | -8,52 | C3 | 4,6 |
| Castellas (T 3) | 333 | -0,12 | -5,67 | M | 4,6 |
| Castellas (T 3) | 333 | -0,02 | -4,48 | M | 4,6 |
| Castellas (T 3) | 333 | 0,42 | -4,60 | M | 4,6 |
| Castellas (T 3) | 337 | 0,19 | -5,59 | B | 9,5 |
| Castellas (T 3) | 302 | -0,53 | -4,50 | B | 11,8 |
| Castellas (T 3) | 302 | -0,49 | -4,74 | B | 11,8 |
| Castellas (T 3) | 302 | -0,62 | -10,38 | C3 | 11,8 |
| Castellas (T 3) | 302 | -0,49 | -10,02 | C3 | 11,8 |
| Castellas (T 3) | 305 | 0,33 | -4,38 | B | 16,0 |
| Castellas (T 3) | 306 | 0,21 | -4,35 | B | 17,8 |
| Castellas (T 3) | 307 | -0,01 | -4,46 | B | 18,2 |
| Castellas (T 3) | 308 | -0,57 | -4,95 | B | 20,0 |
| Castellas (T 3) | 308 | -1,44 | -9,11 | C3 | 20,0 |
| Castellas (T 3) | 308 | -0,23 | -10,40 | C3 | 20,0 |
| Castellas (T 3) | 308 | -0,22 | -10,08 | C3 | 20,0 |
| Castellas (T 3) | 309 | -1,41 | -4,87 | B | 20,5 |
| Castellas (T 3) | 309 | -0,52 | -5,01 | B | 20,5 |
| Castellas (T 3) | 309 | -0,15 | -4,82 | C1 | 20,5 |
| Castellas (T 3) | 309 | -1,56 | -7,96 | C3 | 20,5 |
| Castellas (T 3) | 309 | -1,55 | -8,01 | C3 | 20,5 |
| Castellas (T 3) | 312 | 0,12 | -4,81 | B | 23,2 |
| Castellas (T 3) | 314 | -0,71 | -5,30 | B | 25,9 |
| Castellas (T 3) | 314 | -0,80 | -10,09 | C3 | 25,9 |
| Castellas (T 3) | 314 | -0,49 | -9,90 | C3 | 25,9 |
| Castellas (T 3) | 314 | -0,47 | -10,29 | C3 | 25,9 |
| Castellas (T 3) | 314 | -0,40 | -9,97 | C3 | 25,9 |
| Castellas (T 3) | 314 | 0,06 | -10,30 | C3 | 25,9 |
| Castellas (T 3) | 316 | -1,24 | -5,50 | B | 29,2 |
| Castellas (T 3) | 316 | -1,00 | -5,48 | B | 29,2 |
| Castellas (T 3) | 316 | -0,22 | -4,79 | B | 29,2 |
| Castellas (T 3) | 316 | -1,02 | -10,21 | C3 | 29,2 |
| Castellas (T 3) | 316 | -0,18 | -9,31 | C3 | 29,2 |
| Castellas (T 3) | 316 | 0,30 | -10,37 | C3 | 29,2 |
| Castellas (T 3) | 318 | -0,28 | -4,53 | B | 35,4 |
| Castellas (T 3) | 320 | -0,68 | -5,79 | B | 96,1 |
| Castellas (T 3) | 322 | -0,88 | -6,07 | B | 158,0 |
| Castellas (T 3) | 323 | -0,65 | -5,37 | B | 188,0 |
| Castellas (ZF1) | Z1,1 | 0,17 | -5,26 | C1 | 0,0 |
| Castellas (ZF1) | Z1,1 | 0,39 | -5,23 | C1 | 0,0 |

| Castellas (ZF1) | Z1,1 | 0,46 | -4,70 | C1 | 0,0 |
| Castellas (ZF1) | Z1,2 | 0,21 | -5,98 | C1 | 0,0 |
| Castellas (ZF1) | Z1,1 | -0,55 | -6,40 | C4 | 0,0 |
| Castellas (ZF1) | Z1,1 | -0,52 | -6,10 | C4 | 0,0 |
| Castellas (ZF1) | Z1,2 | -4,12 | -7,45 | C5 | 0,0 |
| Castellas (ZF1) | Z1,2 | -0,15 | -4,99 | FR | 0,0 |
| Castellas (ZF1) | Z1,2 | 0,39 | -4,73 | M | 0,0 |
| Castellas (ZF1) | Z1,2 | 0,61 | -5,77 | M | 0,0 |
| Castellas (ZF1) | Z1,1 | 0,78 | -6,16 | M | 0,0 |
| Castellas (ZF2) | Z2,2 | 0,77 | -5,38 | C1 | 0,0 |
| Castellas (ZF2) | Z2,7 | -1,40 | -9,52 | C3 | 0,0 |
| Castellas (ZF2) | Z2,7 | -4,38 | -7,15 | C5 | 0,0 |
| Castellas (ZF2) | Z2,7 | -3,97 | -7,13 | C5 | 0,0 |
| Castellas (ZF2) | Z2,7 | -3,78 | -8,04 | C5 | 0,0 |
| Castellas (ZF2) | Z2,7 | -3,56 | -7,86 | C5 | 0,0 |
| Castellas (ZF2) | Z2,7 | -3,24 | -7,48 | C5 | 0,0 |
| Castellas (ZF2) | Z2,7 | -3,23 | -8,54 | C5 | 0,0 |
| Castellas (ZF2) | Z2,2 | 0,58 | -5,47 | FR | 0,0 |
| Castellas (ZF2) | Z2,2 | 0,92 | -4,91 | FR | 0,0 |
| Castellas (ZF2) | Z2,7 | -1,68 | -5,63 | FR | 0,0 |
| Castellas (ZF2) | Z2,7 | -2,24 | -6,55 | FR | 0,0 |
| Castellas (ZF2) | Z2,7 | -3,18 | -7,38 | M | 0,0 |
| Castellas (ZF2) | Z2,7 | -2,86 | -6,03 | FR | 1,0 |
| Castellas (ZF5) | Z5,4 | 0,27 | -8,25 | C3 | 0,0 |
| Castellas (ZF5) | Z5,4 | 0,31 | -7,87 | C3 | 0,0 |
| Castellas (ZF5) | Z5,4 | 0,32 | -8,23 | C3 | 0,0 |
| Castellas (ZF5) | Z5,4 | 1,06 | -6,34 | C1 | 0,4 |
| Castellas (ZF5) | Z5,4 | 1,08 | -6,76 | C1 | 0,4 |
| Castellas (ZF5) | Z5,4 | 1,05 | -7,13 | FR | 0,4 |
| Castellas (ZF5) | Z5,4 | 1,37 | -6,03 | FR | 0,4 |
| Castellas (ZF5) | Z5,4 | 1,42 | -6,15 | FR | 0,4 |

| Transect | Sample | $\delta^{13}C$ (‰ vs VPDB) | $\delta^{18}O$ (‰ vs VPDB) | Class | Distance to F1 (m) |
|---|---|---|---|---|---|
| D19 | 3B | -0,81 | -6,52 | B | 0,0 |
| D19 | 3B | -1,20 | -6,50 | C1 | 0,0 |
| D19 | 3B | -1,02 | -6,33 | C1 | 0,0 |
| D19 | 3B | 0,11 | -6,25 | C1 | 0,0 |
| D19 | 3B | -0,74 | -6,23 | M | 0,0 |
| D19 | 9 | -2,32 | -7,30 | B | 9,2 |
| D19 | 13a | -3,44 | -8,11 | B | 14,3 |
| D19 | 13a | -2,96 | -7,93 | B | 14,3 |
| D19 | 13C | -2,97 | -7,62 | M | 14,3 |
| D19 | 13C | -2,86 | -7,79 | M | 14,3 |
| D19 | 13C | -2,70 | -8,12 | M | 14,3 |
| D19 | 13C | -2,67 | -7,96 | M | 14,3 |

| D19 | 13C | -2,66 | -8,16 | M | 14,3 |
|-----|-----|-------|-------|-----|------|
| D19 | 13C | -2,50 | -7,77 | M | 14,3 |
| D19 | 13C | -1,54 | -8,98 | M | 14,3 |
| D19 | 17 | -2,58 | -7,68 | B | 18,7 |
| D19 | 14A | -1,97 | -6,38 | B | 18,7 |
| D19 | 14A | -1,87 | -6,74 | B | 18,7 |
| D19 | 15B | -2,23 | -7,43 | B | 18,7 |
| D19 | 17 | -1,05 | -6,40 | C1 | 18,7 |
| D19 | 14A | -1,77 | -6,74 | C1 | 18,7 |
| D19 | 14A | -2,42 | -6,43 | C4 | 18,7 |
| D19 | 14A | -2,06 | -6,67 | C4 | 18,7 |
| D19 | 21 | -2,23 | -6,54 | B | 24,4 |
| D19 | RSG | -1,90 | -7,66 | B | 28,4 |
| D19 | RSG | -1,70 | -7,83 | B | 28,4 |
| D19 | RSD | -2,87 | -7,10 | B | 29,5 |
| D19 | RSD | -2,76 | -7,14 | B | 29,5 |
| D19 | RSD | -0,93 | -9,40 | C3 | 29,5 |
| D19 | RSF1 | -2,40 | -7,28 | B | 34,7 |
| D19 | RSF2 | -2,14 | -7,39 | B | 34,7 |
| D19 | RSF2 | -1,78 | -7,27 | B | 34,7 |
| D19 | RSF1 | -1,03 | -9,44 | C3 | 34,7 |
| D19 | RSF2 | -1,93 | -8,05 | C3 | 34,7 |
| D19 | RSF2 | -0,59 | -9,40 | C3 | 34,7 |
| D19 | RSF2 | -2,95 | -8,14 | C4 | 34,7 |
| D19 | RSE 1 | -2,53 | -7,33 | B | 35,0 |
| D19 | RSE 2 | -2,59 | -7,41 | B | 35,0 |
| D19 | RSE 1 | -1,71 | -7,68 | C3 | 35,0 |
| D19 | RSE 2 | -1,84 | -6,73 | C3 | 35,0 |
| D19 | 57 | -2,07 | -5,93 | B | 38,1 |
| D19 | 57 | -1,94 | -5,87 | B | 38,1 |
| D19 | 57 | -1,83 | -7,06 | C3 | 38,1 |
| D19 | 57 | -1,10 | -6,75 | C3 | 38,1 |
| D19 | 57 | -4,02 | -7,04 | C4 | 38,1 |
| D19 | 57 | -2,17 | -5,72 | C4 | 38,1 |
| D19 | 57 | -1,58 | -6,52 | FR | 38,1 |
| D19 | 57 | -7,20 | -5,68 | M | 38,1 |
| D19 | 57 | -7,13 | -5,90 | M | 38,1 |
| D19 | 28b | -1,03 | -7,21 | B | 39,3 |
| D19 | 28b | -1,03 | -6,10 | C3 | 39,3 |
| D19 | 28b | -4,09 | -6,92 | C4 | 39,3 |
| D19 | 28b | -2,58 | -7,40 | C4 | 39,3 |
| D19 | 28b | -2,47 | -7,54 | C4 | 39,3 |
| D19 | 30a | -1,61 | -7,04 | B | 42,6 |
| D19 | 30a | -1,41 | -6,87 | B | 42,6 |
| D19 | 30a | -3,23 | -7,03 | C4 | 42,6 |
| D19 | 30a | -2,89 | -7,45 | C4 | 42,6 |

| | | | | | |
|------|------|-------|-------|----|-------|
| D19 | 24a | -1,21 | -7,52 | B | 51,1 |
| D19 | 27b | -1,92 | -7,48 | B | 57,9 |
| D19 | 31 | -1,24 | -6,44 | B | 65,0 |
| D19 | 32 | -1,75 | -7,50 | B | 67,4 |
| D19 | 34 | -1,79 | -7,49 | B | 72,2 |
| D19 | 36 | -1,32 | -7,21 | B | 77,8 |
| D19 | 38 | -1,73 | -7,59 | B | 81,5 |
| D19 | 62 | -1,96 | -7,56 | B | 86,0 |
| D19 | 42 | -0,81 | -6,80 | B | 91,9 |
| D19 | 63 | -0,55 | -5,50 | B | 124,0 |
| D19 | 64 | -1,17 | -5,88 | B | 160,0 |
| D19 | 65 | -1,10 | -6,57 | B | 197,0 |
| D19 | 66 | -1,31 | -5,21 | B | 236,0 |
| D19 | 60a | -3,06 | -9,18 | B | 255,2 |
| D19 | 60B | -4,80 | -8,47 | B | 255,2 |
| D19 | 60B | -4,66 | -8,92 | B | 255,2 |
| D19 | 61 | -1,53 | -9,87 | C3 | 255,2 |
| D19 | 61 | -1,36 | -9,89 | C3 | 255,2 |
| D19 | 60a | -1,15 | -9,70 | C3 | 255,2 |
| D19 | 60a | -3,32 | -9,11 | C4 | 255,2 |
| D19 | 60B | -5,10 | -9,09 | C4 | 255,2 |
| D19 | 60B | -4,73 | -8,84 | C4 | 255,2 |
| D19 | 60B | -4,15 | -9,18 | C4 | 255,2 |
| D19 | 60B | -4,07 | -9,16 | C4 | 255,2 |
| D19 | 60B | -2,90 | -9,06 | C4 | 255,2 |
| D19 | 60a | -3,83 | -7,85 | M | 255,2 |
| D19 | 60B | -5,04 | -9,17 | M | 255,2 |
| D19 | 60B | -4,25 | -8,14 | M | 255,2 |
| D19 | 60B | -3,61 | -8,58 | M | 255,2 |
| D19 | 60B | -3,61 | -8,13 | M | 255,2 |

**6. DISCUSSION**

**1. DIAGENETIC EVOLUTION OF THE FAULT ZONES**

The chronological relations between cements can be established thanks to cross-cutting relation and inclusion principles. Indeed, the veins filled with cement C2 cross-cut cements C1a and C1b (Fig. 5B). Thus, C2 cementation postponed C1 cementation. The C3 veins cross-cut the C2 veins, but are included within FR1 clasts (Fig. B). Hence, C3 cement is ante-FR1 development but post-C2 cementation. The fault rock 1 (FR1) is related to the first normal fault activity, consequently, C1, C2 and C3 cementation phases occurred prior to the proper fault plane and fault core formation and, are related to the fault nucleation. Replacive dolomite is within FR1 matrix (Fig. 3E), therefore, it develop after FR1 formation. Finally, the cement C4 can be noticed within FR2 matrix indicating that C4 cementation event postponed FR2 formation. The fault rock 2 (FR2) developed during the faults strike-slip reactivation. The combined superposition, overlap, cross-cutting principles and isotopic signature of cements brought out the chronology between phases, and revealed the paragenetic sequence (Fig. 7).

[Figure]

**Figure 7**: *Paragenetic sequence of the both fault zones (black: Castellas, grey: D19) with micro-porosity development (blue) and cementation (orange) and fault zone activation (red).*

The Urgonian carbonates in La Fare anticlinal underwent 3 important diagenetic events, which impacted the host rock and/or the fault zones. We discriminate among diagenetic events that occurred before and during faulting.

**a. Pre fault diagenesis – microporosity development**

During Upper Barremian, just after deposition, micro-bores organisms at the sediment-water interface enhanced the formation of micritic calcitic envelopes on bioclasts, ooids and peloids (Purser, 1980; Reid and Macintyre, 2000; Samankassou et al., 2005; Vincent et al., 2007). This micritisation in marine conditions is typical for Urgonian low energy inner platform (Fournier et al., 2011; Masse, 1976). Subsequently, cement C0 formed around grains and formed a solid shelve inducing the conservation of the clast shape during the later burial compaction (Step 0 on Fig. 8). However, the majority of isotopic values do not fit in the Barremian sea water calcite box which ranges from -1.00‰ to -4.00‰ for $\delta^{18}O$ and from +1.00‰ to +3.00‰ for $\delta^{13}C$ (Fouke et al., 1996; Godet et al., 2006). Only 2 data points sampled of micritised grains show isotopic values close the Barremian sea water calcite. The depletion of other data indicates the slight impact of C0 cementation on isotopic values.

The next sub-phase of cementation C1a partly fills intergranular porosity. This non luminescent cement with isotopic values ranging from -6.8‰ to -3.9‰ for $\delta^{18}O$ and from -1.0‰ to +1.3‰ for $\delta^{13}C$ is characteristic for mixed fluids. Léonide et al. (2014) measured a calcite cement S1, near La Fare anticline with similar luminescence and isotopic range values (mean: $\delta^{18}O$= −5.49‰; $\delta^{13}C$=+2.34‰). These authors linked this cementation phase to a shallow burial meteoric flow under equatorial climate during Durancian uplift. This diagenetic event led to micrite re-crystallization, and development of microporosity (MF3). Since La Fare carbonates

309 were exhumed at that time (Léonide et al., 2014) the meteoric fluids led to similar diagenetic
310 modifications (Step 1 on Fig. 8):

    (i)    Cementation of C1a, partly filling intergranular porosity (Fig. 9B1a)
    (ii)    Micrite re-crystallization and microporosity MF3 setup by Ostwald ripening
        processes (Ostwald, 1886; Volery et al., 2010).

314 The micrite re-crystallization strongly increased rock porosity due to enhanced microporosity
315 (Fig. 9B1b). Microporous limestones have a high matrix porosity but low to moderate matrix
316 permeability (Deville de Periere et al., 2011; Jack and Sun, 2003). Indeed, in the case of
317 Barremian limestones of La Fare anticline, porosity is >10% but located in the grains, what
318 restricts possible flow pathways. Resulting from this event, Urgonian carbonates formed a type
319 III reservoir *sensu* Nelson (2001).

    b.  Fault related diagenesis – alteration of reservoir properties

**Normal faulting-related diagenesis**

323 The Castellas fault first nucleated during Durancian uplift (Aubert et al., 2019b; Matonti et al.,
324 2012) impacting the host Urgonian carbonates.
325 In porous granular media, fault nucleation mechanisms can lead to dilation processes (Fossen
326 and Bale, 2007; Fossen and Rotevatn, 2016; Main et al., 2000; Wilkins et al., 2007; Zhu and
327 Wong, 1997), and under low confining pressure (<100KPa; Alikarami & Torabi 2015). Because
328 this process leads to dilatancy, it increases the rock permeability (Alikarami and Torabi, 2015;
329 Bernard et al., 2002) in the first stage of deformation bands (Heiland et al., 2001; Lothe et al.,
330 2002) what allows fluids to flow.
331 Castellas fault zone nucleated within a partially and dimly cemented host rock under low
332 confining pressure, in an extensional stress pattern, at a depth <1km (Lamarche et al. 2012).
333 Barremian host rock presented properties (porosity/stress pattern/confining pressure) close to
334 the porous granular described above. Moreover, Micarelli et al. (2006) showned that, during
335 early stages, fault zones in carbonates have a hydraulic behaviour comparable to deformation
336 bands. Hence, in the Urgonian carbonates of La Fare sector, dilatant processes occurred as an
337 incipient fault mechanism and enhanced fluid circulations along the deformation bands. These
338 fluid flows led to the cementation of C1b (Step 2 on Fig. 8). However, dilation bands are
339 unstable and grain collapse occurs swiftly after the beginning of the deformation due to an
340 increase in the loading stresses (Lothe et al., 2002). This explains why C1b does not fill all
341 intergranular porosity. Consequently, as all micritic grains in fault zone are cemented by C1b,
342 the bulk isotopic measurements are strongly influenced by C1 cement isotopic values. This is
343 the explanation why in transect 3 the bulk isotopic values 30m apart from the fault (means of -
344 5.26‰ for $\delta^{18}$O and -0.82‰ for $\delta^{13}$C) are close to bulk isotopic values far from the fault plane
345 (188m; -5.37‰ for $\delta^{18}$O and -0.65‰ for $\delta^{13}$C, Fig. III 6A). Others dilation bands has also been
346 described by *Kaminskaite et al.* (2019) in the San Vito Lo Capo carbonates grainstones (Sicily,
347 Italy). These dilation bands also led to cementation of the carbonate rock.

348 The C1a and C1b led to a local rock embrittlement and to a porosity decrease by cementation
349 of the microporosity. During the first stages of fault evolution in low porosity limestones,
350 intense fracturing of the fault zone predating fault core formation is known to increase the
351 permeability (Micarelli et al., 2006). In the studied faults, the first brittle event allowed an Alrich fluid to flow with micro-metric quartz grains in the barren fractures, and C2 to 
[revised manuscript text omitted]

830

831

---

## Referee Report (RR2)

Review of manuscript **se-2019-153** submitted to Solid Earth, Aubert et al. "*Diagenetic evolution of fault zones in Urgonian microporous carbonates, impact on reservoir properties (Provence - SE France)*" by Mattia Pizzati.

**General remarks**

Dear Authors and Editor,

here you will find the revised version of the submitted manuscript. Revisions are made by describing the issues found line by line and also on the pdf file of the manuscript, in which critical points were highlighted in green. Comments on figures and figure captions are presented at the end of this file as well.

The present version has been thoroughly reviewed by the authors and massive improvements were made. The few major points highlighted in the previous revision have been fully discussed and fixed. Also the grammar and the clarity of sentences are enhanced and to me manuscript reading is much easier now and concepts are expressed clearly.

The comments I made and reported below are aimed to further increase the clarity and to correct some typo mistakes occurred, I guess, during the revision process. Suggested corrections can be done very quickly.

Still, I suggest all the authors to carefully read many times the text to seek any tiny error. I tried my very best but I'm not a native English speaker, so I could have missed some errors. I'm sure the authors will agree with me that it's not the best to discover errors after the paper is published in the final form.

After the completions of these minor-technical revisions, I believe that the present manuscript can be considered worth of being published on the Solid Earth Special Issue.

I hope this revision may help the authors to improve the final shape of the manuscript. It's a very interesting subject and deserved to be described in a proper way.

In case any doubts and questions arise from corrections, please contact me: mattia.pizzati@studenti.unipr.it

**Detailed comments line by line**

Line 18: Perhaps "*carbonate staining*" or "*staining technique*" would sound better than "*red alizarin*".

Line 27: Here please correct to the plural form "depend" instead of the singular "depends".

Line 50: Check if this structure suits better to the sentence "faulting and diagenetic processes".

Line 57: Perhaps here "*in*" sounds better than "*on*" since you are describing fault zones inside carbonates.

Line 70: Maybe here I would erase "the" before "Late-Cretaceous times".

Line 70: Here is better to cancel the hyphen between "*Late-Cretaceous*", to keep the same style you adopted throughout all the text.

Line 71: I would erase the highlighted "*platform*" which sounds like a repetition of what is stated at the beginning of the sentence.

Line 72: Please erase "plate" after "Iberia", otherwise you will have a repetition.

Line 85: Please correct "swith" with the word "with".

Line 88: I saw you used different styles describing the trend of faults and structures. In this line you used "*N060-070*", while in line 101 it is reported as "*N000° and N170°*". I suggest to keep the same style in the entire text, decide which one you prefer to me they are both correct.

Line 89: Consider changing "composed" with "composed of" or "comprising".

Line 91: Keep a space between the length and the measurement unit as follows "50 m-long".

Line 91: Here maybe better than "extension" you can use "outcrop".

Line 93: Same comment as in Line 88.

Line 100: Please correct "bear" with "bears".

Line 101: Same comment as in Line 88.

Line 102: Please add a space between "whichreactivated" to "which reactivated".

Line 103: Please add a space between "ledto" to "let to".

Line 112: Please add a full stop at the end of this sentence.

Lines 119-120: I would turn "*stereographic projections*" to the singular form, because in Fig. 2F, G you show one single stereonet per figure.

Line 120: Please pluralise "photomicrograph" to "photomicrographs".

Line 120: Please erase one of the parenthesis at the end of the sentence.

Line 129: Here I would erase "in" after "cross-cut".

Line 136: Please erase the space after the first parenthesis.

Line 140: Please correct the term with "strike-slip".

Line 156: Typically Alizarin in studies related to diagenesis and geochemistry is reported as "*Alizarin Red S*", with all first letters as capitals.

Line 158: Here maybe is better to use the plural form "*generations*" instead the singular one.

Line 162: Decide if you want to write the number of samples in extended form or as numbers.

Line 168: Is necessary to put "Bulk" in capitals?

Line 174: Please erase "respectively".

Line 182: Please add a space between "measuredon".

Line 184: Add ":" after "mean" as you did previously in the same sentence.

Line 188: Please add a space between "than7%".

Line 195: Is necessary to put "Red" in capitals?

Line 201: Please correct "identified" with "identify".

Line 209: Here I would erase "of".

Line 211: Here I would erase "of".

Line 215: Please add "are" before "made of".

Line 215: If you are referring to calcite than you should correct to the third person form "*exhibits*".

Line 228: "with maximum size of" sounds better than "sized of maximum".

Line 229: Please add "presence" after "show the".

Line 242: Here are you referring to the "FR1" matrix? If so, please correct accordingly.

Line 243: Maybe here use the plural form of "zonation" "zonations".

Line 245: Please substitute "and" with "while".

Line 253: Please add a space between "layersfrom".

Line 256: Here I believe it is preferable to use the present simple form "*appears*" instead of "*appeared*".

Line 258: Please correct to the third person form "increases" instead of "increase".

Lines 268-269: Please erase "*transect*" because there is a repetition and pluralise "*transects 1 and 3*".

Line 277: In the first line please correct the symbols you used for the delta notation, the correct nomenclature should be " $\delta^{13}C$ ,  $\delta^{18}O$ ".

Line 281: Please change "deplets" with "are more depleted".

Line 285: Please correct "fluctuates" with "fluctuate".

Line 287: Please correct "ranges" with "range".

Line 289: Please correct "ranges" with "range".

Lines 291-292: When you describe the isotopic interval put always as first value the most negative and the second one as the less negative or positive. Here you are doing the opposite of what is conventionally used by geochemists. Keep the same style in the entire text.

Line 293: Please erase "respectively".

Line 300: You should add a semicolon or a full stop at the end of this sentence.

Line 307: Please erase "to".

Line 308: Please put "C2" before "cement".

Line 311: Please change "after" with "is subsequent to".

Line 315: Please add a space between ",cementation".

Line 332: Please correct to the third person form "*does*" instead of "*do*": the subject of the sentence is "*majority*" which is singular.

Line 333: The same comment as in lines 291-292, put first the most negative value you have in your dataset.

Line 361: At this point may sound better to write like this "deformation band development".

Line 365: Please pluralise "rocks" since you write "were likely characterised".

Line 366: Here I think "*petrophysical*" would suit better what you are explaining rather than "*petrographical*".

Line 373: Perhaps here this would be better and simpler "This could explain".

Line 379: At this point "*carbonates*" may be omitted because after you write "*grainstones*", because the reader should already know that you are describing carbonate rocks.

Line 391: Please erase one of the full stops at the end of the sentence. There are two full stops.

Line 407: Here maybe I would change "*C3*" with "*it*" to avoid repetition with the opening of the sentence. Furthermore, maybe use "*precipitate*" instead of "*cement*".

Line 408: Please add a space between ".The".

Line 409: Please correct "fluid" with "fluids".

Line 429: Please correct "Apennines" with "Apennine".

Line 429: Please correct "strike" with "strike-slip".

Line 438: Please add a hyphen between "short-lived".

Line 446: Please add a hyphen between "fault-related".

Line 467: Please change "leading" with "led".

Line 467: Please add a space between "Fig. 9, B7".

Line 480: See if this suggestion suits you "*transects 2 and 3*" instead of "*transect 2 and transect 3*".

Line 498: Here I would erase "the" if you feel this could be an option.

Line 498: "Wider" better than "larger".

Line 499: Please correct "others" with "other".

Line 499: Keep a space between the length and the measurement unit "30 m and 40 m".

Line 500: Please correct "triggerred" with "triggered".

Line 513: Please correct "what" with "that".

Line 527: Please invert the order of the two words "comes mainly" to "mainly comes".

Line 546: Add a space between "8(".

Line 547: Maybe you should capitalise also "porosity".

Line 547: Decide if you want to put in capitals also "*zone*", but do the same both for the fault core and damage zone.

Line 573: I would put the comma before "and".

Line 582: Please correct to the singular form "zone".

Line 586: Please add "in" before "the damage zone".

**Comments on figures and figure caption**

Table 1: in the "*Fault core thickness*" column in the last raw at the bottom please add a space between "*to100*".

Figure 5 caption: In the second sentence please correct "micritszed" to "micritised".

[revised manuscript text omitted]

- 113

109

---

## Author Response (AR2)

Dear editor,

I am pleased to send you the revised version of my paper on "Diagenetic evolution of fault
zones in Urgonian microporous carbonates, impact on reservoir properties (Provence – SE
France).

You will see that most of the corrections have been respected as requested.

You will find below, the comments to all remarks. In addition, I made a table to survey the
reviewer comments with the following colour code

-   Corrections validated are in **green**
-   Corrections in **red** have been considered un-useful or inappropriate
-   Corrections in **blue** refer to comments at the bottom of the table

Best regards,

Irène Aubert

**I.   General remarks**

*"Relative to the scientific part of the manuscript, my main concern is related to the final model*
*in which the permeability contribution was calculated. To me it feels like a good-faith effort,*
*but it would need to be grounded on your field data. To be more explicit, the percentage you*
*calculate for permeability from fault core, fracture network and matrix, should be calibrated*
*according to the width of the fault structure, fracture connectivity and so on."*

The comment of the reviewer is interesting, but cannot be realized in this manuscript because:
- Such a quantification in our field case is too much uncertain because the fault zones are too
heterogeneous to do this exercise.
- This is too apart from the subject of the paper,
-   this   would   be   much   too   much   and   could   represent   another   paper   itself.
- Therefore we mentioned this issue in the text as follows: ""The percentage assigned to the
fault core or to the matrix are qualitatively estimated. Further quantification could be evaluated,
for instance, with the width of the fault core and damage zone domains, or by estimating the
fracture network volume. However, no recent study have provided such quantification." **(line
534 to 537)**

**II.   Specific Comments**

| N° line | validat. | not app. | com. | Reviewer's remarks p |
|---|---|---|---|---|
| 10 | ok | | | Please make plural the term "reservoir" changing it to "reservoirs"; moreover |
| 10-12 | ok | | | Please add "diagenetic" before "processes"; modify "that modify" with |
| 13 | ok | | | Please change "Focussing" with "Focusing". |
| 14 | ok | | | Please move the word "impact" before "the fault zone". Rather than using |
| 14 | ok | | | I would change "It" with "This contribution focuses…". Throughout all the text try to |
| 15 | ok | | | Please correct "La Fare Anticlinal" with "La Fare Anticline". In the same sentence |
| 16 | ok | | | Please correct as follows from the form "orthogonal to the fault zones" to |
| 17 | ok | | | Please change from "Diagenetic elements were determined on 92 thin section…" |
| 18 | ok | | | Why are these words "Polarized Light Microscopy" in capitals? Is that necessary? |
| 18 | ok | | | Maybe here it is better to state more precisely "stable isotope measurements" |
| 19 | ok | | | Here, I would modify "2" with "two". |
| 20 | ok | | | Please correct "highlight" with the third person form "highlights". |
| 20 | ok | | | Here, I would modify "2" with "two". |
| 20 | ok | | | Please modify and make "drain" plural "drains". |
| 21 | ok | | | Please add a line separator between "low temperature" as follows "lowtemperature". |
| 21 | ok | | | Here I would erase "fault zone" before "calcite cementation". |
| 22 | ok | | | Here, I would modify "2" with "two". |
| 22 | ok | | | You are mentioning here "two subsequent phases". Do you refer to tectonic |
| 22 | | x | | Please add "petrophysical" before "properties". |
| 25 | ok | | | Please change "porosities" with "porosity values". |
| 26 | | x | | Please change "heterogeneous properties" with "heterogeneity". |
| 26 | ok | | | Please correct "depend" with the third person form "depends". |
| 27 | ok | | | Please erase "they" and add "carbonates may" before "determine". |
| 29 | ok | | | Please correct the beginning of the sentence as indicated "moreover, fault |
| 32-33 | | x | | Here, I would modify the sentence as follows: "Fault zones in cohesive rocks |
| 37 | ok | | | Instead of "mixed zones", maybe is better "structures with mixed hydraulic |
| 37 | ok | | | Please correct "depending of" with "depending on". |
| 38 | | x | | Please make singular "fluid flows" to "fluid flow" |
| 39 | ok | | | Please correct "Earth crust" with "Earth's crust". |
| 40 | ok | | | Here, maybe you can change the structure of the sentence as indicated: ", and are capable o increasing the…". |
| 40 | ok | | | Please correct " fluids-rock interaction " with "fluid-rock interaction" |
| 41 | ok | | | You can add "diagenetic" before "secondary processes". |
| 42 | ok | | | Please add a line separator between "Fault related" as follows "Fault-related". |
| 45 | ok | | | Here maybe I would change "duplication of fluid pathways/barriers" with |
| 48 | ok | | | Please erase "of the" before "faulting" and restructure the sentence as indicated |
| 51 | ok | | | Maybe here "formations" should be capitalized and also please erase the space |
| 56 | | x | | Please add a hyphen between "poly-phasic", as you did above |
| 60 | ok | | | Here, I would modify "2" with "two". |
| 60 | ok | | | Please correct "crosscutting" with "cross-cutting". |
| 60 | ok | | | Please erase "facies" before "carbonates". |
| 61 | ok | | | Please capitalized "South-East Basin". |

| | | | | |
|---|---|---|---|---|
| 63 | ok | | | With "larger extension" are you referring to the areal extension of the carbonate |
| 64 | | x | | With "larger extension" are you referring to the areal extension of the carbonate |
| 65 | ok | | | Please correct "bauxite deposits" with "bauxite deposition". |
| 67 | ok | | | Please modify these words as follows: ", and development of E-W-trending |
| 67 | | x | | Is necessary the hyphen between "Guyonnet-Benaize"? |
| 68-69 | | x | | Here I would restructure the sentence as indicated: "During Late Cretaceous |
| 70 | ok | | | Please change the structure of this part of the sentence: "...between Iberian and |
| 70 | ok | | | Please modify from "cited references" to "references therein |
| 71 | ok | | | Please modify from "cited references" to "references therein". |
| 72 | ok | | | Please erase "which" before "gave rise". |
| 72 | ok | | | Here you should add "E-W-trending north-verging thrust faults" otherwise the |
| 72 | | x | | Please change "ramp folds" with "thrust-related folds". |
| 73 | ok | | | Please modify from "cited references" to "references therein |
| 75 | | x | | Please change "dimly" with "weakly" |
| 76 | ok | | | To what "structures" are you referring? I guess they are the contractional ones |
| 79 | ok | | | Here, I would modify "2" with "two". |
| 79 | ok | | | Maybe it is better to write "a Km-scale" instead of "kilometric-scale". |
| 79 | | x | | Please change from "...fault system on the E-W-trending..." to "...fault system |
| 80 | ok | | | Please correct "anticlinal" with "anticline". |
| 82 | ok | | | Please add a space between "120m-thick" to separate the length from the |
| 82 | ok | | | Please change "calcarenite unit" with "calcarenitic unit" |
| 83 | ok | | | Please add a space between "40m-thick" to separate the length from the |
| 83 | ok | | | Please change the structure as indicated: "...coral-rich calcarenite unit, and an |
| 84-85 | ok | | | Here, I would modify as follows: "Santonian age coarse grained rudist |
| 86 | ok | | | Here you state that the Castellas Fault has a length of one Km, but in Fig.1A the |
| 86 | ok | | | You should also define the kinematic of the fault: I would recommend to write as |
| 87 | ok | | | Here change the structure of the sentence as suggested: "(Fig. 2A, B; table 1). Its |
| 88 | ok | | | You can modify from "The second fault zone" to "The second investigated fault |
| 88 | | x | | Here, I would modify "5" with "five". |
| 89 | | x | | Maybe here change "50m-long interval" with "50 m-wide outcrop". Pay attention to |
| 88 | ok | | | Sub-fault are organised in two sets sounds better than "Sub-faults are |
| 90 | ok | | | Instead of "Set one" use "The first one". |
| 91 | ok | | | Please change "orange on Fig. 2F" to "orange traces in Fig. 2F". |
| 91 | ok | | | Add the kinematics of the fault set "with left-lateral strike slip...". |
| 92 | ok | | | Please change "orange on Fig. 2F" to "red traces in Fig. 2F". |
| 92 | | x | | Here, I would modify "5" with "five" |
| 93 | ok | | | What is this asymmetry about? Is related to a different structure or width of the |
| 95 | ok | | | Please add "distinct" before "tectonic events". |
| 96-98 | ok | | | I tried to modify the two sentences as indicated: "...the Middle-Cretaceous |
| 100 | ok | | | I merged the first two sentences: "...(see references cited in Espurt et al. 2012), |
| 101 | ok | | | Please change "leads" to "led". |
| 101 | ok | | | Please modify "neo-formed" with "newly-formed". |
| 105 | ok | | | Please add "present-day" before "reverse throw". |
| 108 | | x | | Here, I would modify "4" with "four". |
| 108 | ok | | | Throughout all the text I saw that sometimes you have used numbers to identify |

| | | | | |
|---|---|---|---|---|
| 109 | ok | | | Please change "transect T1" with "transect 1". |
| 109 | ok | | | Instead of "bed", which can be misunderstood maybe here you can use |
| 109 | ok | | | Please correct "pelloidal" with "peloidal", I think it should be the correct form. |
| 110 | ok | | | Please erase the space in "Fig. 2D a" to "Fig. 2Da". |
| 110-111 | ok | | | you can restructure the sentence as suggested: "Transects 2 and 3 crosscut |
| 112 | ok | | | Make the plural form of "echinoderm" to "echinoderms". |
| 114 | ok | | | Make the plural form of "amount" to "amounts". |
| 114 | ok | | | Make the plural form of "bryozoan" to "bryozoans". |
| 115 | ok | | | Eliminate the comma in the reference to the figure: "Fig. 2G, a" change to "Fig. |
| 117-119 | ok | | | tried to restructure the sentence as follows: "Three different fault rock |
| 119 | ok | | | Please change "normal" with "extensional |
| 121 | ok | | | Add this detail at the end of the sentence: "...<30% of fine-grained grey matrix". |
| 122 | ok | | | Add "strike-slip" before "reactivation" |
| 122 | ok | | | Please add "to" before "the onset". |
| 123 | ok | | | Please correct the third person form "present" with "presents" and also change |
| 125 | ok | | | Here maybe modify the sentence in this way: "...sub-rounded clasts belonging to |
| 126 | | x | | What is the nature of the cemented matrix? Are you able to distinguish the |
| 127 | ok | | | Here you are mentioning the reactivation of the D19 fault zone: be more specific |
| 128 | ok | | | Make the plural form of "clast" to "clasts". |
| 128 | ok | | | Add this word at the end of the sentence: "from the nearby damage zone |
| 133 | | x | | Here, I would modify "4" with "four". |
| 134 | ok | | | Please correct the sentence as indicated: "Microfacies were determined...". |
| 136 | ok | | | Please, modify this detail: "with a solution of hydrochloric acid, Alizarin Red S |
| 137 | ok | | | Please erase "The" at the beginning of the sentence and capitalize "Thin...". |
| 137 | ok | | | Please modify "analyzed" with "analysed". Pay attention to the use of UK or USA |
| 138 | ok | | | Please add these words: "...the different generation of calcite cements". |
| 140 | | x | | Is this underscore necessary to identify the instrument? |
| 141 | | x | | Is this underscore necessary to identify the instrument? |
| 141 | ok | | | Here, I would modify "2" with "two". |
| 142 | ok | | | Here, I would modify "2" with "two". |
| 143 | ok | | | Please add "beam" before "current". |
| 143 | ok | | | Keep always a space between the value and the measurement unit; do this in all |
| 143 | ok | | | Please make the plural form of "surface" "surfaces |
| 146 | ok | | | Keep always a space between the value and the measurement unit: "80 µm," |
| 147 | | | 1 | What do you mean with the words "bulk rock"? Are you referring to the |
| 148 | ok | | | Please add "isotopic" before "values". |
| 149 | | x | | Is correct the name of the spectrometer with the symbol "+" rather than the word |
| 151 | ok | | | The symbol you used for the delta is not the same adopted previously in Line |
| 151-152 | ok | | | I modified the sentence as follows, check if suits you: "The standard |
| 159-160 | ok | | | I tried to fix in this way: "Porosity measured on 92 plug samples show a |
| 161 | ok | | | Please change "in" with "within". |
| 163 | ok | | | Please add these words at the beginning of the sentence: "Along transects, |
| 164 | ok | | | Insert a space between the value and the measurement unit: "60 m" and not |
| 164 | ok | | | Please change "transect T2" to "transect 2". |
| 165 | ok | | | Please change "low < 7%" to "lower than 7%". |

| | | | | |
|---|---|---|---|---|
| 165 | ok | | | Delete the space after 1.53. |
| 165 | ok | | | Substitute the comma at the end with a full stop. |
| 166 | ok | | | Here I would write "is wider than 40 m" rather than "is >40m". |
| 166 | ok | | | Insert a space between the value and the measurement unit: "30 m" and not |
| 167 | ok | | | Here please change "In a 10m-thick" with "In a 10 m-wide". |
| 168 | ok | | | If you want to keep the nomenclature used above than change "T1 and T3" with |
| 169 | ok | | | Please add "found" between "are in narrow". |
| 169-170 | ok | | | tried to write as follows this part of the sentence: "...in narrow zones (less |
| 170 | ok | | | With the term "lens" are you describing the rock volume comprised between F4 |
| 172-173 | ok | | | Please modify the beginning of the sentence as indicated: "Microscope |
| 173 | ok | | | Please remove the space in "Fig. 3C a" to "Fig. 3Ca". |
| 174 | ok | | | Add a comma after "φ<5%". |
| 174 | ok | | | Please remove the space in "Fig. 3C b, c" to "Fig. 3Cb, c". |
| 174 | ok | | | It is not clear to me what do you mean with "barren stylolites". Is this word |
| 174 | ok | | | Erase "are distinguished" at the end of the sentence. |
| 177 | ok | | | Please correct "micritized" with "micritised", if you want to keep the UK version of |
| 177 | ok | | | Here, I would modify "2" with "two". |
| 178 | ok | | | Here, I would modify "2" with "two". |
| 179 | ok | | | Why is this citation reported in italics? |
| 180 | ok | | | The reference to "Fig. 4A, 4B" should be "Fig.4A, B". |
| 181 | | x | | Here I'm really struggling with the terminology you used "puntic and serrate" are |
| 182 | ok | | | Please put the porosity value between parenthesis and erase the space before |
| 183 | ok | | | Please insert the porosity percentage in parenthesis |
| 177-183 | ok | | | Inside this paragraph you should also give some info concerning the |
| 185 | ok | | | Please add "different" between "Eight cement". |
| 185 | ok | | | Alizarin Red S should be with all first letters in capitals |
| 185 | ok | | | Add "and" after "coloration". |
| 186 | ok | | | Please correct "made up of" with "made of". |
| 186 | ok | | | Please correct from the third person form to plural "exhibits" to "exhibit". |
| 188 | ok | | | Here, I would modify "2" with "two". |
| 189 | | x | | With "thickness" here are you describing the maximum thickness of the |
| 189 | ok | | | Here, I think it would be more correct to use this symbol "~" to indicate the word |
| 190 | ok | | | Here, I would modify "2" with "two". |
| 190 | ok | | | Add an hyphen between "dog-tooth". |
| 192 | ok | | | Again, here is that possible to describe the crystal size rather than the |
| 192 | ok | | | Here, I think it would be more correct to use this symbol "~" to indicate the word |
| 192 | ok | | | Be more specific concerning the reference to the figure: I believe here you are |
| 193 | ok | | | Better than "C1b values" maybe you should use "C1b areal occurrence". |
| 193 | ok | | | Please correct the third person form "increases" instead of "increase |
| 201 | ok | | | Please change the sequence of words from "replacive phases occur largely..." to |
| 203 | ok | | | Use the hyphen between "dull orange" "dull-orange". |
| 203 | ok | | | Please change the structure as indicated from "...only found in fault core veins" |
| 204 | ok | | | Please erase "elements" after "Si and Al". |
| 205 | ok | | | Please erase "an" and add at the end of the sentence "and have black |
| 203-205 | ok | | | Please add also some info relative to the size of the cement crystals you |

| | | | | |
|---|---|---|---|---|
| 206 | ok | | | Use the hyphen between "red dull" "red-dull". |
| 208 | ok | | | Add "only" before "to the fault zone". |
| 206-208 | | x | | Please add also some info relative to the size of the cement crystals you |
| 211 | ok | | | Please insert a space between "500" and "μm". |
| 211 | ok | | | Please insert "previous" before "dolomitization phase |
| 212-213 | ok | | | Please change here from "micritic inclusion in the crystal and..." to "micritic |
| 215 | ok | | | Please insert a space between "300" and "μm". |
| 217 | ok | | | Please correct the reference to "Fig. 5G, 5H" with "Fig. 5G, H". |
| 219 | ok | | | Here, I would modify "2" with "two". |
| 219 | ok | | | Please erase "which" after "C4a". |
| 220 | ok | | | Please correct the plural form of "band" with "bands". |
| 220 | ok | | | Please add "thin" before "non-luminescent zones". Change also "bands" with |
| 221 | ok | | | Correct the nomenclature of transects, here maybe use "transects 1 and 3" as |
| 223 | ok | | | Use the hyphen between "red dull" "red-dull". |
| 227 | ok | | | Please change "formation" with "karst deposit". |
| 228-229 | ok | | | See if this correction suits you: "This karst deposit present a stack of |
| 230 | ok | | | Please correct "clasts fall" with "grain fall". Keep in mind that the term clast is |
| 230 | ok | | | Please correct the singular "has" to plural form "have". |
| 230-231 | ok | | | tried to improve the clarity of the sentence: "Micritic layers have been |
| 233 | ok | | | Please change "proportion" with "areal amount" and also modify the third person |
| 236-238 | | x | | I fell this is a repetition of what was previously presented inside the method |
| 239 | ok | | | Again, what is the meaning of "bulk rock"? If you are referring to the undeformed |
| 239 | ok | | | Intergranular volume is better than "intergranular space". |
| 241-242 | ok | | | If it is more correct change the reference to the figure from "Fig. 6A, 6B" to |
| 242 | ok | | | Again, "bulk rock" isn't this the "host rock"? |
| 243 | ok | | | Here, I would modify "2" with "two". |
| 243 | ok | | | For "Set one" you can use the number "Set 1". I believe to identify the type of |
| 244 | ok | | | I don't think that using the symbol "&" is suitable here, just write "and". |
| 245 | ok | | | As above, "Bulk values" is too generic. If you are describing the isotopic dataset |
| 247 | ok | | | Please erase "the" before "transect 3". |
| 247 | ok | | | Please erase "along transect" after "slightly vary". |
| 248 | ok | | | At the end of this sentence after "δ13C " add ", respectively". |
| 248 | ok | | | Change "Contrarily" with "On the contrary, ...". |
| 248-251 | ok | | | I tried to fix these sentences as follows: "On the contrary, values are more |
| 252 | ok | | | Please change "spaces" with "volume". |
| 252 | ok | | | Maybe "infillings" is better than "fills". |
| 252 | ok | | | Please make the plural form "fault rocks". |
| 253 | | x | | Here, I would modify "5" with "five". |
| 254 | ok | | | Personally I modified the sentence as follows and fell like it is clearer: "isotopic |
| 256 | ok | | | Similar to the comment above: "isotopic values of C3 cement...". |
| 258 | ok | | | Again: "isotopic values of C4 cement...". |
| 258 | ok | | | Maybe "infillings" is better than "fill". |
| 259 | ok | | | Please add a space between "from-5.10‰" to have "from -5.10‰". |
| 260 | ok | | | Please erase the space between "FR 2" to "FR2". |
| 260 | ok | | | Please add a space between "from-6.55‰" to have "from -6.55‰". |

| | | | | |
|---|---|---|---|---|
| 261 | ok | | | Maybe "infillings" is better than "fill". |
| 262 | ok | | | Please restructure the last part of the sentence as follows: "...for δ18O and δ13C |
| 263 | | x | | Here, I would modify "4" with "four". |
| 263 | ok | | | Here, I would modify "2" with "two". |
| 264 | ok | | | Please add a semicolon ";" at the end of the sentence. |
| 265-266 | ok | | | Please start the sentence as follows: "isotopic values of C5 cement |
| 267-268 | ok | | | Please start the sentence as follows: "isotopic values of FR3 matrix have a |
| 275 | ok | | | Please change "thanks to" with "via". |
| 275 | ok | | | Please pluralise "cross-cutting relation" to "cross-cutting relations". |
| 276 | ok | | | modified the order and position of some words here: "Indeed, the veins filled |
| 277 | ok | | | I would change this sentence as follows: "Thus, C2 cementation postdated C1 |
| 277 | ok | | | Please erase "The" at the beginning of the sentence |
| 278 | ok | | | Here there is a missing reference to the correct figure. I guess it should be "Fig. |
| 278 | ok | | | Please change "is ante-FR1..." with "formed prior to FR1...". |
| 279 | ok | | | Please change "post-C2" with "after C2". |
| 279 | ok | | | would use "extensional" rather than "normal". |
| 281 | ok | | | Please remove the comma after "formation and, are related...". |
| 281-282 | ok | | | Please modify the sentence as indicated: "Replacive dolomite is found |
| 282 | ok | | | Here I guess that this is the wrong reference to the figure. I believe it should be |
| 282 | ok | | | Please make the past simple version of "develop" "developed". |
| 282 | ok | | | Please, move "C4" before "cement". |
| 283 | ok | | | Please change "postponed" with "postdated |
| 284 | ok | | | Please modify the order of the words: "...developed during the strike-slip |
| 287 | ok | | | Please correct "La Fare anticlinal" with "La Fare anticline". |
| 287 | ok | | | Here, I would modify "3" with "three". Also substitute "important" with "major". |
| 291 | | x | | I don't know if this is the right term, but perhaps "micro-boring organisms" |
| 294 | ok | | | Please add a hyphen between "low and energy" "low-energy". |
| 294 | ok | | | Please add "environment" at the end of the sentence after "inner platform". |
| 295 | ok | | | Move "C0" before "cement". |
| 295-296 | ok | | | I tried to fix this sentence: "...formed around grains giving rise to a solid |
| 299 | ok | | | Here, I would modify "2" with "two". |
| 299 | ok | | | Please substitute "points" with "pairs", and "sampled of" with "pertaining to". |
| 300 | ok | | | Please add "isotopic" before "depletion". |
| 304 | ok | | | Please modify "characteristic for" with "characteristic of". |
| 307 | ok | | | Please change "meteoric flow" with "meteoric fluid circulation". |
| 308 | ok | | | Please add "to the" before "development". |
| 315 | ok | | | Please add a hyphens as indicated: "low-to-moderate matrix....". |
| 316-318 | ok | | | did some changes to this sentence: "Even if Barremian limestones of La |
| 318 | ok | | | Please change "Resulting from this event,..." with "Due to this characteristic,...". |
| 320 | ok | | | Please add a hyphen between "Fault and related" "Fault-related". |
| 324 | ok | | | 324: Better that "impacting" maybe you should try with "affecting". |
| 327 | ok | | | Please erase ", and" just after the references in parenthesis and also add a |
| 327 | ok | | | Add a space between "<100KPa" to have "<100 KPa". |
| 327 | ok | | | Modify the reference to "Alikarami & Torabi 2015" as "Alikarami and Torabi, |
| 329-330 | ok | | | Here I fixed in this way: "...of deformation band development (Heiland et |

| | | | |
|---|---|---|---|
| 331 | | x | poorly sounds better than "dimly". |
| 331-332 | ok | | Add a hyphen between "low confining pressure" "low-confining pressure". |
| 332 | ok | | Please change the term "pattern" with "regime". |
| 332 | ok | | Add a space between "<1Km" to have "<1 Km". |
| 333-334 | ok | | I reworked this sentence as indicated: "Under these conditions, Barremian |
| 334 | ok | | Please correct "showned" with "showed". |
| 335 | ok | | Please add "of deformation" after "early stages". |
| 336 | ok | | Please add "in carbonates" after "deformation bands". |
| 336 | ok | | Please change "sector" with "area". |
| 337 | | x | Please correct to the singular form "circulation" instead of "circulations". |
| 337-338 | ok | | Please erase "These" at the beginning of the sentence and make the |
| 338-339 | ok | | would change the sentence as follows: "however, dilation bands were |
| 340 | ok | | Please modify to the singular form "loading stress" instead of "loading stresses". |
| 342 | ok | | Here maybe start the sentence as indicated. "This could be the explanation...". |
| 343 | ok | | Add a space between "<30m" to have "<30 m". |
| 345 | ok | | Add a space between "<188m" to have "<188 m". |
| 345 | ok | | Here adjust the reference to "Fig. III 6A" with "Fig. 6A". |
| 345 | ok | | Modify the beginning of the sentence as suggested: "Dilation bands have also |
| 346 | ok | | Please correct "Sicilly" with "Sicily". |
| 347 | ok | | Please add "selective" before "cementation". Also pluralise "rock" to "rocks". |
| 348-349 | ok | | tried to change the structure of this sentence: "Cementation (C1a and |
| 349 | ok | | Add a hyphen between "low porosity" "low-porosity". |
| 350 | ok | | Please change the last part of the sentence as suggested: "...is known to |
| 351 | ok | | Please erase "an" before "Al-rich". |
| 352 | ok | | Please pluralise "fluid" to "fluids". |
| 352 | ok | | You may explain this with "fine-grained" instead of "micro-metric". |
| 352 | ok | | Again the term "barren" is very unfamiliar to me. Are you referring to an incipient |
| 352 | ok | | If you feel this is an option try to put this last part of the sentence as indicated: |
| 358 | ok | | Use "may" and not "must" you are not 100% sure that this happened, you are |
| 361 | ok | | A few adjustments to this sentence: "As the fault grew, new fracture sets formed, |
| 364-365 | ok | | Please correct "at high depth" with "in deep burial conditions". in |
| 365-366 | ok | | A bit of reworking on this sentence: "...corroborate the hypothesis of |
| 367 | ok | | Please change the beginning of the sentence from "Resulting from" to "Due to". |
| 368 | ok | | Before "down to <5%" insert "with porosity". |
| 368 | ok | | Cancel the space in "Fig. 9 B5" to "Fig. 9B5". Also in Fig. 9 what is stage 5 since |
| 371 | ok | | Please change "Implicitly", with "Following this". |
| 371 | | x | Please add "in this stage" before "was a barrier". |
| 372 | ok | | I would change the beginning of the sentence: "Fluids responsible for |
| 375 | ok | | Please insert a space between "100" and "µm". |
| 376 | ok | | I modified the sentence as indicated: "...came from silica found inside C2 cement |
| 377 | ok | | Please modify the beginning of the sentence: "Silica crystals in C2 veins...". |
| 377 | ok | | Please insert a space between "100" and "µm". |
| 378 | | x | Please add "grains" before "quartz". |
| 379 | ok | | Please add "also" before "Aptian rocks". |
| 380 | ok | | Please substitute "Implicitly" with "According to this,". |

| | | | | |
|---|---|---|---|---|
| 381 | ok | | | Add a hyphen between "Uncemented" "Un-cemented" |
| 381 | ok | | | Please change to the pas simple form "formed" instead of "form". |
| 383 | ok | | | Please add "lateral extent of the..." before "drainage area". |
| 384 | ok | | | Add a hyphen between "high permeable" "high-permeable". |
| 385-386 | ok | | | I would change as follows: "...formations within fault core of strike-slip and |
| 390 | ok | | | Please make the past simple form of "lead" to "led". |
| 392 | ok | | | Please substitute "can be" with "could have been". |
| 395 | ok | | | Please add a reference at the end of this sentence, relative to the difference in |
| 398 | ok | | | I would change a bit the end of the sentence as indicated: "...not favourable for |
| 399 | ok | | | Add a hyphen between "Low temperature" "Low-temperature". |
| 400 | ok | | | Add a hyphen between "high temperature" "high-temperature". |
| 402 | ok | | | Here you can change from "high Mg fluid circulation" to "Mg-rich fluid |
| 403 | ok | | | Please add "domain" after "fault core" at the end of the sentence. |
| 405 | ok | | | Please insert a space between "23" and "Km". |
| 405-406 | ok | | | Please change from "compressive conditions" to "contractional stress |
| 406 | ok | | | At the beginning of the sentence please change from "From these authors..." to |
| 407 | ok | | | Add a hyphen between "low temperature" "low-temperature". |
| 407 | ok | | | After "upwelling fluids" add ", likely Mg-enriched". |
| 409 | ok | | | Add a hyphen between "low temperature" "low-temperature". |
| 410 | ok | | | Please correct "were" with "was". |
| 412 | ok | | | Please correct "reduce" with "reduces". |
| 415 | ok | | | Please change "to" with "into". Also check the reference to "Fig. 9 B6 and C6" |
| 417 | ok | | | Please substitute "finally" with "eventually". |
| 419 | ok | | | Please erase the hyphen between "back-ground" to "background". |
| 420 | ok | | | Please change from "lead to FR2..." to "formed FR2...". |
| 422 | ok | | | Please erase the hyphen between "back-ground" to "background". |
| 423-424 | ok | | | I would modify the structure of the sentence as indicated: "This fluid flow is |
| 424 | ok | | | Please correct "micritized" with "micritised". |
| 425 | ok | | | Please change "what led" with "leading to". |
| 425 | ok | | | Erase the comma and space in the reference to figure "Fig. 9, B7 and C7" to |
| 426 | ok | | | Please change "the" before "fracture porosity" with "that." |
| 426 | ok | | | At the end of the sentence modify as follows: "...permeability was still partially |
| 427 | ok | | | Use italics for the term "sensu". |
| 428 | ok | | | would modify this as indicated: "...and high fracture-related secondary |
| 429 | ok | | | Maybe here is better to use "infillings" rather than "fill". |
| 429 | ok | | | After "dissolution/cementation" add the term "processes". |
| 431 | ok | | | Please add "cement" after "C4". |
| 432 | ok | | | Maybe here is better to use "infillings" rather than "fill". |
| 433 | ok | | | did a few changes to this part of the sentence: "...fluid circulation in the vadose |
| 435 | | x | | Maybe the last part of this sentence would sound better as: "... of meteoric and |
| 436-437 | | x | | Please change "these" with "this" and also make the singular form of |
| 437 | ok | | | Please add "cement" after "C4". |
| 437 | | x | | Please change "on" with "in" in reference to Fig. 8. |
| 438 | ok | | | Higher is not the most precise term to describe isotopic data. if you are talking |
| 439 | ok | | | Please add "cement" after "C1". |

| | | | | |
|---|---|---|---|---|
| 440 | ok | | | Please add "cement" after "C4". |
| 440 | ok | | | Move "C4" before "cement", and change from the plural to the singular form |
| 441 | ok | | | Please change "are" with "is". |
| 441-442 | ok | | | I tried to fix this sentence in this way: "Transect 2 cross-cuts the Castellas |
| 442 | ok | | | Here, I would modify "2" with "two". |
| 446 | ok | | | Please erase the term "local" before "permeability" and also change "allowed" to |
| 447 | | x | | Please make the singular of "circulations" to "circulation". |
| 448 | ok | | | Move "C4" before "cement. |
| 449 | ok | | | Please change "Contrarily" with "On the contrary,". |
| 450 | ok | | | Please erase the word "a" before "drains". |
| 51 | ok | | | Please change "That" with "This". |
| 452 | ok | | | Please correct "formation" with "development". |
| 453 | ok | | | Please change "normal" with "extensional". |
| 453 | ok | | | I modified the last part of the sentence as follows: "...C4 fluids to flow through the |
| 455 | ok | | | Please change "T2" with "transect 2". |
| 455 | ok | | | Please insert a space between "60" and "m". |
| 456 | ok | | | Use "transect 1 and transect 3" instead of "T1 and T3", add a space between "30 |
| 461 | | | 2 | I don't think that the word "sieve" is appropriate to describe the evolution of th |
| 462 | ok | | | Please correct "de-dolomitization" with "de-dolomitized". |
| 466 | ok | | | Please correct "recrystallized" with "recrystallised". |
| 469 | ok | | | I would put also "alpine" in capitals as you did for "Pyrenean". |
| 471 | ok | | | tried to fix this part as indicated: "This implies fluids percolating soils, as results |
| 475 | ok | | | Please correct "Finally" with "Eventually". |
| 475 | ok | | | Please change from "incurring" to "inducing". |
| 476 | ok | | | Please change from "triggered" to "produced". |
| 477 | ok | | | Please change from "flows" to "fluids". |
| 481-482 | ok | | | tried to adjust this part as: "...reservoir where fractures behave as |
| 483 | ok | | | Please change "polyphase" with "polyphasic". |
| 490-491 | ok | | | See if these changes suit you: "...Castellas fault zone, permeability evolves |
| 493 | ok | | | Here, I would modify "2" with "two". |
| 493 | ok | | | Please change "fracture" with "fracturing". |
| 493 | ok | | | Please change "link" with "linked". |
| 494 | ok | | | If you feel it could be an option you can use the extended form "fault core" |
| 495-496 | ok | | | I adjusted the second part of the sentence: "permeability contribution is |
| 497 | ok | | | Again here if you can use the extended version of the name "fault core". |
| 498 | ok | | | Please correct "at 20%" with "for 20%". |
| 498-499 | | | 3 | The calculation of the permeability contribution is nice and to me it |
| 501 | ok | | | Here, I would modify "2" with "two". |
| 501 | ok | | | Erase "the" before "reservoir". |
| 502 | ok | | | Please pluralise "fault" to "faults". |
| 503 | ok | | | I think you should capitalise "SE basin" as "SE Basin". |
| 508 | ok | | | Please correct "their" with "its". |
| 511 | ok | | | Please change "thinly" with "slightly". |
| 512 | ok | | | Please substitute "formation" with "development". |
| 514 | ok | | | Please change "the flows" with "flowing fluids". |

| | | | | |
|---|---|---|---|---|
| 516-517 | ok | | | Add a hyphen between "low temperature" "low-temperature". |
| 517 | ok | | | Please correct "flows" with "fluids". |
| 517 | ok | | | Please correct "This" with "These". |
| 517 | ok | | | Add a hyphen between "low temperature" "low-temperature |
| 517 | ok | | | Please add "fluid" before "flows". |
| 519 | ok | | | Add a hyphen between "high temperature" "high-temperature". |
| 519 | ok | | | Please change "flows" with "fluids". |
| 519 | ok | | | Please add "significant" before "hydrothermal influence". |
| 520 | ok | | | tried to improve the last part as: "...broader rules for complex faults with |
| 522 | ok | | | Please correct "extensive" with "extensional". |
| 522 | ok | | | See if this is better: "...can lead to the development of dilation bands acting...". |
| 523 | ok | | | tried to improve the clarity: "Carbonates are very sensitive to rock-fluid |
| 529-531 | ok | | | Again a bit of reworking: "Late-stage fluids flowed preferentially within the |
| 566 | ok | | | Check and erase the highlighted space |
| 568 | ok | | | There are too much spaces that must be corrected |
| 569 | ok | | | Check and erase the highlighted space. |
| 570 | ok | | | Check and erase the highlighted space. |
| 571 | ok | | | Erase the highlighted full stops. |
| 573 | ok | | | Erase the highlighted full stop. |
| 580 | ok | | | Erase the highlighted full stop. |
| 586 | ok | | | Check and erase the highlighted space. |
| 590 | ok | | | Check and erase the highlighted space. |
| 607 | ok | | | Please erase the comma |
| 610 | ok | | | Check and erase the highlighted space. |
| 611 | ok | | | The name of the institution is not complete. |
| 621 | ok | | | Check and erase the highlighted space. |
| 621 | ok | | | Please erase the comma. |
| 639 | ok | | | Please erase the comma. |
| 641 | ok | | | Erase the highlighted part since it is a repetition and should not be in front of the |
| 656 | ok | | | Please capitalise "jurassic" to "Jurassic". |
| 664 | ok | | | Erase the highlighted full stop |
| 669-670 | ok | | | Check and erase the highlighted spaces. |
| 679 | ok | | | Check and erase the highlighted space. |
| 705 | ok | | | Check and erase the highlighted spaces |
| 714 | ok | | | Check and erase the highlighted spaces and comma. |
| 721 | ok | | | Check the spelling of the journal title. |
| 728 | ok | | | Check and erase the highlighted full stops. |
| 741-742 | ok | | | Please eliminate the duplicated title. |
| 754-755 | ok | | | Check and erase the highlighted full stops. |
| 772-773 | ok | | | Check and erase the highlighted spaces. |
| 774-775 | ok | | | Check and erase the highlighted space and full stop. |
| 791 | ok | | | Check and erase the highlighted space. |
| 803 | ok | | | Check and erase the highlighted space. |
| 809 | ok | | | Check and erase the highlighted space. |
| 814 | ok | | | Check and erase the highlighted space and comma. |

| | | | | |
|---|---|---|---|---|
| 816-817 | ok | | | Check and erase the highlighted space and comma. |
| 819 | ok | | | Check and erase the highlighted comma. |
| 821 | ok | | | Check and erase the highlighted space. |
| | | | | |
| **comment on figures and figure captions** | | | | |
| fig 1 | ok | | | You should insert the symbol of the La Fare anticline in Fig. 1A. |
| fig 1 | ok | | | The kinematic indicators alongside faults are missing in Fig. 1B. |
| fig 1 | ok | | | These names in the legend of Fig. 1A should be all in capitals "Upper Cretaceous, Lower |
| fig 1 | ok | | | Maybe better than "thin calcarenite" you can use "fine-grained", if you are referring to the |
| fig 1 cap | ok | | | You should erase the space between "Figure 1 :" to "Figure 1:". |
| fig 1 cap | ok | | | Please add "trace of" before "stratigraphic column". |
| fig 1 cap | ok | | | Please correct the reference to "C" part of the figure rather than "B". |
| fig 2 | ok | | | The kinematic indicators in both stereo-nets are indistinguishable from the fault traces. |
| fig 2 | ok | | | I would mirror the transect 3 to have SSE on the left and NNW on the right side, just like |
| fig 2 | ok | | | What are the red stars? Are they the positions of samples? If so you should mention them |
| fig 2 cap | ok | | | You should erase the space between "Figure 2 :" to "Figure 2:". |
| fig 2 cap | ok | | | Please change the term "localization" with "position". |
| fig 2 cap | ok | | | What are the "red points" you are referring in the stereo-nets? I can't seen anything but red |
| fig 2 cap | ok | | | In the third line please add "C: Photos of transect 1 and 2." |
| fig 2 cap | ok | | | In the third line please add "D: Photomicrographs of carbonate host-rock facies...". |
| fig 2 cap | ok | | | In the fourth line add "FR1 and FR2" after "fault rock 1 and 2". |
| fig 2 cap | ok | | | In the fifth line please add "G: Photomicrographs of host-rock facies...". |
| fig 3 | ok | | | Try to improve the visibility of the three petrographic images in Fig. 3C, change the |
| fig 3 cap | ok | | | In the first line correct "&" with "and". |
| fig 3 cap | ok | | | In the third line correct "b&c" with "b and c". |
| fig 4 cap | ok | | | You should erase the space between "Figure 4 :" to "Figure 4:". |
| fig 4 cap | ok | | | In the first line please pluralise "white arrow" to "white arrows". |
| fig 4 cap | ok | | | In the first line add a space between "MF1micrite" to "MF1 micrite". |
| fig 4 cap | ok | | | In the second line add a space between "2.5m" to "2.5 m". |
| fig 4 cap | ok | | | In the second line add a space between "MF1micrite" to "MF1 micrite". |
| fig 4 cap | ok | | | In the second line add a space between "2m" to "2 m". |
| fig 4 cap | ok | | | In the second line please erase the "C" which is duplicated. |
| fig 4 cap | ok | | | In the third line add a space between "188m" to "188 m". |
| fig 4 cap | ok | | | In the third line add a space between "95m" to "95 m". |
| fig 4 cap | ok | | | In the fourth line please change "F." to "F:". |
| fig 5 cap | ok | | | You should erase the space between "Figure 5 :" to "Figure 5:". |
| fig 5 cap | ok | | | In the first line please change "micritized" with "micritised". |
| fig 5 cap | ok | | | In the second line please change "space" with "volume". |
| fig 5 cap | ok | | | In the second line please substitute "a&b" with "a and b". |
| fig 5 cap | ok | | | In the second line please pluralize "clast" to "clasts". |
| fig 5 cap | ok | | | In the second line please change "micritized" with "micritised". |
| fig 5 cap | ok | | | In the third line I modified as follows: "C: C3 veins, cements and intergranular volume in...". |
| fig 5 cap | ok | | | In the third line please substitute "a&b" with "a and b". |
| fig 5 cap | ok | | | in the fourth line after "replacive dolomite" add "(RD)". |
| fig 5 cap | ok | | | In the fifth line please correct "quart" with "quartz". |

| | | | | |
|---|---|---|---|---|
| fig 5 cap | ok | | In the fifth line please substitute "a&b" with "a and b"; do it twice. |
| fig 6 | | x | To me it would be more logical to invert Fig. 6A and 6B, to show the reader first all bulk |
| fig 6 | ok | | In both graphs insert the X axis labels for every increment of 2 per mil (2, 4, 6...). |
| fig 6 | | x | In the legend of Fig. 6A it is written "Bulk rock", I wonder if this is actually the undeformed |
| fig 6 | ok | | In Fig. 6C the title of the graph states "Distance to Castellas Fault plane", maybe "Fault |
| fig 6 cap | ok | | You should erase the space between "Figure 6 :" to "Figure 6:" |
| fig 6 cap | ok | | In the first line please correct the symbols you used for the delta notation, it should be |
| fig 6 cap | | x | In the first line you state again "bulk rock" why not "host rock"? |
| fig 6 cap | ok | | In the third line please correct the symbols you used for the delta notation, it should be |
| fig 7 | ok | | The three photomicrographs are too small to appreciate the details. You have plenty of |
| fig 7 cap | ok | | In the second line I slightly modified as follows: "... development (blue), cementation |
| fig 8 | | x | Again, photomicrographs are quite small, but still the reader should be able to see |
| fig 8 | ok | | In the legend please correct "Micro-facies & cement types" with "Micro-facies and cement |
| fig 8 cap | ok | | You should erase the space between "Figure 8 :" to "Figure 8:". |
| fig 9 | ok | | It would be nice to have bigger sketches in Fig. 9A. |
| fig 9 | | x | Also why stage 5 is not reported? In the text it is mentioned. You should consider to |
| fig 9 cap | ok | | You should erase the space between "Figure 9 :" to "Figure 9:". |
| fig 9 cap | ok | | In the second line I would modify "2" with "two". |
| fig 9 cap | ok | | In the second line please correct "curved" with "curve". |
| fig 10 | ok | | Also here size matters! Please make these sketches bigger otherwise you will lose al lot of |
| fig 10cap | ok | | You should erase the space between "Figure 10 :" to "Figure 10:". |
| fig 10cap | ok | | In the third line please add spaces between "1to 8correspond" to "1 to 8 correspond". |
| fig 10cap | ok | | At the end of the caption you should add also explanations of the symbols used: FZ, DZ, |
| table 1 | ok | | In the caption add a full stop at the end as highlighted. |
| table 1 | ok | | In the table header increase the width to include entirely the words "Fault zones", check |
| table 1 | ok | | Check also the French name "Faille" and correct it accordingly. |
| table 1 | ok | | Capitalize "pitch striation" to "Pitch striation". |
| table 1 | ok | | Add a space between the cardinal point and angular value every time has been |
| table 1 | ok | | Non constant" is not precise, I would use "variable". |
| table 2 | ok | | Please eliminate "vs" from the table header. |
| table 2 | ok | | Check also the nomenclature of the transects to be the same to the symbols adopted in |
| table 2 | ok | | In the caption it is not clear what do you mean for "bulk carbonates", "bulk measurements". |
| table 2 | ok | | Pay attention also to put the reference always to the singular form (es. micrite value, |

Comment N°1

*"What do you mean with the words "bulk rock"? Are you referring to the undeformed host rock*
*outside the damage zone, if so I think you should correct this and be more precise."*
**Done**.

"Bulkk rock" is a conventional wording commonly used in papers dealing with isotopes.
Anyway, to respect the reviewer's comment, we defined the word "bulk" as follows: "The Bulk
rock values are related to a non-selective sampling giving information on the whole rock isotopic values. These values do not capture the signature of isolated cement (Swart, 2015)."
**(lines 168-170**)

Comment N°2

*"I don't think that the word "sieve" is appropriate to describe the evolution of the hydraulic*
*properties of a fault zone. Maybe "valve" is more suitable, since you are describing media*
*behaving as a drain and then as a barrier."*

In this study we show that the **faults in carbonates are not valves**. The valve concept for fault
zones is not fully appropriate for carbonates. Indeed, the valve induces that it is alternatively
closed or open. Our study shows that the most appropriate concept would be a sieve, because
in this analogy, it is synchronously closed in places and open in other places, what rather reflects
the hydraulic behavior of the studied fault zones. We added in the text: "In this case, the most
appropriate concept would be a sieve, because in this analogy, it is synchronously closed in
places and open in other places." (**Lines 503-504**)

Comment N°3

*"The calculation of the permeability contribution is nice and to me it provides useful info*
*relative to the hydraulic evolution of the fault zone in time. I'm sorry for being so blunt here,*
*but maybe you should ground you statement and discussion on the field data. What I mean is*
*try to explain why you assigned such percentage contribution to the fault core or to the matrix*
*and so on... Maybe you can do this by evaluating the width of the fault core and damage zone*
*domains, or by estimating the fracture network volume."*

The comment of the reviewer is interesting, but cannot be realized in this manuscript because:
- Such a quantification in our field case is too much uncertain because the fault zones are too
heterogeneous                    to              do              this              exercise.
-      This      is      too      much      apart      from      the      subject      of      the      paper,
- this would be much too much for one paper and could rather represent another paper itself.

[revised manuscript text omitted]